# The influence of present-day regional surface mass balance uncertainties on the future evolution of the Antarctic Ice Sheet

Christian Wirths[1,2], Thomas F. Stocker[1,2], and Johannes C. R. Sutter[1,2]

[1]Climate and Environmental Physics, University of Bern, Switzerland
[2]Oeschger Centre for Climate Change Research, University of Bern, Switzerland

**Correspondence:** Christian Wirths (christian.wirths@unibe.ch)

**Abstract.** Rising global sea levels are one of many impacts the current anthropogenic global warming poses to humanity. The Antarctic Ice Sheet (AIS) has the potential to contribute several meters of sea level rise over the next few centuries. To predict future sea level rise contributions from ice sheets, both global and regional climate model (RCM) outputs are used as forcing in ice sheet model simulations. While the impact of different global models on future projections is well-studied, the effect of different regional models on the evolution of the AIS is mostly unknown. In our study, we present the impact of the choice of present-day reference RCM forcing on the evolution of the AIS. We used the Parallel Ice Sheet Model (PISM) to study the AIS in a quasi-equilibrium state and under future projections, combining present-day RCM output with global climate model projections. Our study suggests differences in projected Antarctic sea-level contributions, due to the choice of different present-day SMB and temperature baseline forcings of 10.6 mm in the year 2100 and 70.0 mm in 2300 under the RCP8.5 scenario. Those uncertainties are one order of magnitude smaller than what is estimated from uncertainties related to ice sheet and climate models. However, we observe an increase of RCM induced uncertainties over time and for higher emission scenarios. Additionally, our study shows that the complex relationship between the selected RCM baseline climatology and its impact on future sea-level rise is closely related to the stability of West Antarctic Ice Sheet (WAIS), particularly the dynamic response of Thwaites and Pine Island Glaciers. On millennial time scale, the choice of the RCM reference leads to ice volume differences up to 2.3 m and can result in the long-term collapse of the West Antarctic Ice Sheet.

## 1 Introduction

Global sea level rise is one of many climate impacts due to anthropogenic global warming (IPCC, 2022). Until the end of this century, model based estimates of global sea level rise range from 0.44–0.76 m for SSP3-4.5 (IPCC, 2022) threatening flood prone areas populated by over 420 million people (Hooijer and Vernimmen, 2021). Besides ocean thermal expansion, the melting of the Greenland and Antarctic Ice Sheets is the largest current contributor to sea level rise (IPCC, 2022). Despite the fact that the Antarctic Ice Sheet (AIS) is 7.8 times larger than the Greenland ice sheet (GrIS) (Morlighem et al., 2017, 2020), it currently contributes $3.6 \pm 0.5$ mm per decade to global sea level rise (Rignot et al., 2019), which is an almost two times smaller contribution compared to the GrIS (WCRP Global Sea Level Budget Group, 2018). However observations show that the Antarctic melt contribution has been accelerating (Otosaka et al., 2023) and could become the largest contributor by the

end of the century (Seroussi et al., 2020). The West Antarctic Ice Sheet (WAIS), which holds ice masses equivalent to ca. 3.3 m of sea level rise (Bamber et al., 2009), might undergo a rapid melt in the coming centuries due to its exposition to the so called marine ice sheet (MISI Schoof (2007); Pattyn (2018)) and ice cliff instabilities (MICI DeConto and Pollard (2016); Pattyn (2018)). Model based projections of Antarctic sea level contributions at the end of the century are associated with large uncertainties which can be reduced by careful calibration of the model (Bevan et al., 2023; Nias et al., 2019; Coulon et al., 2024; Edwards et al., 2019; Lowry et al., 2021). Ice Sheet Model projections of sea level equivalent ice volume change vary from -37 ± 34 mm to 96 ± 76 mm (Seroussi et al., 2020) for the RCP8.5 scenario. This uncertainty has many reasons spanning from largely unconstrained boundary conditions like basal friction (Bulthuis et al., 2019), ice shelf mass balance uncertainties arising from melt rate parameterizations and projected ocean temperature changes below the ice shelves and the evolution of the surface mass balance (SMB) (Coulon et al., 2024). Uncertainties in estimates of Antarctica's SMB - the net accumulation rate of snow and ice on the surface of Antarctica - have been discussed in detail recently (Mottram et al., 2021). Direct observations of accumulation and surface melt are sparse, while SMB products from regional climate models have a large spread ranging from 1961 ± 50 to 2519 ± 118 Gt yr$^{-1}$ (Mottram et al., 2021).

Uncertainties in Antarctica's current SMB affect prognostic or paleo ice sheet model (ISM) simulations which often use output from RCMs as a reference baseline forcing upon which climate anomalies are then added. The SMB data from RCMs are used to establish the present-day reference forcing and projections or reconstructions of future and past Antarctic climate change are added to this forcing via anomalies, usually computed against the pre-industrial or historical mean of the respective climate model (Sutter et al., 2019; Nowicki et al., 2020; Seroussi et al., 2020; Sutter et al., 2021; Reese et al., 2022). There is a variety of different RCM SMB-products available for ice sheet modeling, from which a selection is presented in Fig. 1. Those SMB fields do not only differ in the total SMB they produce for Antarctica but also in the spatial distribution (Mottram et al., 2021). However, many modelling studies utilize data from the RACMO model (Seroussi et al., 2020). This model is designed to simulate polar regions since it accounts for many relevant processes as snow drifting, melt, refreezing and percolation (van Wessem et al., 2018). However, there is not a specific reason to exclusively use one model, since other models are also designed to simulate polar regions by taking those processes into account (Mottram et al., 2021). A recent study by Li et al. (2023) suggests that the difference in SMB from different global models can have a substantial impact on the equilibrium state of the AIS. Furthermore, Seroussi et al. (2020) showed that there is also a significant impact of GCM differences in future projections.

In this study we investigate the response of the AIS to different forcings derived from a range of RCMs. We address the following questions: i) How does the choice of reference SMB and surface air temperature forcing affect the quasi-equilibrium state of the AIS? ii) How does this choice affect the evolution of the AIS under different climate scenarios? iii) Does this choice have an impact the projected stability of marine ice sheets?

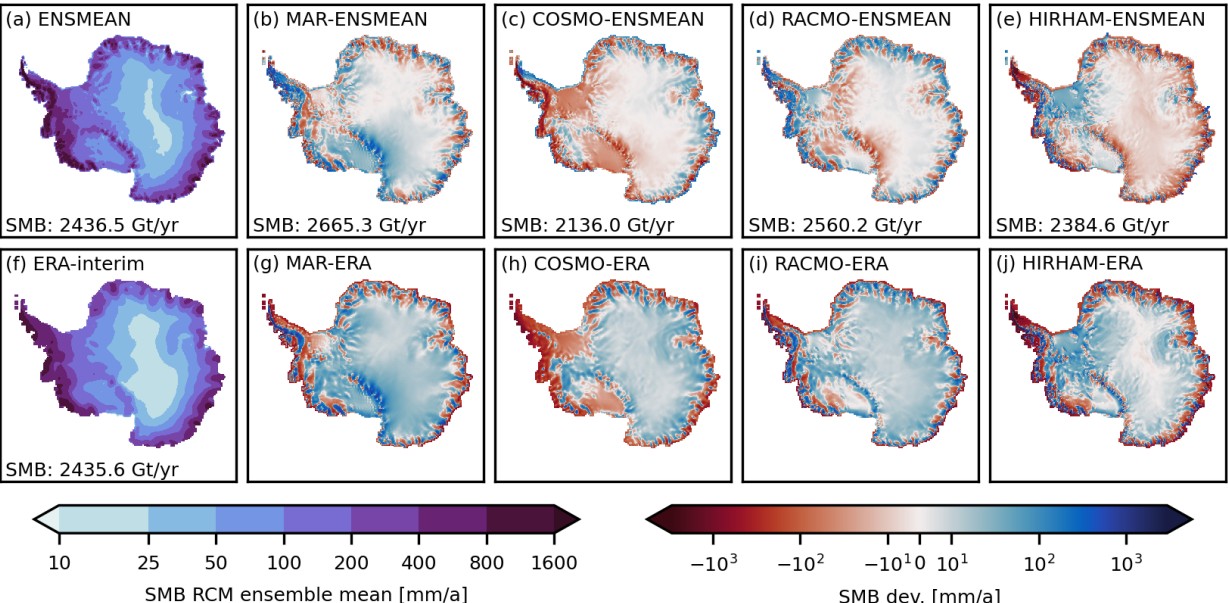

**Figure 1.** Surface mass balance (SMB) of the (a) multi-RCM mean and anomalies of the (b) MARv3.10 (Kittel et al., 2020), (c) COSMO-CLM[2] (Souverijns et al., 2019), (d) RACMO2.3p3 (van Dalum et al., 2021), and (e) HIRHAM5 (Hansen et al., 2022) regional climate model from this mean. SMB of the ERA-Interim dataset (Dee et al., 2011) (f) and SMB differences between (g) MARv3.10 (h) COSMO-CLM[2] (i) RACMO2.3p3 (j) HIRHAM5 and ERA-interim. The surface mass balance was averaged over the period from 1987-2016.

In the following sections, we introduce the RCM products utilized in our study and describe our simulation setup for the AIS. We then present the results from our long-term equilibrium simulations and future projections. Finally, we discuss the
implications of the choice of RCM product on the evolution and stability of the AIS.

## 2 Methods

To address the questions posed above, we consider two different model setups. In the first, we assess the equilibrium ice sheet response for a range of reference present-day baseline climate forcings. In the second, we investigate the imprint of the present-day baseline climatology on ice sheet model projections under a set of CMIP5 scenarios. In the following we
will briefly describe the ice sheet model setup (spinup, present-day equilibrium and prognostic simulations) and introduce the applied regional climate model forcing which will be used as a baseline climatology in all experiments.

### 2.1 SMB forcings

There is a considerable spread of model-based present-day surface mass balance (SMB) estimates (Mottram et al., 2021) (c.f. Fig. 1). To assess the equilibrium and transient ice sheet response to this spread we force the ISM with surface air temperature
and surface mass balance derived from four regional climate models (RCMs): MARv3.10 (Mottram et al., 2021; Kittel et al.,

|  | MARv3.10 | COSMO-CLM2 | RACMO2.3p3 | HIRHAM5 |
|---|---|---|---|---|
| Resolution | 35km | 25km | 27km | 12/50km |
|  | 24 layers | 40 layers | 40 layers | 31 layers |
| Surface scheme | SISVAT | CLM | internal snow model | - |
|  | based on (Ridder and Gallée, 1998) | (Oleson et al., 2013) | (Ettema et al., 2010) |  |
| Boundary conditions | ERA-Interim | ERA-Interim | ERA-Interim | ERA-Interim |
| Boundary interval | 6h | 6h | 6h | 6h |
| Nudging | Yes | Yes | Yes | No |
| Direct SMB | Yes | No | Yes | No |

**Table 1.** Summary of the regional climate model configuration for MARv3.10 (Kittel et al., 2020), COSMO-CLM2 (Souverijns et al., 2019), RACMO2.3p3 (van Dalum et al., 2021), and HIRHAM5 (Hansen et al., 2021).

2020), COSMO-CLM$^2$ (Souverijns et al., 2019), RACMO2.3p3 (van Dalum et al., 2021) and HIRHAM5 (Hansen et al., 2022). A general overview of these models as well as the applied forcings, parametrizations, and submodules are provided in Table 1. An additional SMB comparison for the individual IMBIE drainage basins can be found in Fig. A1. All four models were forced with the ERA-Interim reanalysis (Dee et al., 2011) at the domain boundaries, with the MARv3.10, RACMO2.3p3 and

COSMO-CLM$^2$ model being nudged into the domain by applying upper air relaxation (van de Berg and Medley, 2016). In contrast HIRHAM evolved freely and was only forced at the boundary of the domain (Mottram et al., 2021). RACMO and MAR have optimized subsurface snow and ice schemes to take meltwater, refreezing and retention into account. Additionally, RACMO accounts for wind driven erosion and sublimation of blown-off snow (Lenaerts et al., 2012). A more detailed discussion and comparison of the applied RCMs can be found in Mottram et al. (2021). Please note that Mottram et al. (2021) used

data from RACMO2.3p2 while this study uses RACMO2.3p3.

To conduct our study, we obtained the SMB-forcing either directly from the model output (MAR and RACMO) or calculated SMB as described by Mottram et al. (2021) from precipitation, evaporation, and sublimation. Please note that for COSMO the difference between precipitation and evaporation and for HIRHAM the difference between precipitation, sublimation, and

evaporation is used to calculate the SMB. We then calculated the climatic mean of the SMB and surface temperature for the common period from 1987 to 2016 and bi-linearly regridded the data to the PISM domain at 8km resolution. The SMB ensemble mean of those RCMs together with the deviation of the individual RCM SMBs from this mean and the total mass balance are shown in Fig. 1. Please note that due to the regridding as well as the chosen ice mask we expect the total SBM to differ slightly from values of other publications (Hansen et al., 2022).

## 2.2 Ice sheet model set up

To simulate the response of the Antarctic Ice Sheet, we employ the thermodynamially-coupled Parallel Ice Sheet Model (PISM) (Bueler and Brown, 2009; Winkelmann et al., 2011). PISM is used in a hybrid mode using the shallow ice (SIA) and shallow shelf approximation (SSA) to efficiently simulate the slow interior ice sheets as well as the fast ice-streams of outlet glaciers and shelves. The stress at which the ice starts to slide by deformation of the till layer, also called yield stress

$$\tau_C = c_0 + tan(\phi) \, N_{till} \tag{1}$$

is calculated following the Mohr-Coulomb law (Cuffey and Paterson, 2010), with the till friction angle $\phi$, the effective till pressure $N_{till}$ and the "till cohesion" $c_0$. The till friction angle depends on the bed topography and is linearly interpolated between $\phi_{min}$ and $\phi_{max}$ for bed elevations between $b_{min}$ and $b_{max}$ with the gradient $M = (\phi_{max} - \phi_{min})/(b_{max} - b_{min})$ by

$$\phi(x,y) = \begin{cases} \phi_{min}, & b(x,y) \le b_{min}, \\ \phi_{min} + (b(x,y) - b_{min})M, & b_{min} < b(x,y) < b_{max}, \\ \phi_{max}, & b_{max} \le b(x,y) \end{cases} \tag{2}$$

(Aschwanden et al., 2013; Winkelmann et al., 2011; Martin et al., 2011). Sub-shelf melt and refreezing at the ice ocean interface is calculated using PICO (Reese et al., 2018), an ocean box model which mimics the overturning circulation in the cavities below the ice shelf.

In this study we consider two model set ups (c.f. Fig. 2): (i) A long term (30 ka) present-day equilibrium simulations with a constant present-day forcing, and (ii), centennial projections until the year 2300 applying climate anomalies from HadGEM-ES2 (Jones et al., 2018) for the RCP2.6, RCP4.5 and RCP8.5 scenarios. In both cases, the model is initialized from the BedMachine (Morlighem et al., 2020) bedrock topography and ice thickness. Additionally, geothermal heat-flux data by Shapiro and Ritzwoller (2004) is applied.

### 2.2.1 Long term present-day equilibrium simulations

To explore the equilibrium response of the Antarctic Ice Sheet to the four RCM forcings considered here, we perform a set of long term present-day equilibrium simulations. As an initial step we perform a 200 ka thermal spinup during which the ice surface elevation is fixed and the ice sheet is forced with geothermal heat flux from Shapiro and Ritzwoller (2004) and surface air temperature from MARv3.10 (Kittel et al., 2020), COSMO-CLM2 (Souverijns et al., 2019), RACMO2.3p3 (van Dalum et al., 2021), and HIRHAM5 (Hansen et al., 2021) respectively (c.f. Fig. 2). After the thermal spinup, we restart the ice sheet model from the thermal spinup for each RCM individually. We apply constant SMB and temperature forcing fields from the individual RCMs and let the model freely evolve for 30 000 years (c.f. Fig. 2) at a 16 km resolution. A summary of the described simulation setups can be found in Table B2. To compute basal melting underneath the ice shelves, we additionally force the

model with ocean temperature and salinity from the observational climatology provided in the ISMIP6 protocol (Nowicki et al., 2020). For every RCM forcing we run 24 simulations with different combinations of the shallow ice approximation enhancement factor $sia_e$, the pseudo plastic parameter $pQ$ (used in the pseudo plastic sliding law), the minimum till friction angle $\phi_{min}$ (the angle we assume for marine basins below $-700$m as in Albrecht et al. (2020b)), and the heat conductivity at the ice-ocean interface $\gamma$. For a detailed list of the parameters see Table 2 and B1. The parameter combinations were selected based on the model skill to reproduce the observed present-day ice thickness, surface velocities and grounding line (Morlighem et al., 2020) after 15 000 years under constant RACMO forcing. An additional constraint was the sea-level equivalent ice volume after 15 000 years under constant RACMO2.3p3 forcing. Here we penalized deviations from present-day estimates of Antarctica's current ice volume. The comparison and scoring was performed following the scoring method by Albrecht et al. (2020b). The chosen spinup method and parameter selection process is not necessarily the most rigid in terms of producing a good match with present-day ice sheet observations. A much better fit can be achieved via inversion or iterative optimization of e.g. a sliding parameter as e.g. done in Pollard and DeConto (2012) and Li et al. (2023). The latter method can either be applied for only one forcing or for all forcings individually. Nevertheless, both approaches have drawbacks when applied to our context. On the one hand, individually applying the inversion or iterative optimization technique to each forcing field would select basal sliding properties for each forcing field in a manner that converges toward a state closest to observational data, thereby concealing disparities inherent to the forcing fields within the basal sliding coefficient/parameterization. This evidently runs counter to the core objective of this study, which centers around discerning the distinct influences exerted by individual forcing fields on the ice sheet.

Conversely, fine-tuning the basal sliding coefficients exclusively to a specific forcing field carries the risk of overly tailoring the model to that particular forcing field, potentially leading to overfitting. Such meticulously calibrated basal sliding conditions might not be suitable for other forcing fields, potentially resulting in unrealistic ice sheet dynamics when subjected to those fields. Generally, this trade-off necessitates consideration for each instance of tuning and parameter selection. Nonetheless, our approach employs consistent basal sliding conditions across all forcing fields, while also calibrating for overarching global parameters. This not only reduces the computational cost, but also mitigates the possibility of introducing artifacts by excessively tuning the model to any single forcing field.

### 2.2.2 Centennial projections of Antarctic Ice Sheet evolution.

In order to assess the impact of present-day baseline climate forcings on centennial sea level projections we perform simulations until the year 2300 applying transient annual climate anomalies derived from HadGem2-ES (Jones et al., 2018) RCP-scenarios. Starting from the BEDMACHINE (Morlighem et al., 2020) bedrock topography and ice thickness, we perform a 200 ka thermal spinup under a constant pre-industrial (PI) temperature forcing. The pre-industrial temperature forcing is the mean of all four RCMs together with HadGem2-ES (Jones et al., 2018) pre-industrial to present-day anomalies. After the thermal spinup we restart the model on 8km resolution for 12 different parameter configuration (c.f. Tab. 2 and Tab. C1) to evolve under constant RCM-mean PI forcing for 300 years to relax the initial shock after the ice can flow freely, which leads to 12 initial pre-industrial ice sheet geometries, one for each parameter configuration. At the end of the 300 year relaxation each simulation

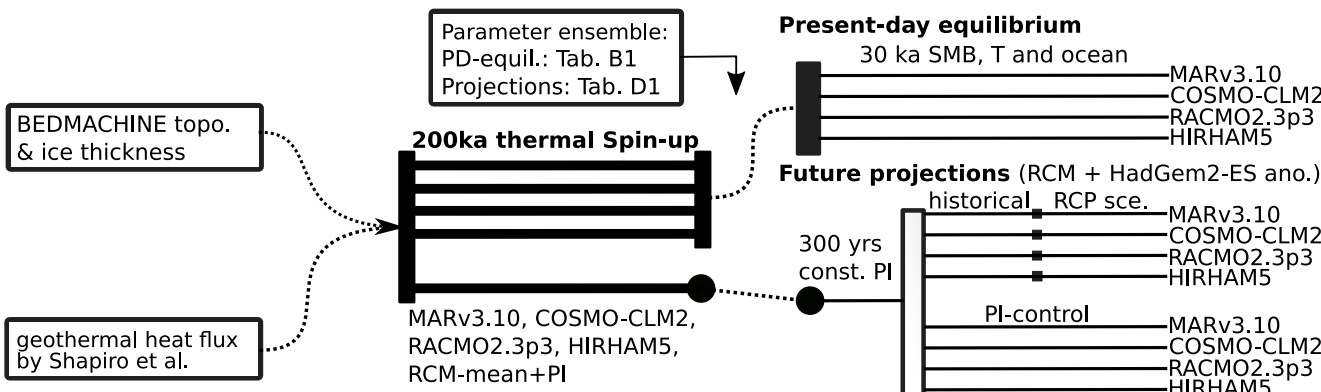

**Figure 2.** Illustration of the present-day equilibrium simulation and future projections setup: Starting from present-day ice sheet geometry, a 200 ka thermal spinup for each RCM forcing is performed individually (indicated by bold solid lines). During the thermal spinup, the ice sheet geometry is fixed. For the present-day equilibrium 24 individual simulations (cmp. Tab. B1) are performed for each of the four constant RCM forcing fields and allowing the model to freely evolve for 30 000 years. Freely evolving ensemble simulations are indicated by thin solid lines. For illustrative reasons same start-/end-states are connected by dashed lines. For the future projections a 200 ka thermal spinup under the mean forcing of all four RCMs together with HadGem2-ES (Jones et al., 2018) PI-PD anomalies is performed. Starting after the thermal spinup, a 300 years relaxation for 12 individual parameter configurations (cmp. Tab. C1), using constant PI forcing, is performed. Then, we simulate the historical period followed by the three RCP scenarios for every combination of the RCM forcing fields. All simulation setups are summarized in Tab. B2.

has an annual sea-level contribution of below 0.15 mm/yr and a grounding line close to present-day observation. From these initial geometries, we then branch off four 440 year long PI-control simulations, one for each RCM baseline. Additionally, we simulate the historical period from 1860-2005 C.E. (c.f. Fig. 2) for each RCM baseline. For the historical period and future

projections, transient annual mean surface air temperature and SMB anomalies from HadGEM2-ES (Jones et al., 2018) are added to the respective present-day RCM climatology to produce the climate forcing until the year 2300. Likewise ocean temperature and salinity anomalies are added to the ISMIP6 ocean forcing. After the historical period we branch the individual simulations into three different forcing scenarios and let the model evolve until the year 2300. The three different branches consist of the RCP2.6, RCP4.5, and RCP8.5 scenarios which use the anomalies from the HadGEM2-ES model. Similar to

the present-day equilibrium runs, we choose an ensemble of different parameter configurations. The complete list of selected parameter configurations is provided in Table 2 and table C1. Please note that since the model spinup (thermal + constant PI forcing) we chose here is relatively simple compared to e.g. inversion or a full glacial interglacial paleo-spinup, remaining model biases are to be expected. In consequence, the initial ice sheet configuration lacks a realistic thermal state, and as we do not use iterative optimization (see above) model deviations with respect to ice thickness, between a simulated present-day

state and observations, can be large, which is typical for continental scale model setups not employing inversion methods (see

| Setup | $sia_e$ | $pQ$ | $\gamma \, [\times 10^{-5}]$ | $\phi_{till \, min}$ |
|---|---|---|---|---|
| PD-equilibrium | 1.00, 1.25, 1.50 | 0.75, 0.80 | 2.0, 2.5, 3.0, 3.5, 4.0 | 2, 4, 6 |
| Future projections | 1.00, 1.25 | 0.75, 0.8, 0.85 | 2.0, 2.5, 3.0 | 4, 6, 8 |

**Table 2.** Chosen parameter space for the shallow ice approximation enhancement factor $sia_e$, the pseudoplastic $pQ$ factor, the heat conductivity at the ice ocean interface $\gamma$, and the minimum till friction angle $\phi_{till \, min}$ for the present-day equilibrium runs as well as the future projections. A detailed list of every individual parameter configuration can be found in Tab. B1 and Tab. C1.

e.g. (Reese et al., 2023)). However, as these kinds of simple spinup (thermal + constant PI forcing) routines have been used in the past (Seroussi et al., 2019; Levermann et al., 2020; Sutter et al., 2023) we considered this to be a valid approach to assess the impact of present-day climate forcing uncertainties on future and equilibrium ice sheet evolution in typical model setups. Therefore, this setup is not designed to give robust projections on future Antarctic sea level contributions but rather serves to estimate uncertainties arising from different RCM forcing fields.

## 3 Results

In this section we present the evolution of ice volume and area under constant present-day forcing and centennial future projections. We further discuss the imprint of SMB forcing differences on ice thickness and grounding line position at the end of the respective simulations.

### 3.1 Impact of RCM forcing on the present-day quasi-equilibrium state

#### 3.1.1 Impact of RCM forcing on global ice mass and extent

Starting from the present-day observations the total ice volume (c.f. Fig. 3 a-d ) undergoes an initial shock after which the rate of change (c.f. Fig. 3 e-h ) converges towards zero. After 30 000 years of simulation the median change in ice volume is negative (i.e. ice loss) for three (COSMO: -1.71 m, RACMO: -1.20 m, HIRHAM: -1.63 m) out of the four models. The simulations which apply the RACMO forcing show the least change in ice volume. In contrast, ice sheet simulations using the MAR forcing exhibit an increase in meadian SLE ice volume compared to present-day observations by 0.55 meters. Although the annual rate of mass change converges towards zero, the ensemble spread indicates that the ice sheets are still undergoing small fluctuations in ice sheet mass between -0.20 mm/a and 0.10 mm/a (current AIS sea level contribution is $\approx 0.3$ mm per year (Shepherd et al., 2018)). The initial shock observed in the ice volume change is also reflected in the respective ice area change of floating and grounded ice. For all models an increase in the grounded ice area with a coinciding decrease in the ice shelf area is observable - a result of an advancing grounding line in the Filchner-Ronne ice shelf (c.f. Fig. 4). The median decrease in ice shelf area varies between -43% and -36% with COSMO showing the largest and MAR the smallest decrease. This might be due to a stronger grounding line retreat, especially when WAIS is unstable, allowing for larger shelves. Additionally, we observe a grounding line advance with MAR showing the largest (3.4%) and COSMO the smallest (2.8%) corresponding

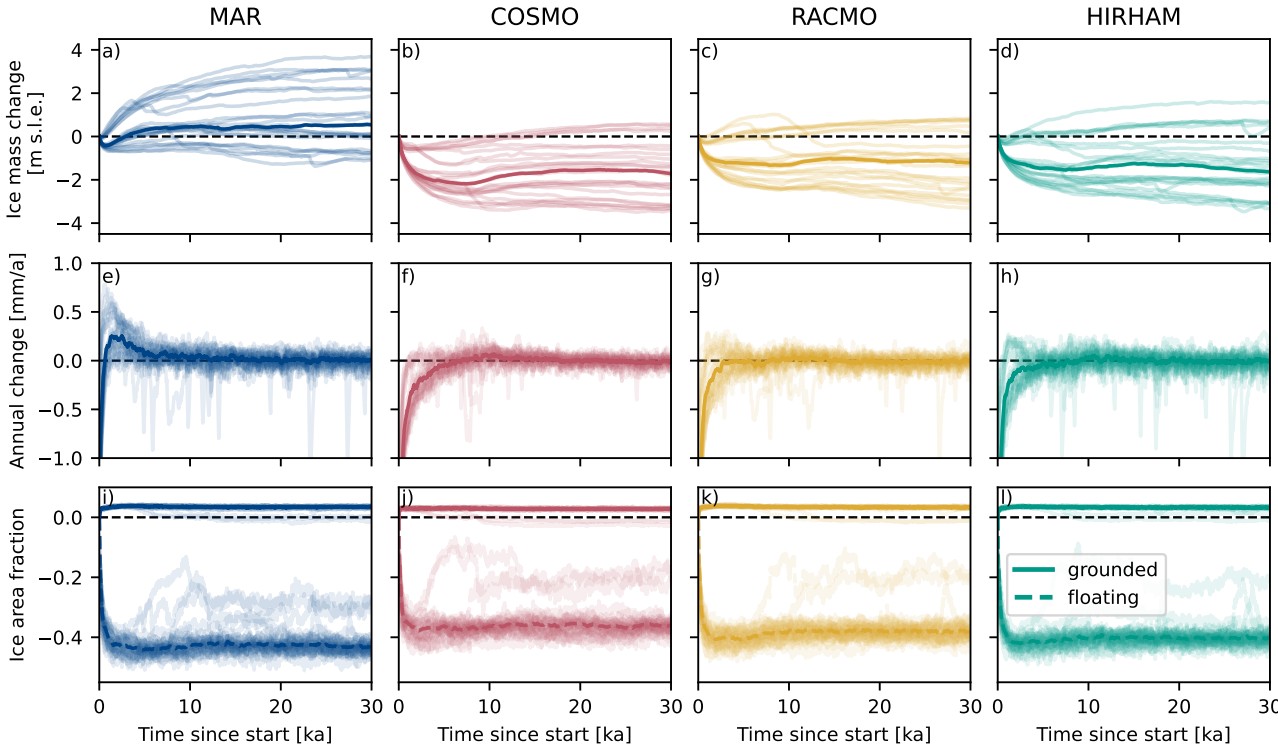

**Figure 3.** Time series of the total ice mass above flotation change since the start of the simulation (a-d), the annual rate of ice mass above flotation change (e-h), and the fraction of grounded (solid line) and floating (dashed line) ice area (i-l) relative to observations for the four different RCM forcing fields. Bold line shows ensemble median while shaded lines indicate the individual ensemble members.

growth in grounded ice area. The time series presenting the impact of the ensemble mean RCM forcing can be found in Fig. B3.

### 3.1.2 Impact of RCM forcing on regional ice cover

We now turn to regional characteristics of the simulated ice sheet. When we compare the ice thickness between present-day observations and our simulations after 30 000 years a distinct pattern arises which is independent of ensemble member and RCM forcing field (c.f. Fig. B1). All our simulations show a strong negative ice thickness anomaly in the WAIS which is mainly driven by ice sheet model parameterisation. Specifically the applied heuristics calculating the till friction angle, results in anomalies with respect to present-day observations. This is a persistent model bias for the setup employed here and in other studies (Martin et al., 2011; Albrecht et al., 2020a; Sutter et al., 2023; Reese et al., 2023). Additionally, all realizations show ice loss at the EAIS margins with substantial coastal ice sheet thinning in George V and Wilkes Land. In contrast, larger ice thickness is simulated in Oates Land along the Transantarctic mountains and also on the Antarctic Peninsula, the Ellsworth and Scott Mountains as well as at the Shackleton range. The inter-model differences caused by the different RCM-forcings

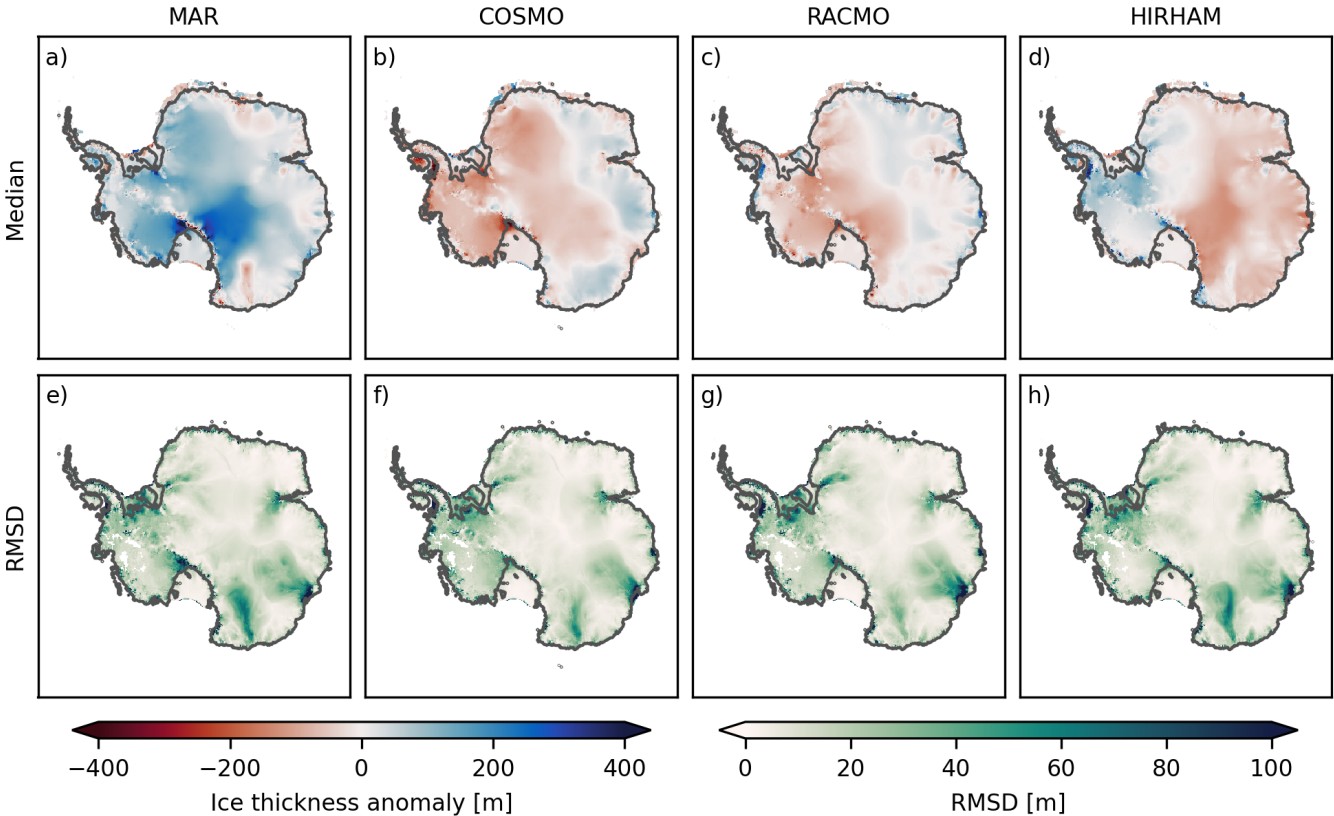

**Figure 4.** Ice thickness anomalies $\Delta h$ from common mean (a-d), position of the simulated (grey) grounding line for the present-day equilibrium simulations. Root mean square deviation (RMSD) of the individual ensemble members from the median (e-h). The difference between the common mean and the result of the forcing ensemble mean are illustrated in Fig. B4.

(mainly the impact of SMB forcing differences) are around four times smaller compared to overall model bias (effect of ice sheet model spinup and parameter choices mentioned above).

Therefore, we explicitly illustrate the differences between the individual RCM forcing sets in Fig. 4. Panel (a-d) depicts the ice thickness differences $\Delta h$ for each individual RCM forcing set from the common mean of all four forcing sets. At every grid cell (i,j) and for a given $RCM \in \{MAR, COSMO, RACMO, HIRHAM\}$, $\Delta h$ is given as:

$$\Delta h_{i,j}^{RCM} = h_{i,j}^{RCM} - \frac{h_{i,j}^{MAR} + h_{i,j}^{COSMO} + h_{i,j}^{RACMO} + h_{i,j}^{HIRHAM}}{4}. \tag{3}$$

Due to its overall higher SMB, the simulations forced with the MAR data show positive $\Delta h$, except for small negative 210  $\Delta h$ values at the Princess Ragnhild Coast and the George V Land. Additionally, simulations forced with SMB and surface temperature from COSMO, RACMO, and HIRHAM show diverging $\Delta h$ patterns between East and West Antarctica which

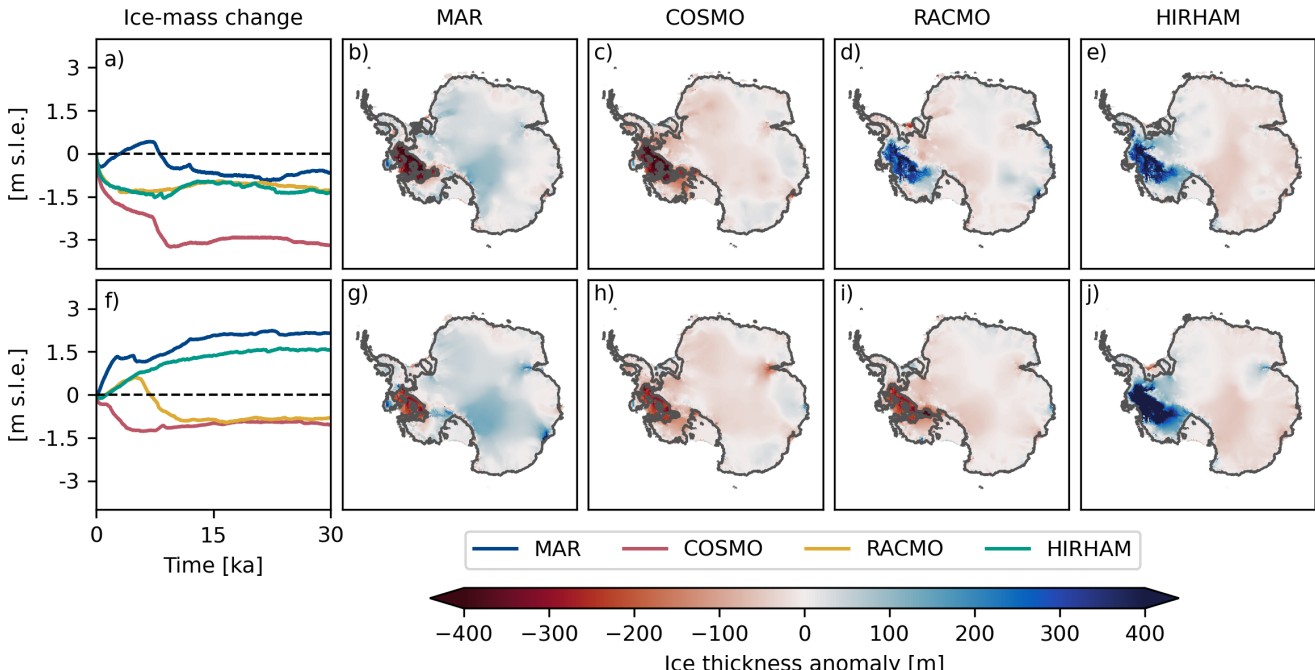

**Figure 5.** Evolution of the sea level change relevant ice masses (a,f) over the simulation period and ice thickness differences from the common mean as well as grounding line position (grey line) at the end of the simulation (b-e, g-j). Panel a-e show the results of the simulation with $\phi_{min} = 2°, sia_e = 1.00, pQ = 0.8, \gamma = 4.0 \times 10^{-5}$. Panel f-j show the results the simulation with $\phi_{min} = 6°, sia_e = 1.50, pQ = 0.80, \gamma = 2.5 \times 10^{-5}$.

generally agree with the differences in their undelining SMB forcing. For all four forcing fields we illustrate the regions of highest ensemble variability in ice thickness (Fig. 4 (e-f)). We do this by calculating the root-mean-square-deviation (RMSD) of $\Delta h$ from the parameter ensemble mean for each RCM. Wilkes and Aurora Subglacial Basins are the regions of highest

ensemble variability. Additionally, the Shackleton Range shows large ice thickness variability across all four forcing fields. The HIRHAM forcing field exhibits high variability in Ellsworth Land.

### 3.1.3   Amplification of ISM uncertainties due to RCM selection

In order to illustrate differences in individual model runs and their evolution under constant present-day climate conditions,

we illustrate the change in ice volume and ice thickness differences at the end of the simulation (after 30 ka) for parameter configuration No. 6 and 24 (c.f. Tab. B1) in Fig. 5.

   Under parameter configuration No. 6 (panel a-e), the simulations forced by RACMO and HIRHAM undergo initial ice loss until they mostly stabilize for the rest of the simulation period. In contrast, the simulations forced by COSMO and MAR show initial ice mass increase (MAR) or decrease (COSMO) for the first 7000 years. Then both simulations exhibit a dynamical

reorganization of WAIS, leading to a strong grounding line retreat and ice mass loss of over 1 m s.l.e.. While in the COSMO forced simulation, the retreat of the grounding line slowly begins from the start of the simulation for several thousand years then leading to the fast collapse of WAIS (c.f. Fig. B5), the MAR simulation exhibits no differences in grounding line position to the HIRHAM or RACMO forced simulations for the first 6000 years (c.f. Fig. B5). Similar behavior can be observed for parameter configuration No. 24 (panel j-f), where the relatively small differences in SMB forcing lead to large differences in

the simulation outcome. In this case, all but the simulation forced by HIRHAM exhibit a dynamical collapse of WAIS.

## 3.2  Projections of present-day RCM imprint on centennial Antarctic Ice Sheet evolution

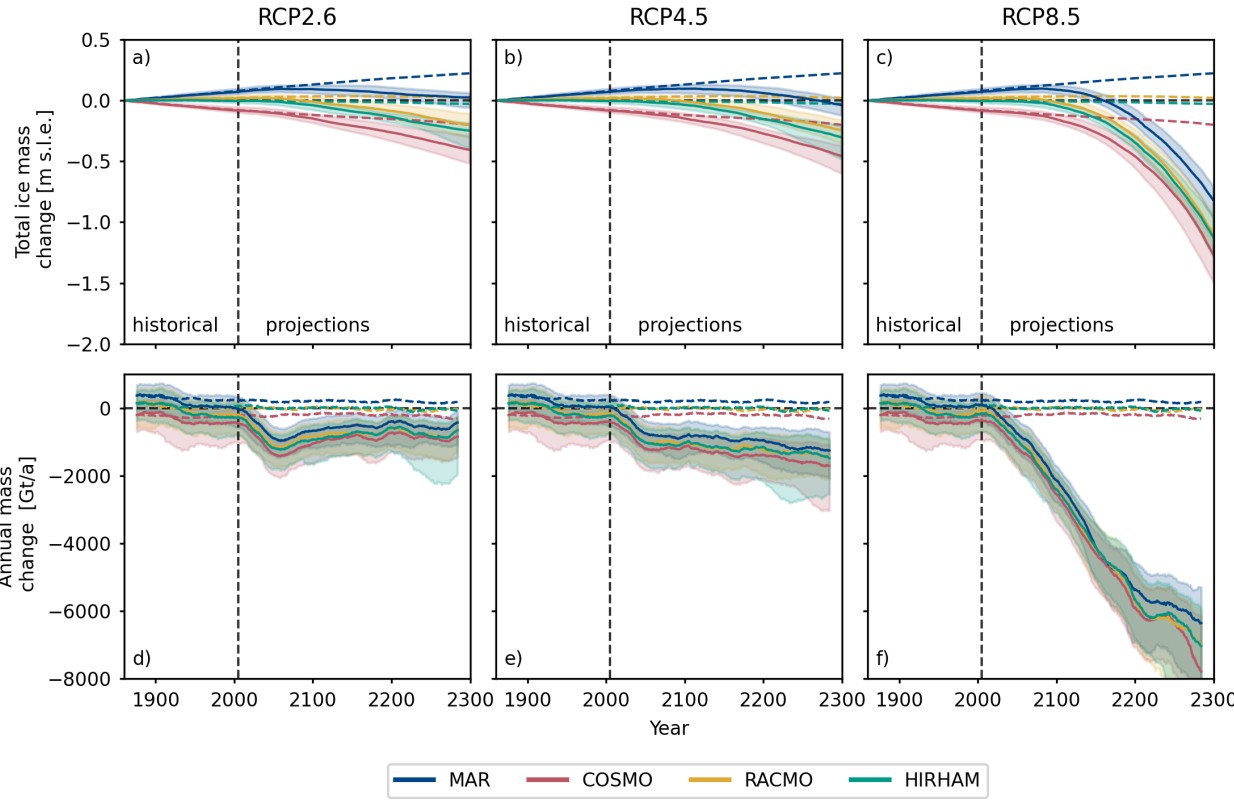

**Figure 6.** Time series of the median (solid lines) total ice mass above flotation change (a-c) and the annual rate of change (d-f) for the four different RCM forcing fields and the RCP2.6, RCP4.5, and RCP8.5 climate scenarios. Dashed lines represent the median PI-control simulations. Shadings indicate the 5th to 95th quantile.

To investigate the effect of differences in the underlying RCM baseline data in climate scenarios we simulated the historical period from 1860 to 2005, followed by the RCP2.6, RCP4.5, and RCP8.5 scenario until 2300, by additionally applying GCM anomalies to the baseline RCM forcing on a 8 km grid resolution. The evolution of the total ice volume is shown in Fig. 6.

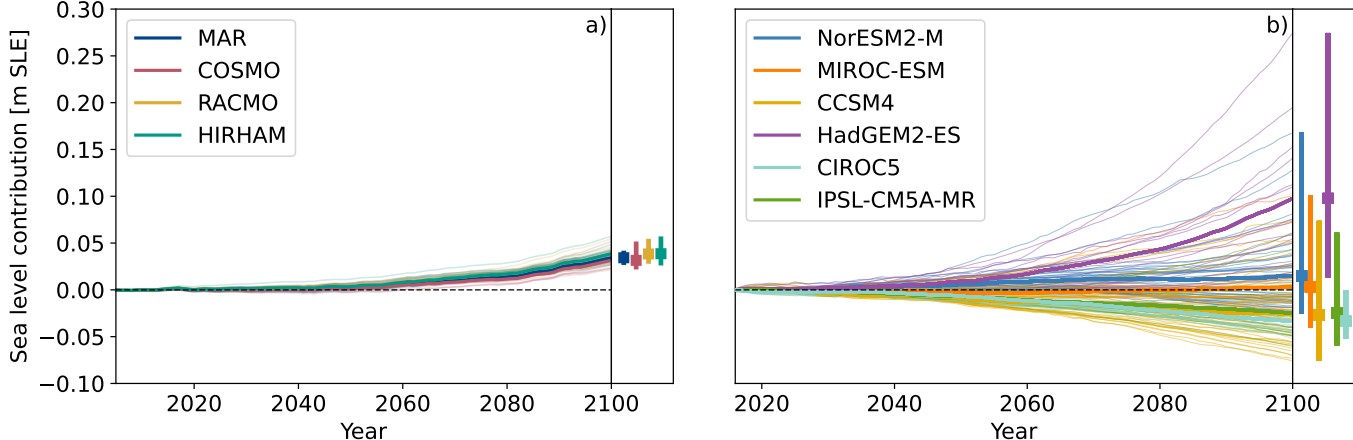

**Figure 7.** PI-control corrected antarctic sea level rise contribution for the RCP8.5 scenario with different RCM present-day fields (a) as well as from ISMIP6 (Seroussi et al., 2020) (b) until the year 2100. Thin lines show individual simulations, bold lines the mean state for the different models. Note that in our study, GCM anomalies from HadGEM2-ES (Jones et al., 2018) were used together with RCM forcing fields. Find a zoomed version of a) in Fig. C1 and an extension until the year 2300 in Fig. C2.

Starting from 1860, the historical and PI-control simulations are almost identical until the second half of the 20th century, where ice mass loss is increased in the historical simulations when compared to the respective PI-control (c.f. Fig. 6). For the PI-control simulations and the early historical period simulations applying MAR forcing tend to produce a slightly positive mass balance. In contrast, simulations applying COSMO tend to have a slightly negative and simulations applying RACMO or HIRHAM forcing tend to have a neutral mass balance. From 2005 onwards, an increase in ice loss is modeled in all three

centennial projection scenarios. However, the RCP2.6 scenario seems to stabilize after the first half of the 21st century. The RCP4.5 scenario also shows initial stabilization but then tends to slowly increase in ice loss again during the 23rd century. Contrasting, the RCP8.5 scenario shows no stabilization of ice loss, which constantly increases until the end of the 23rd century. Due to the rather limited ice loss in the RCP2.6 and RCP4.5 scenarios, we observe similar results with respect to the total ice mass above flotation change. In contrast, the RCP8.5 scenario yields over one meter more of ice mass change at the end of

the simulation, independent of the applied RCM present-day baseline forcing.

### 3.2.1    Imprint of RCM present-day forcing on sea level rise projections

In the following we show the uncertainties in sea-level rise projections which arise from the choice of RCM baseline data. Figure 7 illustrates the PI-control corrected sea level rise contribution of the individual simulations until the year 2100 in the

RCP8.5 scenario contrasted with the results from Seroussi et al. (2020). We estimate the maximum difference in sea level rise contribution between simulations with different RCM reference forcings. Therefore we calculate for every member ($par$) of the parameter ensemble the maximum sea level contribution difference

$$\Delta slr_{par}^{max} = \max_{\mu,\nu \in \Omega} \left( slr_{par}^{\mu} - slr_{par}^{\nu} \right). \tag{4}$$

Here, $\Omega = \{MAR, COSMO, RACMO, HIRHAM\}$ denotes the list of potential RCM reference forcing. The calculation
is carried out on two time series data sets: the raw centennial sea level rise projection and the projection with the respective
pre-industrial control simulation subtracted. The simulated differences in sea level rise can be approximated and decomposed
into two components: variations arising solely from differences in the Regional Climate Model (RCM) forcing, and variations
arising from the interplay between RCM forcing differences and transient anomalies in the Global Climate Model (GCM). In
the pre-industrial control simulations, the GCM anomalies remain constant over time. Consequently, the pre-industrial control
simulation exhibits sea level changes resulting purely from differences in the RCM forcing. By subtracting the pre-industrial
control simulation from the centennial projections, the interplay between RCM forcing differences and GCM anomalies is isolated. This allows for the analysis of how the RCM forcing variations interact with the transient GCM anomalies to influence
sea level rise projections.

The resulting $\Delta slr^{max}$ for all three RCP scenarios in the PI-corrected (blue) and uncorrected (orange) case are illustrated
in Fig. 8 for the year 2100 (a) and 2300 (b). In both cases we only consider differences which arose after 2005. For the year
2100 the PI-control corrected median $\Delta slr^{max}$ is 9.2 mm for the RCP2.6, 10.5 mm for RCP4.5 and 10.6 mm for RCP8.5,
which is around one order of magnitude smaller than the uncorrected $\Delta slr^{max}$. The $\Delta slr^{max}$ is mostly independent of the
projection scenario, due to the fact that our simulations only differ minimally until the year 2100 (c.f. Fig. 6). In the year 2300
the PI-control corrected $\Delta slr^{max}$ is around one third of the uncorrected $\Delta slr^{max}$. Additionally one can observe an increase
in the median $\Delta slr^{max}$ from 36.1 mm for RCP2.6 to 52.6 mm for RCP4.5 and to 70.0 mm for RCP8.5. Please not the shaded
outlier in the PI-control corrected $\Delta slr^{max}$. This outlier shows the $\Delta slr^{max}$ for parameter configuration No. 12 (cmp. Tab.
C1), which shows a partial collapse of the WAIS in the PI-control simulation, when forced with COSMO (c.f. Fig. C6). This is
further discussed in Section 3.2.4. Since this configuration resulted in reasonable PI-control grounding lines for all other RCM
forcings we did not exclude it from our ensemble.

### 3.2.2 Impact of reference RCM on regional ice thickness

The spatial distribution of $\Delta h$ (thickness deviation from the common mean; cmp. Eq. 3) and the simulated grounding line
position at the year 2300 is illustrated in Fig. 9 for Thwaites and Pine Island glacier and in Fig. C3 for the entire AIS. In all
three scenarios a similar ice sheet response is simulated: Simulations forced with MAR generally show higher $\Delta h$ especially
at Thwaites glacier (c.f. Fig. 9) as well as along the Transantarctic mountains and in the Filchner ice shelf (c.f. Fig. C3). In
contrast, simulations forced by COSMO mainly depict negative $\Delta h$. Simulations forced by RACMO or HIRHAM show a
generally diverging pattern with mostly positive $\Delta h$ over Thwaites glacier for HIRHAM and negative for RACMO. In East
Antarctica the opposite is observable with positive $\Delta h$ in simulations forced by RACMO and negative $\Delta h$ for HIRHAM.

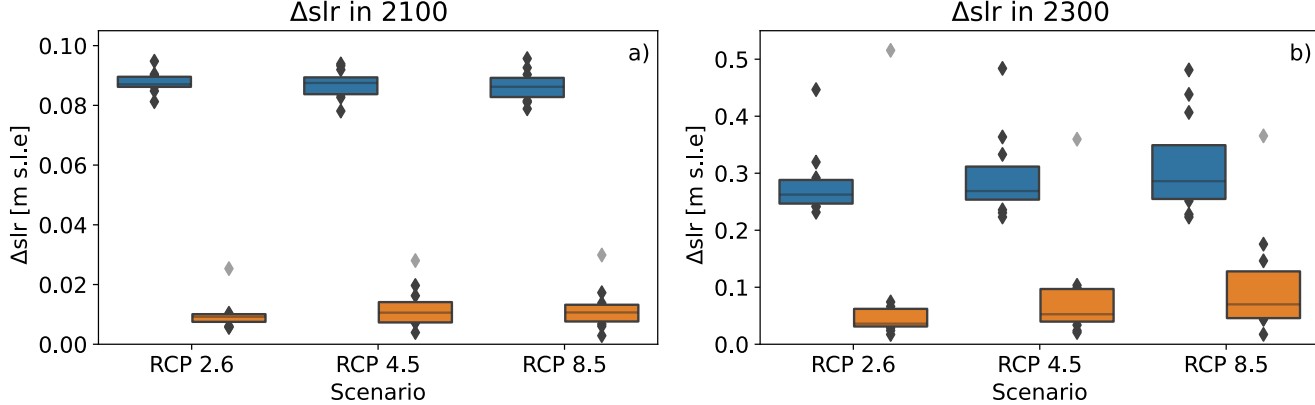

**Figure 8.** $\Delta slr_{par}^{max}$ in the year 2100 (a) and 2300 (b) for the PI-control corrected (orange) and uncorrected (blue) case. Boxes indicate the 25th to 75th percentile as well as the median. Diamonds indicate values outside of the 25th to 75th percentile. The shaded diamonds mark the $\Delta slr_{par}^{max}$ for ensemble configuration No. 12 (cmp. Tab C1), which shows a WAIS collapse for the COMSO forcing in the PI-control simulation (c.f. C6).

Although the overall patterns of ice $\Delta h$ are rather similar, for all three RCP scenario, major differences in ice thickness are observable at Thwaites glacier. For MAR simulations, $\Delta h$ is substantially larger for the RCP8.5 scenario than for the RCP4.5 scenario. On the other hand COSMO yields, a lower $\Delta h$ for the RCP8.5 than RCP4.5 scenario. The changes in $\Delta h$ between the RCP4.5 and RCP8.5 scenario are less pronounced for RACMO and HIRHAM, where RACMO shows a decrease and HIRHAM shows an increase in $\Delta h$.

### 3.2.3 Ensemble sensitivity to reference RCM forcing

The sensitivity of $\Delta h$ to ice sheet model parameters under a single RCM baseline reference forcing is shown by the root-mean-square-deviation (RSMD) depicted in Figs. 10 and C4. This allows the identification of regions where the chosen ice sheet model parametrization has a high impact on simulated ice thickness differences. For for all scenarios the largest parameter sensitivity can be observed at Thwaites and Pine Island glacier. For the RCP8.5 scenario, Thwaites and Pine Island Glacier still exhibit the strongest parameter sensitivity. However, the Filchner-Ronne and Ross shelves show significant additional parameter sensitivity. A large $\Delta h$ does not necessarily also imply a large parameter-sensitivity. The Transantartic mountain range for example shows high absolute $\Delta h$ values between different baseline models, while the parameter sensitivity is particularly small.

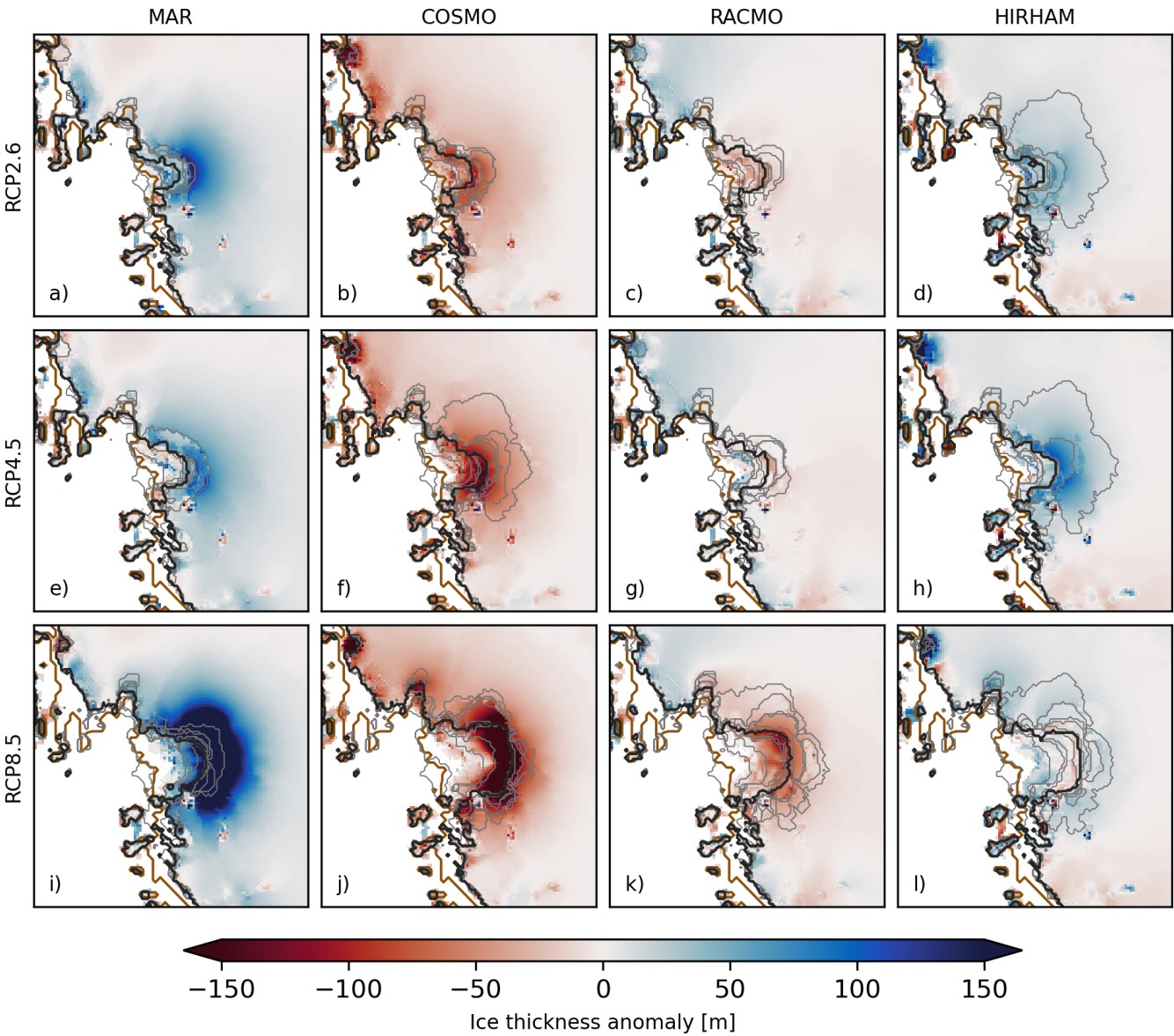

**Figure 9.** Median ice thickness anomalies from common mean for RCP2.6 (a-d), RCP4.5 (e-h), and RCP8.5 (i-l) together with the position of the simulated median (black) and observed (brown) grounding line at the year 2300. The thin grey lines indicate the simulated grounding line position of the individual ensemble members (cmp. Tab. C1). Please be aware that the used color-scale in this section is smaller than in the previous sections. Please be aware of the changed color-scale w.r.t. Fig. 4.

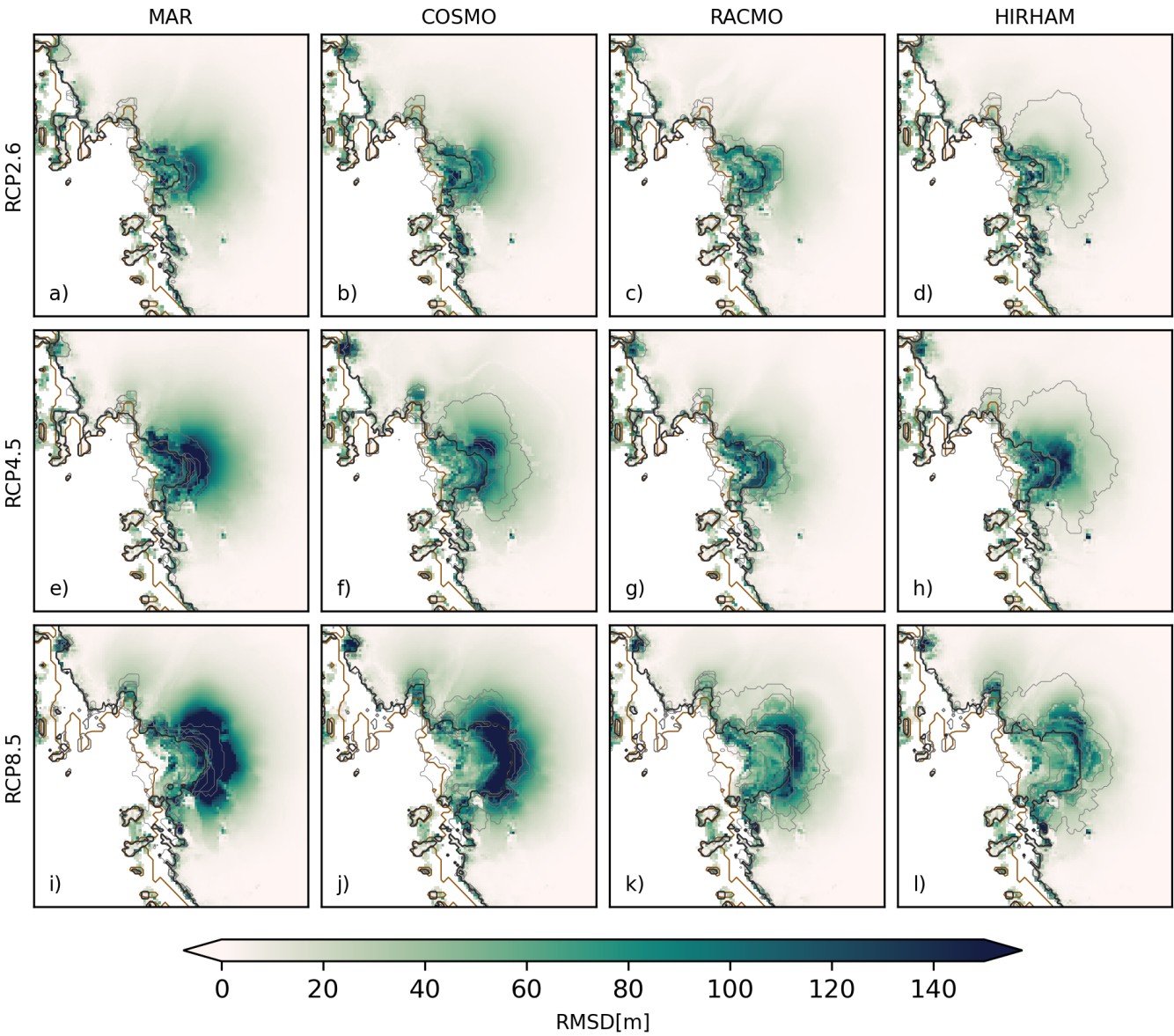

**Figure 10.** RMSD for RCP2.6 (a-d), RCP4.5 (e-h), and RCP8.5 (i-l) together with the position of the simulated median (black) and observed (brown) grounding line at the year 2300. The thin grey lines indicate the simulated grounding line position of the individual ensemble members (cmp. Tab. C1). Please be aware that the used color-scale in this section is narrower than in the previous sections.

### 3.2.4 Grounding line sensitivity to RCM baseline forcing

The regional impact (Amundsen Sea sector) of the applied forcings and parameter configuration are illustrated in Fig. 9 and 11. In detail, Fig. 11 illustrates the ensemble statistics of the grounding line position in the Amundsen Sea sector. Percentiles were drawn from the grounded ice mask density, which states, for every grid location (i,j), the percentage of ensemble members which have grounded ice at this location. In our PI-control simulations changes of grounding line positions with respect to the observed grounding line and the spread due to the RCM baseline forcings are rather small. However, a difference between simulations forced by COSMO is observable, especially for the most extreme case (95th percentile). In this case (Configuration No. 12), the simulation forced by COSMO shows collapse of WAIS until the year 2300, while there is almost no grounding line retreat in the simulations forced by the other models (cmp. Fig. C6). This is quite remarkable since under the same parameters, the simulation forced with COSMO shows minimal grounding line change for the RCP2.6 scenario (cmp. Fig. C7). While the grounding line position does not significantly change for the least extreme grounding line position (5th percentile) in any of our scenarios, we observe a increase in grounding line retreat within our scenarios for the 50th and 95th percentile of grounding line position. Differences in grounding line position due to the choice of RCM are rather small in the RCP2.6 and RCP4.5 median grounding lines. For the most retreated grounding line (95th percentile), there are large differences between the individual RCM forcings for all scenarios. Please be aware that we show the ensemble statistics of the grounding line extent, which not necessarily represent a specific parameter configuration. On the level of specific ensemble members, the difference between the individual RCMs tends to be higher (cmp. Fig. C7, C8, C9).

## 4 Discussion

The aim of this modelling study was to quantify and demonstrate the impact of different baseline SMB and temperature forcings on the evolution of the Antarctic Ice Sheet under a 30 000 years present-day equilibrium climate as well as projections using RCP scenarios extended to the year 2300. We now discuss the modelled future sea-level rise contributions and their uncertainties, ice sheet stability and equilibrium states under present-day forcing.

### 4.1 Uncertainty of Antarctic sea-level contributions due to the choice of RCM baseline forcing

Our simulations suggest differences in projected Antarctic sea-level contributions, due to the choice of present-day SMB and temperature baseline forcing, of $10.6\,(2.9-30.0)$ mm in 2100 and $70.0\,(17.3-365.5)$ mm in 2300 for RCP8.5 if we correct by the PI-control simulations. Those numbers are around a factor of 10 (for 2100) and 3 (for 2300) smaller than without the PI-control corrections. Since we often assume Antarctica has been stable during the Holocene, the base RCM forcing should not yield any changes under PI-conditions. This leads us to the conclusion that the interplay between the RCM and the GCM anomalies (PI-corrected $\Delta slr^{max}$) is the more accurate estimate of RCM induced uncertainties in centennial projections. In comparison to our results, the ISMIP6 project demonstrated that the choice of ISM and GCM forcing creates an uncertainty spread of $13.3\pm8.0$ cm SLE (projections ranging from $-3.7\pm3.4$ to $9.6\pm7.2$ cm SLE) for the end of the 21th century under

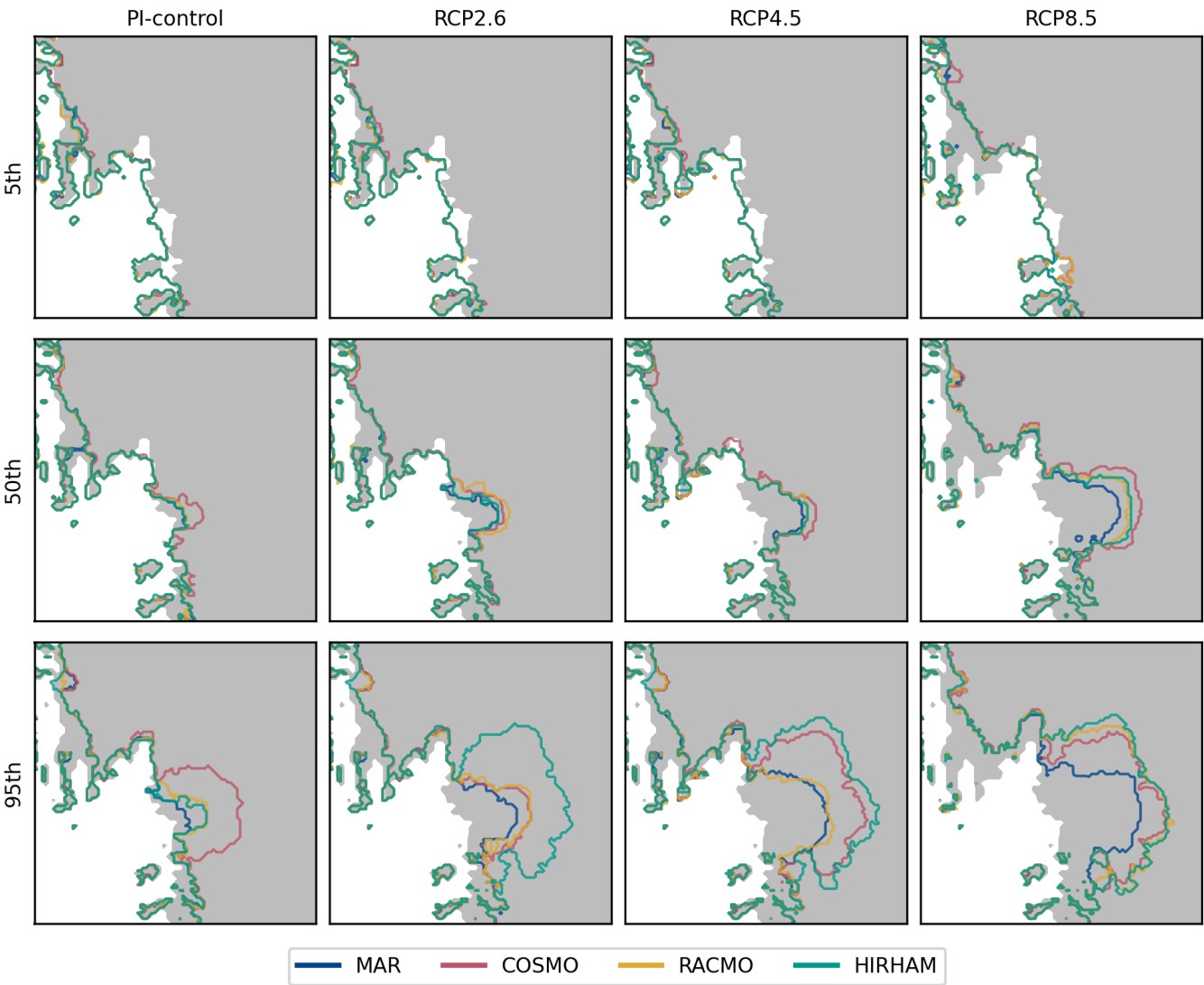

**Figure 11.** 5th, 50th and 95th percentile of grounding line position at the year 2300 for simulations using the four RCM forcing sets in the PI-control (a-c), RCP2.6 (d-f), RCP4.5 (g-i) and RCP8.5 (j-l) scenario at Thwaites and Pine Island Glacier. Percentiles were drawn from the grounded ice mask density, which states, for every grid location (i,j), the percentage of ensemble members which have grounded ice at this location. Grey shaded area indicate observed present-day grounded ice extend.

the RCP8.5 scenario (Seroussi et al., 2020). Our estimates of RCM uncertainties are over one order of magnitude smaller, and therefore probably only play a small role in projections until the year 2100. Nevertheless, the ISMIP6 ocean forcing induces a sea level rise projection spread by 2100 between 1.4 - 4.0 cm (Reese et al., 2020) with the same ISM used in this study. Additionally, we do not observe significant differences in $\Delta slr^{max}$ between different RCP scenarios in 2100. This is probably because differences between those RCP scenarios won't become apparent until after the 21st century, as noted by Lowry et al. (2021). For the year 2300, results for the extended ISMIP6 simulations are not publicly available yet. However based on our results, we expect the importance of the RCM uncertainty to increase, since the relation between PI-corrected and uncorrected $\Delta slr^{max}$ increases threefold from 2100 to 2300 (c.f. Fig. 8). The increase in $\Delta slr^{max}$ from the RCP2.6 to RCP8.5 scenario additionally suggests, that the RCM induced uncertainty might be larger for stronger warming scenarios. Nevertheless, projections by Bulthuis et al. (2019) show a parameter choice dependent spread in projected sea-level rise from 0.17 - 3.12m (5-95% confidence interval), compared to which our estimated impact of RCM choice seems to be rather small. Nevertheless, the parameter uncertainties by Bulthuis et al. (2019) may be considered an upper limit compared to other studies (Golledge et al., 2015; Ritz et al., 2015; DeConto and Pollard, 2016; Schlegel et al., 2018). It is important to note that the uncertainty presented here depends partly on the initialization method and the GCM forcing applied. In contrast to this study, in the IS-MIP6 protocol models were initialized with different reference present-day forcing fields. Furthermore, various types of model tuning to match present-day observations were applied e.g. nudging or inversion (Seroussi et al., 2020; Nowicki et al., 2020). Specifically, initialization techniques, such as basal friction inversion or nudging, have the capability to incorporate substantial portions of the reference forcing differences into the refined basal friction field. While this could reduce model deviations from the observed state of the ice sheet, it might concurrently give rise to larger differences in a changing climate. Notably, the WAIS is especially sensitive to these minor discrepancies in basal friction due to its overall high sensitivity particularly because of the Marine Ice Sheet Instability (MISI) (DeConto and Pollard, 2016).

## 4.2 Choice of RCM baseline affects the stability of the WAIS in future projections

The complex relationship between the selected RCM baseline climatology and its impact on future sea-level rise is closely related to the stability of WAIS, particularly the dynamic response of Thwaites and Pine Island Glaciers. This becomes particularly evident in the context of the RCP8.5 scenario. Depending on the choice of RCM baseline forcing differences in grounding line migration (c.f. Fig. 11) and corresponding ice thickness changes (c.f. Fig. 9) are simulated. This underscores the WAIS's sensitivity to the choice of present-day reference forcing data and underlines the importance of careful model parameterization and selection of forcing data. It is important to note that the reference forcing data does not influence whether the WAIS enters an grounding line retreat in our centennial projections, but rather modulates the rate of ice loss (c.f. C9). The initiation of unforced grounding line retreat seems to be predominantly dependent on the ocean thermal forcing. This is evident from the fast ice loss observed soon after the beginning of the RCP8.5 scenario, in stark contrast to the control runs which mostly exhibit minimal changes (c.f. Fig. 6).

## 4.3 Millennial-Scale response predisposed by choice of RCM reference forcing

To demonstrate the influence of reference SMB and temperature fields on the long-term evolution of WAIS, we presented individual simulation results from our ensemble, as shown in Fig. 5. Our simulations indicate that differences in reference SMB and temperature forcing can lead to not only a slow, gradual response of the ice sheet thickness, but also centennial scale, non-linear responses. When constant COSMO and MAR forcing is applied for identical ice sheet model parameters, a collapse of the WAIS is observed (c.f. Fig. 5), while exactly the same parameters yield a stable WAIS under RACMO and HIRHAM forcing. In those simulations forced by MAR and COSMO individual grid boxes unground as shown in Fig. B5, which leads to accelerated ice flow. Notably, the long term evolution of the ice sheets might also be affected by the thermal spinup. In this study, we only performed a thermal spinup using the constant PD temperature fields. Therefore, we might lack the historical temperature imprint of the last glacial cycles in the ice sheet which might affect dynamics especially when the configuration is close to an instability. Additionally, several of our simulations do not seem to have reached a (quasi-) steady state after 30kyrs. We therefore can not exclude a potential WAIS collapse at a later stage (i.e. after the initial 30kyrs of our simulations). This would imply that the committed ice sheet response is mainly driven by the parameter set itself, while the RCM climatology might modulate the timing of the collapse. However, this hypothesis would require longer simulations which are beyond the scope of this study.

## 4.4 SMB anomalies are imprinted in ice thickness equilibrium

The simulated change in sea level-relevant ice mass in the present-day equilibrium simulations demonstrates the expected behavior, where simulations with the highest SMB forcing (e.g. MAR) lead to the largest ice volume (c.f. Fig. 3). The observed spatial distributions of $\Delta h$ roughly agree with the anomalies observed in the SMB forcing. Regional scale structures in $\Delta h$ often differ from the underlying SMB anomalies, perhaps unsurprisingly given the inherent non-linearities of WAIS dynamics. Here we discuss the relationship between regional SMB forcing and quasi-equilibrium state of the ice sheet for individual catchment areas (i.e. IMBIE basins; Rignot et al. (2011)). Averaged over those basins (c.f. Fig. A2, B2), the ice sheet responds inline with the SMB forcing, with only a few exceptions. One exception is the WAIS in simulations forced with the SMB and temperature fields from RACMO. While we mostly observe small positive SMB anomalies with respect to the RCM mean for all WAIS basins except the Ross basin, the thickness anomalies ($\Delta h$) are all negative for those basins. The reason for this might be a shift in the ice divide, which would result in an outflux of ice towards the Ross drainage basin. However, a more in-depth analysis would be necessary to assess if this is the main driver of the observed behavior. In addition, one has to note that our study did not account for any ice sheet feedbacks of the ice thickness change on surface temperature and mass balance, which might lead to a different result (Coulon et al., 2024; Li et al., 2023).

## 4.5 Parameter sensitivity of the RCM impact

The ice sheet-wide parameter sensitivity of $\Delta h$ is larger for the present-day equilibrium simulations than in the centennial projections. This can be attributed to the long response time of the AIS, especially in vast areas of the East Antarctic Ice

Sheet (EAIS), and the much smaller simulation time in the future projections. However, for Thwaites and Pine Island Glacier, the parameter sensitivity of $\Delta h$ is higher in the centennial projections. This can be explained by the fact that the present-day forcing exposes the Antarctic Ice Sheet (AIS) to weaker climate drivers (e.g. ocean induced melt) than all of the RCP scenarios.

Therefore, we would expect the parameter sensitivity of $\Delta h$ to be more similar to the RCP2.6 than the RCP8.5 scenario. Generally, a high parameter sensitivity of $\Delta h$ highlights regions where the interplay between SMB and chosen parameters is highest. Therefore, the model representation of these regions would especially benefit from a better constraint surface mass balance forcing.

## 5    Conclusion

Regional climate model SMB and temperature data are a standard resource for studies employing large scale ISMs. In this study, we investigated the influence of a set of different RCM products on the evolution and dynamics of the AIS in 30 000 year constant forcing equilibrium simulations as well as in several future projections employing RCP2.6, RCP4.5, and RCP8.5 using a parameter ensemble reflecting various ice sheet sensitivities. Our results demonstrated that although all surface mass balance and temperature products are externally driven by the same reanalysis and simulate the same fields, the impact of their

differences on both ice thickness and grounding line dynamics is considerable. For the long term "quasi-equilibrium" state after 30 ka of simulation, we showed that ice thickness anomalies averaged over the individual IMBIE drainage basins mostly reflect SMB differences between RCMs. However, the differences in SMB forcing can lead to non-linear ice sheet responses on regional scales. For the centennial term projections, our findings indicate that differences mostly arise in, and are limited to, the vicinity of the grounding line. Our simulations further show that the model-uncertainty for sea level rise projections from

the difference in reference present-day forcing is around 10% of the uncertainty arising from the choice of ISM and GCM forcing based on ISMIP6 results. However, the RCM uncertainty seems to increase for longer projections and higher emission scenarios. Additionally, our simulations depict differences in the pacing and timing of grounding line retreat and ice thickness for Thwaites and Pine Island Glacier under all emission scenarios. Our sensitivity analysis, indicates that the imprint of the SMB forcing on the ice thickness is especially dependent on the chosen parameters for the ISM (model sensitivity). Our long

term equilibrium study additionally indicates that the difference in the SMB products is potentially large enough to result in long term instabilities of the WAIS in one forcing set, while long term WAIS stability can be observed in another. Our study displays the large impact of the choice of a reference present-day forcing onto projections of ice sheet evolution, however this sensitivity is model dependent and should be explored on a case-by-case basis. Prospectively, a more rigorous approach employing a wider sweep of parameters, more sophisticated ice sheet initialization methods (less model drift) and a larger set

of GCM and RCM climate projections, would be desirable. Yet, here we demonstrate the problem that occurs due to the spread in RCM products, the order of magnitude of this uncertainty and potential implications on the stability of the ice sheet.

## Appendix A: Annual SMB in the different drainage areas

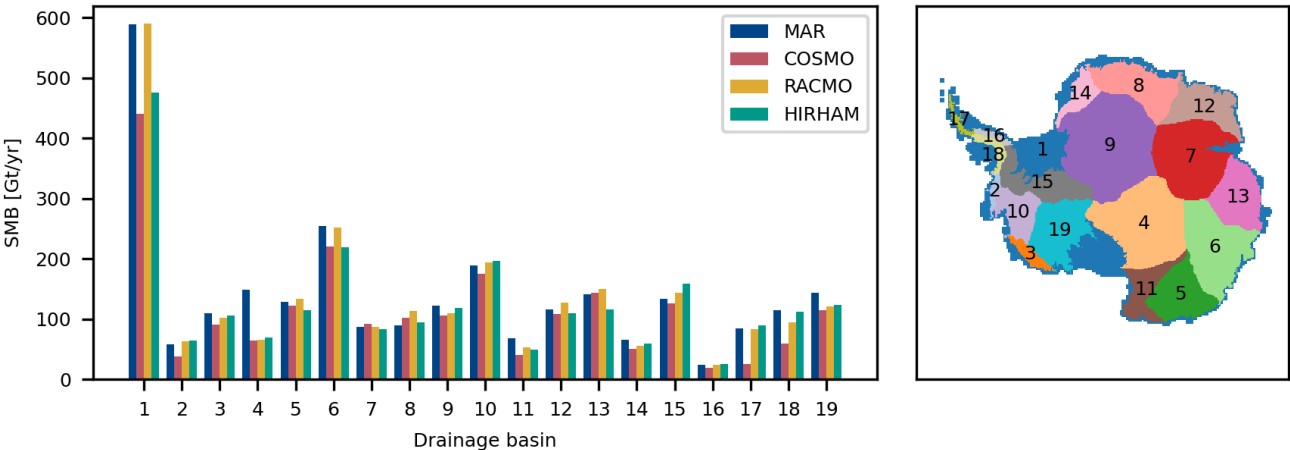

**Figure A1.** Annual SMB for the 19 IMBIE drainage areas. Corresponding areas are marked on the map. Please be aware that area 1 includes all shelf ice areas.

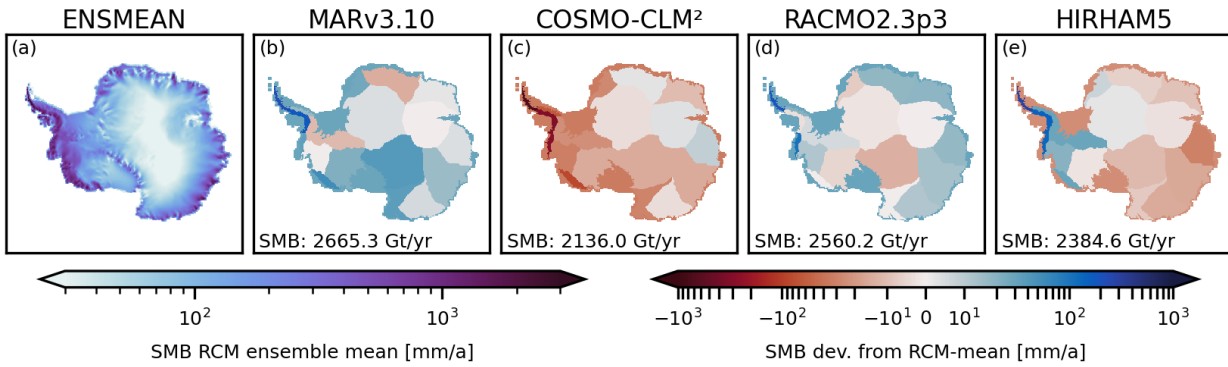

**Figure A2.** Surface mass balance (SMB) of the (a) multi-RCM mean and anomalies of the (b) MARv3.10, (c) COSMO-CLM2, (d) RACMO2.3p3, and (e) HIRHAM5 regional climate model from this mean averaged over the individual IMBIE basins.

| Parameter configuration | $sia_e$ | $pQ$ | $\gamma$ [$\times 10^{-5}$] | $\phi_{till\,min}$ |
| --- | --- | --- | --- | --- |
| 1 | 1.0 | 0.75 | 2.0 | 2 |
| 2 | 1.0 | 0.75 | 2.5 | 2 |
| 3 | 1.0 | 0.75 | 3.0 | 2 |
| 4 | 1.0 | 0.80 | 3.0 | 2 |
| 5 | 1.0 | 0.80 | 3.5 | 2 |
| 6 | 1.0 | 0.80 | 4.0 | 2 |
| 7 | 1.25 | 0.75 | 2.5 | 2 |
| 8 | 1.25 | 0.80 | 2.0 | 2 |
| 9 | 1.25 | 0.80 | 2.5 | 2 |
| 10 | 1.25 | 0.80 | 3.0 | 2 |
| 11 | 1.25 | 0.80 | 4.0 | 2 |
| 12 | 1.5 | 0.75 | 2.0 | 2 |
| 13 | 1.5 | 0.75 | 2.5 | 2 |
| 14 | 1.5 | 0.80 | 2.0 | 2 |
| 15 | 1.5 | 0.80 | 2.5 | 2 |
| 16 | 1.5 | 0.80 | 3.5 | 2 |
| 17 | 1.25 | 0.75 | 2.0 | 4 |
| 18 | 1.25 | 0.75 | 2.5 | 4 |
| 19 | 1.25 | 0.80 | 2.0 | 4 |
| 20 | 1.25 | 0.80 | 2.5 | 4 |
| 21 | 1.5 | 0.80 | 2.0 | 4 |
| 22 | 1.5 | 0.80 | 2.5 | 4 |
| 23 | 1.5 | 0.75 | 2.0 | 6 |
| 24 | 1.5 | 0.80 | 2.5 | 6 |

**Table B1.** Chosen parameter space for the shallow ice approximation enhancement factor $sia_e$, the pseudoplastic $pQ$ factor, the heat conductivity at the ice ocean interface $\gamma$, and the minimum till friction angle $\phi_{till\,min}$ for the present-day equilibrium runs.

| Experiment name | Climate Forcing | Init. cond. | Parameters | Length | Description |
| --- | --- | --- | --- | --- | --- |
| MAR_PD_thermal | MARv3.10 | BedMachine | B1 | 200 kyrs | TS |
| COSMO_PD_thermal | COSMO-CLM2 | BedMachine | B1 | 200 kyrs | TS |
| RACMO_PD_thermal | RACMO2.3p3 | BedMachine | B1 | 200 kyrs | TS |
| HIRHAM_PD_thermal | HIRHAM5 | BedMachine | B1 | 200 kyrs | TS |
| MAR_PD | MARv3.10 | MAR_PD_thermal | B1 | 30 kyrs | FE |
| COSMO_PD | COSMO-CLM2 | COSMO_PD_thermal | B1 | 30 kyrs | FE |
| RACMO_PD | RACMO2.3p3 | RACMO_PD_thermal | B1 | 30 kyrs | FE |
| HIRHAM_PD | HIRHAM5 | HIRHAM_PD_thermal | B1 | 30 kyrs | FE |
| PI_thermal | RCM mean + HadGem2-ES PI-PD ano. | BedMachine | C1 | 200 kyrs | TS |
| PI_relax | RCM mean + HadGem2-ES PI-PD ano. | PI_thermal | C1 | 300 yrs | FE |
| MAR_PIcontrol | MARv3.10 + HadGem2-ES PI-PD ano. | PI_relax | C1 | 440 yrs | FE |
| COSMO_PIcontrol | COSMO-CLM2 + HadGem2-ES PI-PD ano. | PI_relax | C1 | 440 yrs | FE |
| RACMO_PIcontrol | RACMO2.3p3 + HadGem2-ES PI-PD ano. | PI_relax | C1 | 440 yrs | FE |
| HIRHAM_PIcontrol | HIRHAM5 + HadGem2-ES PI-PD ano. | PI_relax | C1 | 440 yrs | FE |
| MAR_hist | MARv3.10 + HadGem2-ES hist. ano. | PI_relax | C1 | 145 yrs | FE |
| COSMO_hist | COSMO-CLM2 + HadGem2-ES hist. ano. | PI_relax | C1 | 145 yrs | FE |
| RACMO_hist | RACMO2.3p3 + HadGem2-ES hist. ano. | PI_relax | C1 | 145 yrs | FE |
| HIRHAM_hist | HIRHAM5 + HadGem2-ES hist. ano. | PI_relax | C1 | 145 yrs | FE |
| MAR_RCP2.6 | MARv3.10 + HadGem2-ES RCP2.6 ano. | MAR_hist | C1 | 295 yrs | FE |
| COSMO_RCP2.6 | COSMO-CLM2 + HadGem2-ES RCP2.6 ano. | COSMO_hist | C1 | 295 yrs | FE |
| RACMO_RCP2.6 | RACMO2.3p3 + HadGem2-ES RCP2.6 ano. | RACMO_hist | C1 | 295 yrs | FE |
| HIRHAM_RCP2.6 | HIRHAM5 + HadGem2-ES RCP2.6 ano. | HIRHAM_hist | C1 | 295 yrs | FE |
| MAR_RCP4.5 | MARv3.10 + HadGem2-ES ano. | MAR_hist | C1 | 295 yrs | FE |
| COSMO_RCP4.5 | COSMO-CLM2 + HadGem2-ES RCP4.5 ano. | COSMO_hist | C1 | 295 yrs | FE |
| RACMO_RCP4.5 | RACMO2.3p3 + HadGem2-ES RCP4.5 ano. | RACMO_hist | C1 | 295 yrs | FE |
| HIRHAM_RCP4.5 | HIRHAM5 + HadGem2-ES RCP4.5 ano. | HIRHAM_hist | C1 | 295 yrs | FE |
| MAR_RCP8.5 | MARv3.10 + HadGem2-ES RCP8.5 ano. | MAR_hist | C1 | 295 yrs | FE |
| COSMO_RCP8.5 | COSMO-CLM2 + HadGem2-ES RCP8.5 ano. | COSMO_hist | C1 | 295 yrs | FE |
| RACMO_RCP8.5 | RACMO2.3p3 + HadGem2-ES RCP8.5 ano. | RACMO_hist | C1 | 295 yrs | FE |
| HIRHAM_RCP8.5 | HIRHAM5 + HadGem2-ES RCP8.5 ano. | HIRHAM_hist | C1 | 295 yrs | FE |

**Table B2.** Simulation setups for present-day equilibrium as well as future projections. B1 and D1 refer to the parameter configurations stated in Table B1 and C1. The description indicates if the run is a thermal spinup (TS) under which the ice geometry is keep constant, or the ice can freely evolve (FE). For the historical simulations, HadGem2-ES anomalies to present-day from 1860 to 2005 are applied. For the RCP scenarios HadGem2-ES anomalies to present-day from 2005 to 2300 are applied.

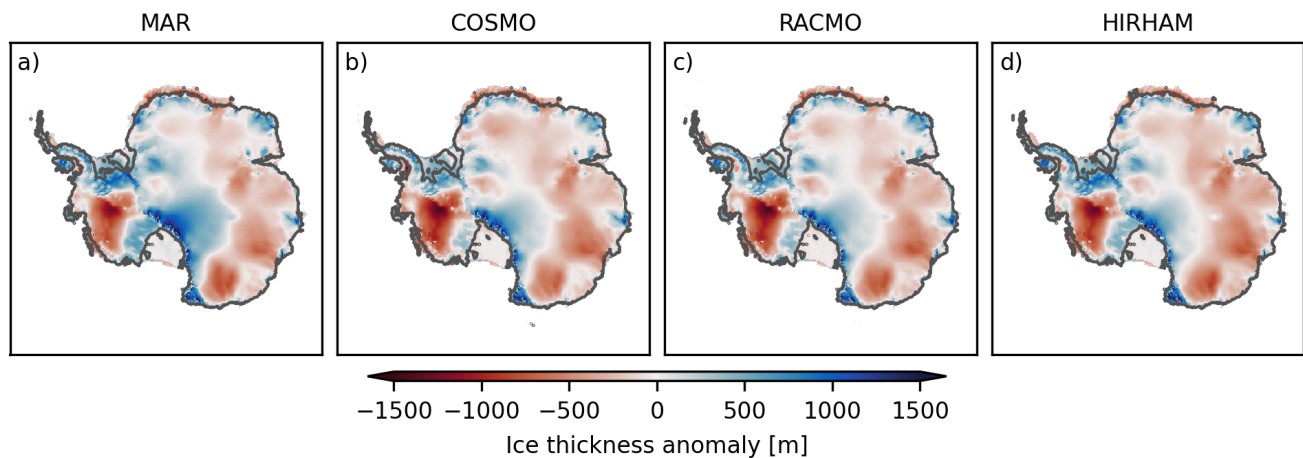

**Figure B1.** Median ice thickness anomalies towards present-day observations after 30 000 years under the given RCM SMB.

## Appendix B: Ice thickness difference to PD observations for PD-equilibrium

**Figure B2.** Ice thickness anomalies from common mean (a-d), position of the simulated (purple) and observed (grey) grounding line for the present-day equilibrium simulations. Root mean square error (RMSD) of the individual ensemble members from the median (e-h). All values averaged over the individual IMBIE drainage basins.

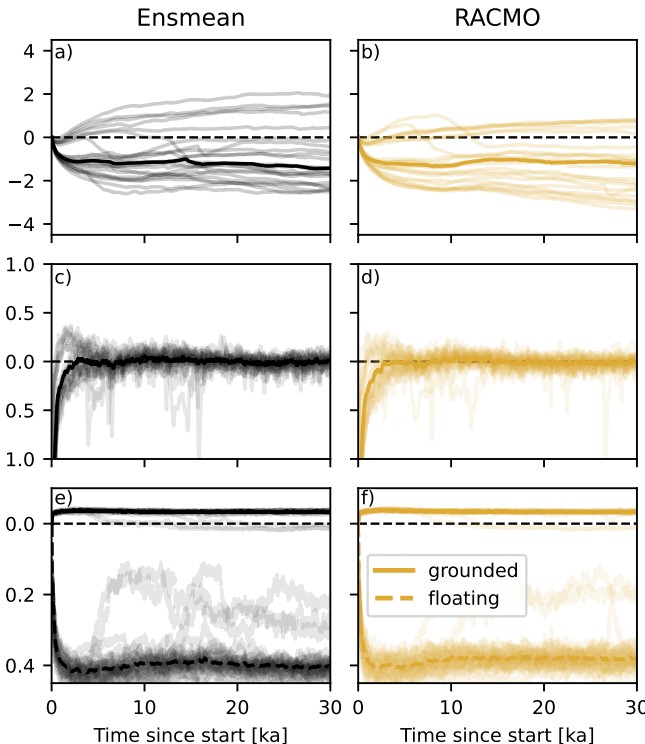

**Figure B3.** Time series of the total ice mass change (a,b), the annual rate of change (c,d), and the fraction of grounded (solid line) and floating (dashed line) ice area (e,f) relative to present-day observations for the simulations forced by the mean of all four RCM products (referred to as ensmean) and simulations forced by RACMO. Bold line shows ensemble median while shaded lines indicate the individual ensemble members.

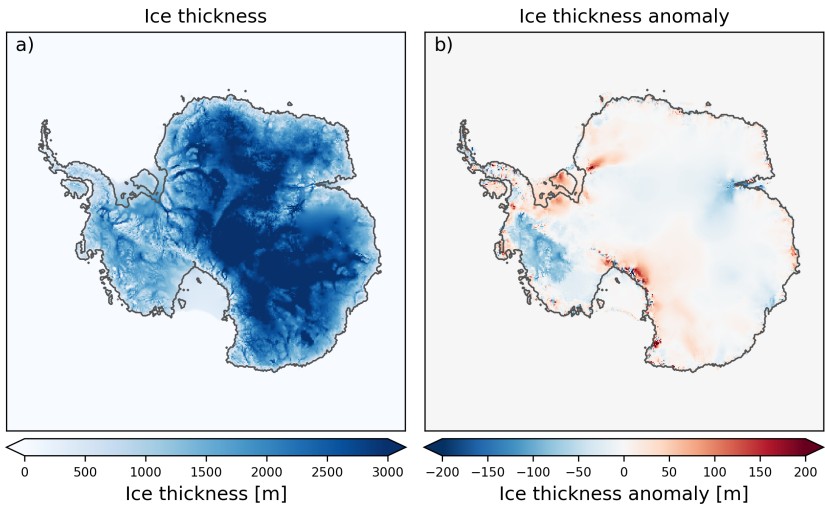

**Figure B4.** Mean ice thickness after 30000 years for a simulation forced with the mean of all four RCMs (a) and the mean ice thickness anomalies between a) and the mean of the individual forcing simulations (b).

# Appendix C: Future projections

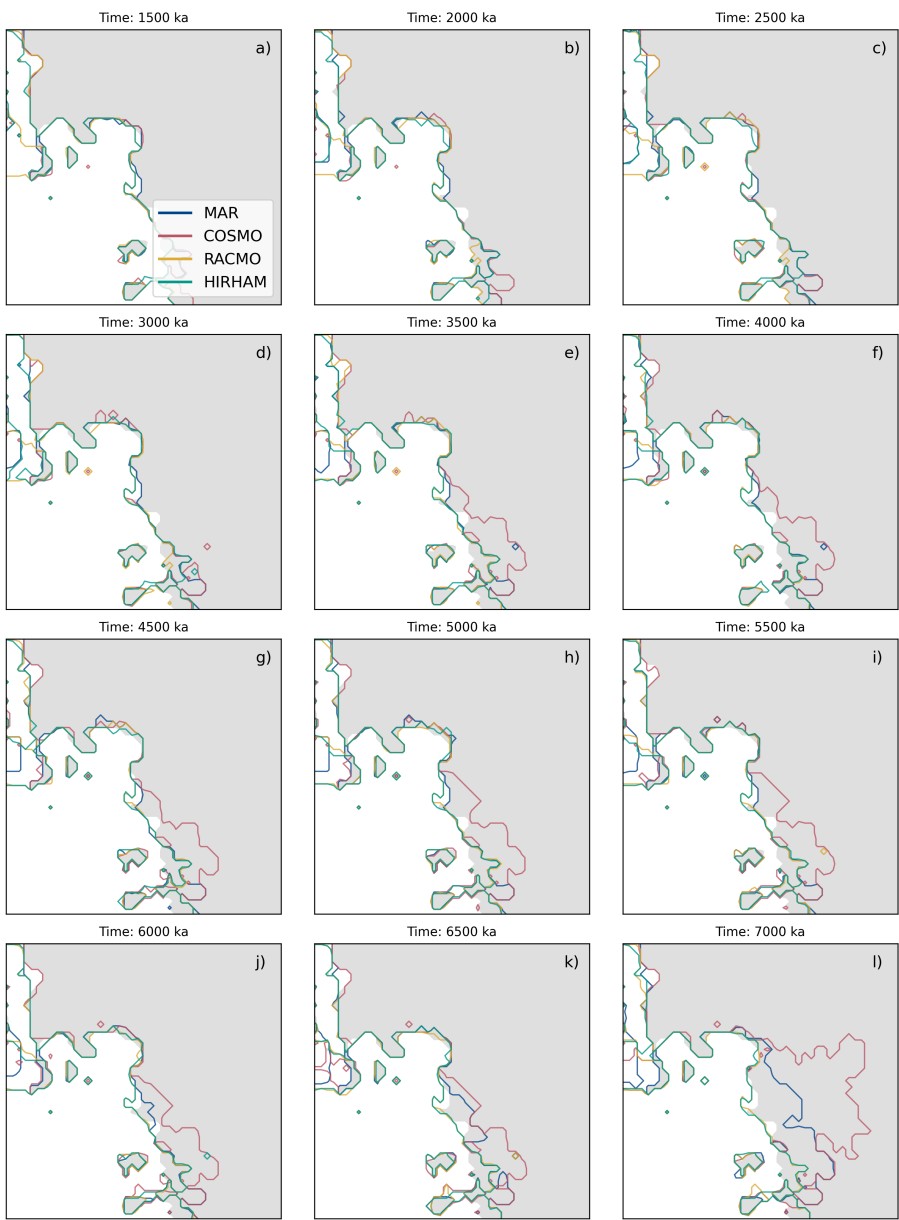

**Figure B5.** Evolution of the grounding line at thwaites glacier for PD parameter configuration No. 6 in time steps of 500 years. The grey shaded patches represent the present-day grounding line position. The plot indicates that an early grounding line retreat of the simulation forced by the COSMO and later by the MAR model leads to a accelerating retreat of the grounding line, causing the later collapse of WAIS in this simulation as seen in Fig. 5.

| Parameter configuration | $sia_e$ | $pQ$ | $\gamma\,[\times 10^{-5}]$ | $\phi_{till\,min}$ |
|---|---|---|---|---|
| 1 | 1.0 | 0.80 | 2.0 | 4 |
| 2 | 1.0 | 0.85 | 2.5 | 4 |
| 3 | 1.0 | 0.75 | 3.0 | 6 |
| 4 | 1.0 | 0.85 | 3.0 | 6 |
| 5 | 1.25 | 0.75 | 3.5 | 6 |
| 6 | 1.25 | 0.75 | 4.0 | 6 |
| 7 | 1.25 | 0.80 | 2.5 | 6 |
| 8 | 1.25 | 0.80 | 2.0 | 6 |
| 9 | 1.25 | 0.80 | 2.5 | 6 |
| 10 | 1.25 | 0.85 | 3.0 | 6 |
| 11 | 1.25 | 0.85 | 4.0 | 6 |
| 12 | 1.2 | 0.85 | 2.0 | 8 |

**Table C1.** Chosen parameter space for the shallow ice approximation enhancement factor $sia_e$, the pseudoplastic $pQ$ factor, the heat conductivity at the ice ocean interface $\gamma$, and the minimum till friction angle $\phi_{till\,min}$ for the centennial projections.

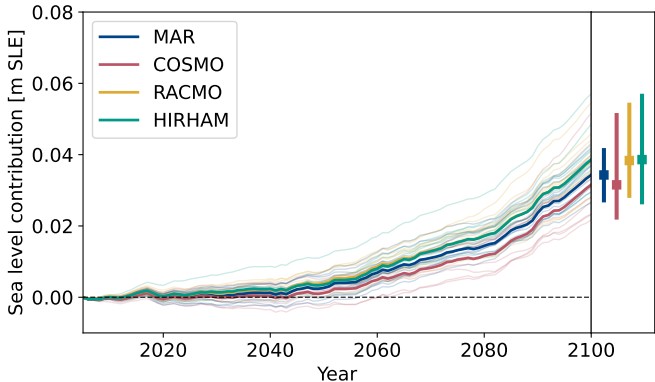

**Figure C1.** Antactic sea level rise contribution for the RCP8.5 scenario with different RCM present-day fields until the year 2100. Thin lines show individual simulations, bold lines the mean state for the different models.

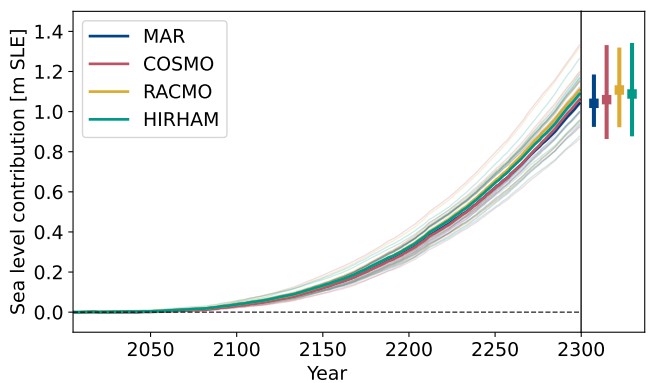

**Figure C2.** Antactic sea level rise contribution for the RCP8.5 scenario with different RCM present-day fields until the year 2300. Thin lines show individual simulations, bold lines the mean state for the different models.

| Model | Mean ice mass change (Gt) | Max ice mass change (Gt) | Min ice mass change (Gt) |
|---|---|---|---|
| MAR | 50939 | 72755 | 33015 |
| COSMO | -46946 | -35208 | -64889 |
| RACMO | -1098 | 26415 | -22334 |
| HIRHAM | -8536 | 14299 | -24527 |

**Table C2.** Ensemble mean, maximum and minimum, total ice mass change from 2015 until 2100 in the PI-control simulations.

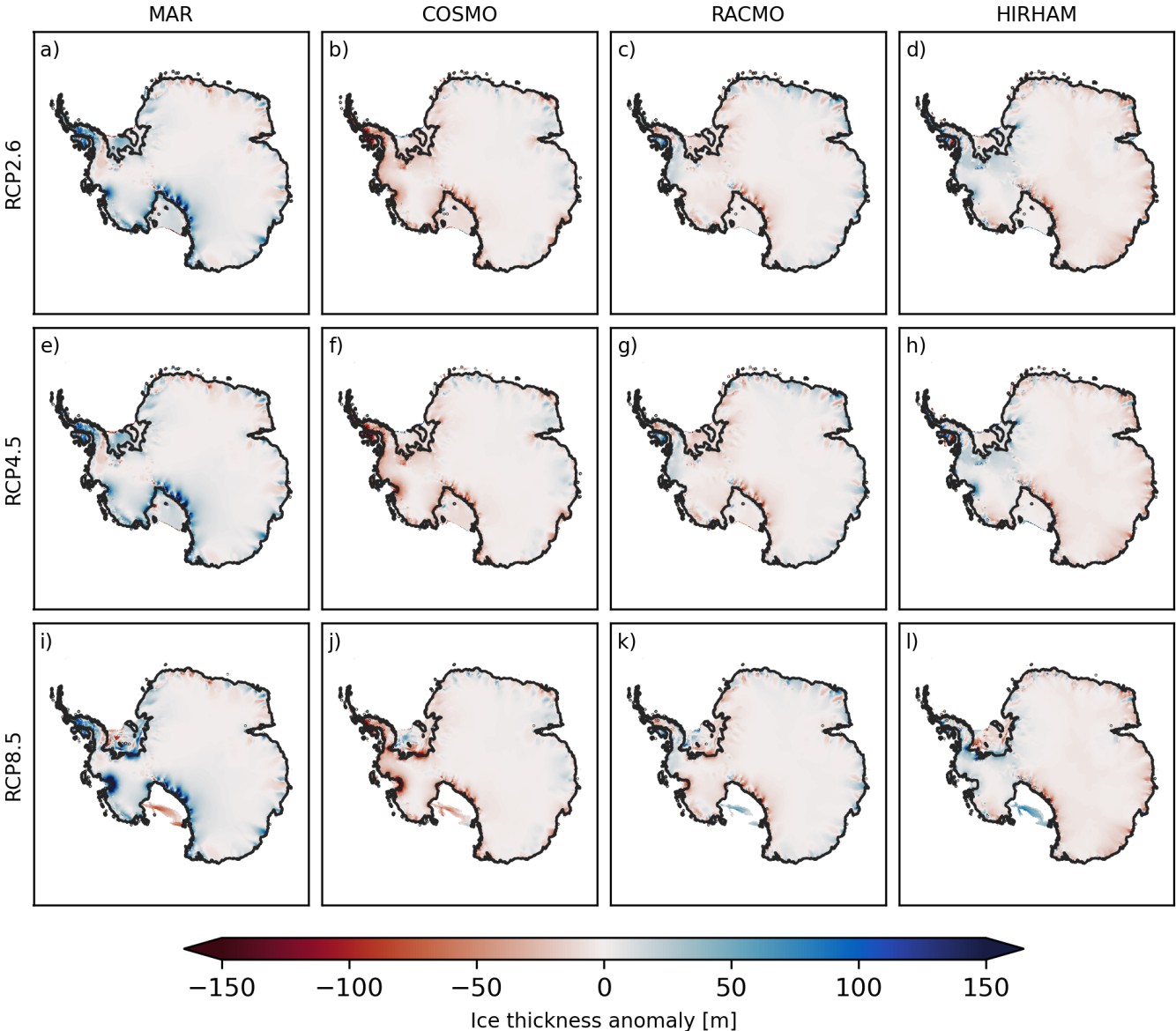

**Figure C3.** Median ice thickness anomalies from common mean for RCP2.6 (a-d), RCP4.5 (e-h), and RCP8.5 (i-l) together with the position of the simulated median (black) and observed (brown) grounding line at the year 2300. The thin grey lines indicate the simulated grounding line position of the individual ensemble members. Please be aware that the used color-scale in this section is smaller than in the previous sections.

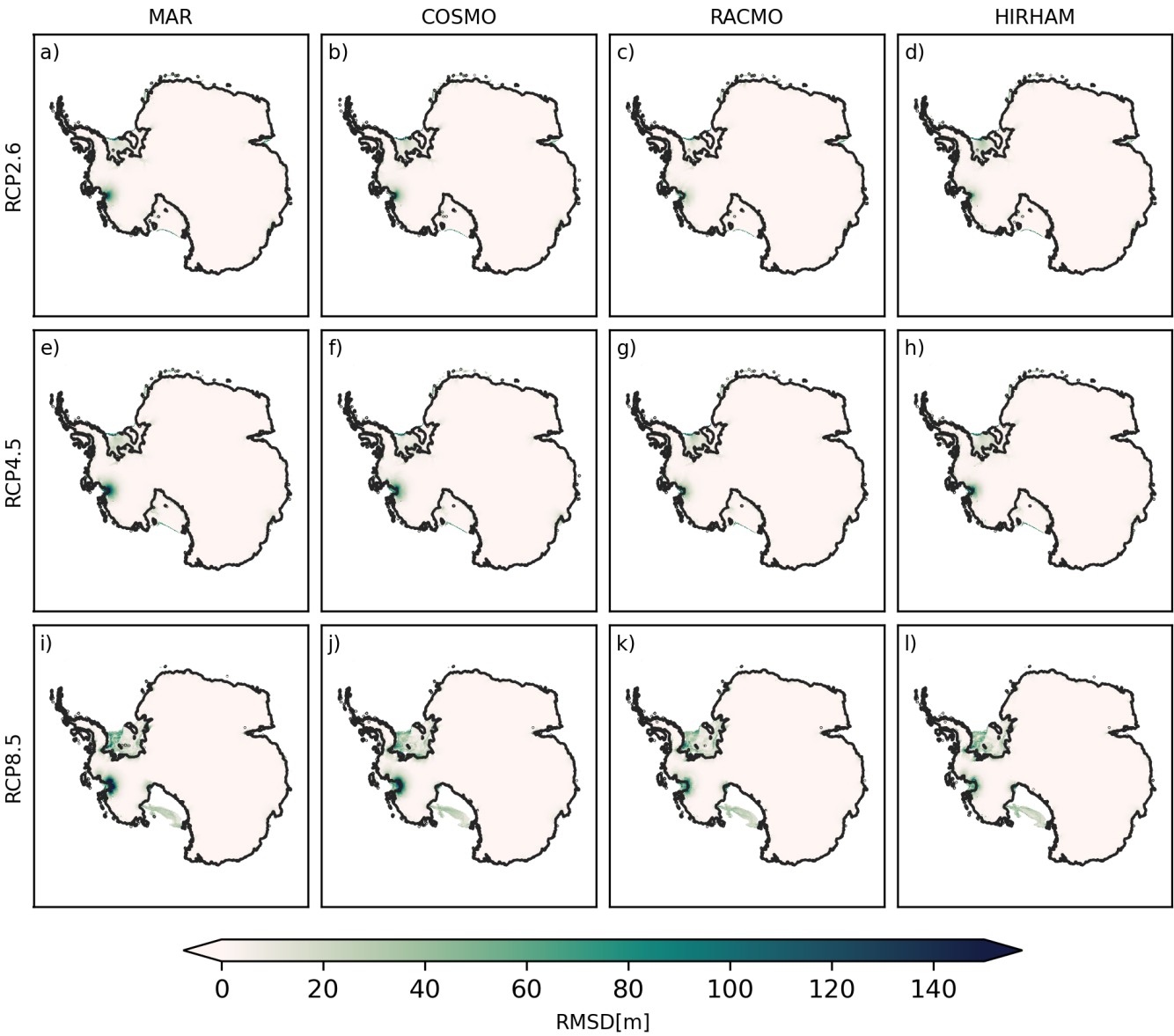

**Figure C4.** RMSD for RCP2.6 (a-d), RCP4.5 (e-h), and RCP8.5 (i-l) together with the position of the simulated median (black) and observed (brown) grounding line at the year 2300. The thin grey lines indicate the simulated grounding line position of the individual ensemble members. Please be aware that the used color-scale in this section is smaller than in the previous sections.

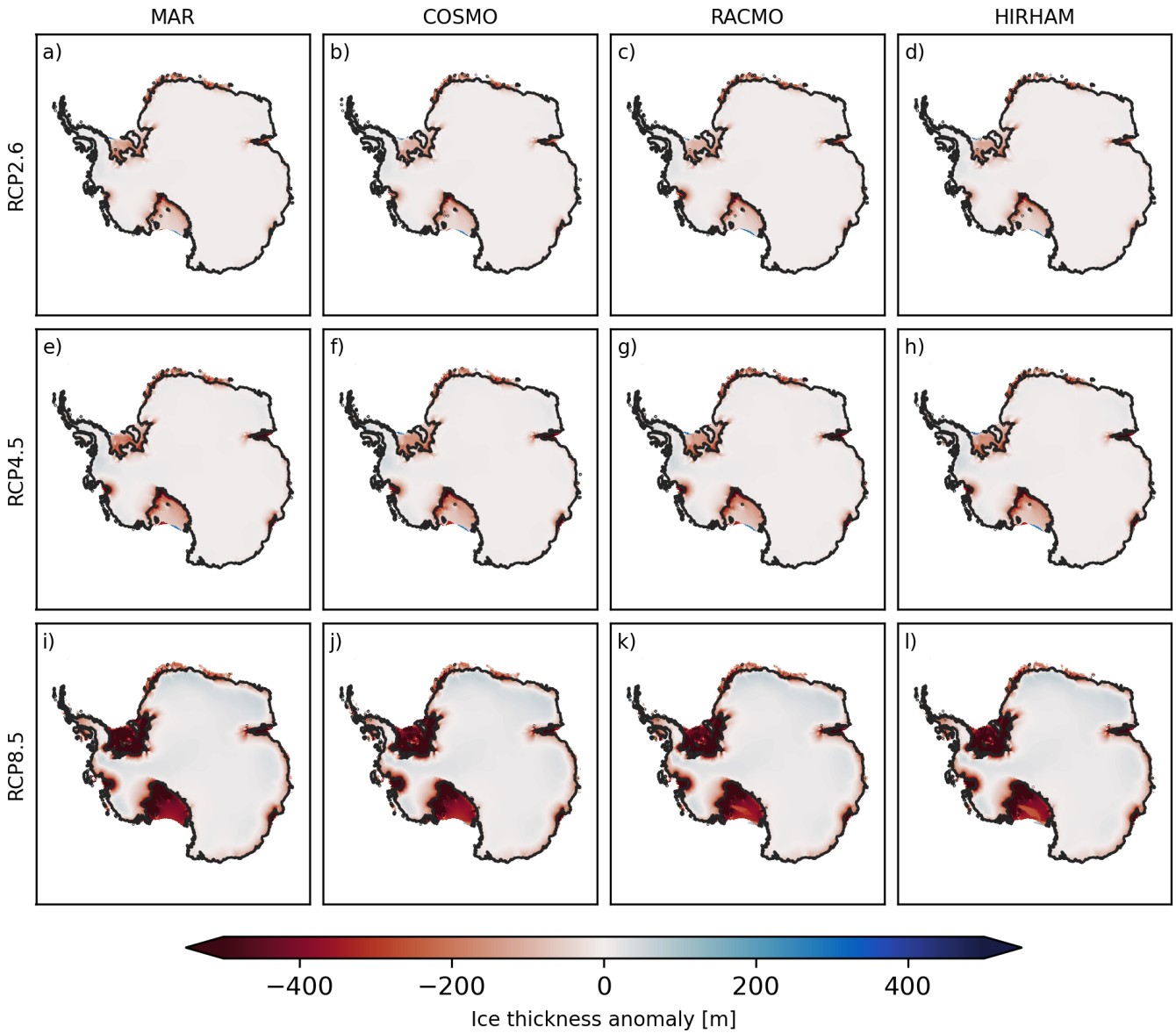

**Figure C5.** Median ice thickness anomalies with respect to the PI-control simulations for RCP2.6 (a-d), RCP4.5 (e-h), and RCP8.5 (i-l) together with the position of the simulated median (black) grounding line at the year 2300.

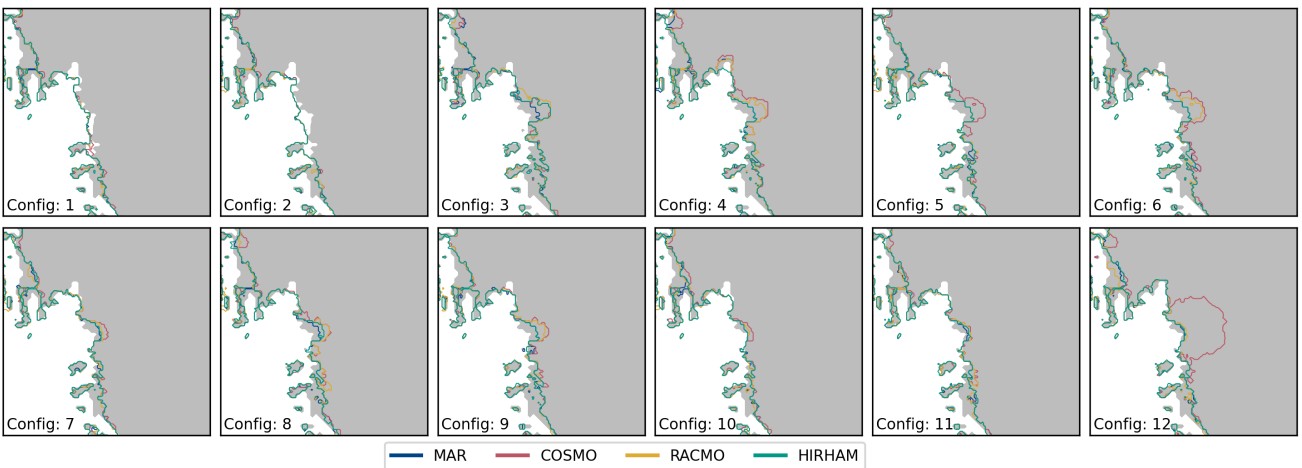

**Figure C6.** Individual grounding line extend for each individual RCM forcing and parameter configuration (c.f. Tab. C1) for the PI-control scenario at the year 2300.

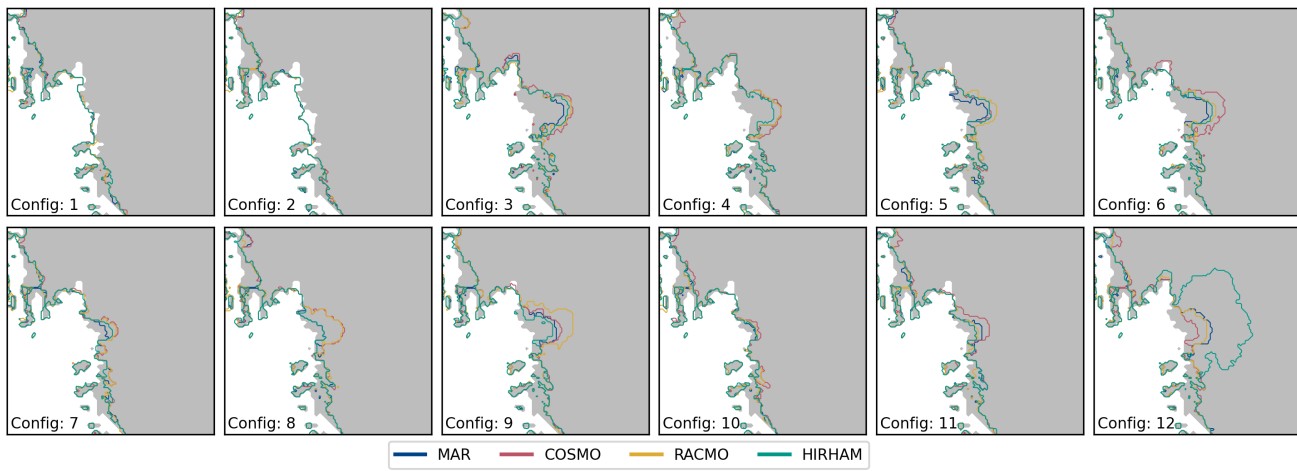

**Figure C7.** Individual grounding line extend for each individual RCM forcing and parameter configuration (c.f. Tab. C1) for the RCP2.6 scenario at the year 2300.

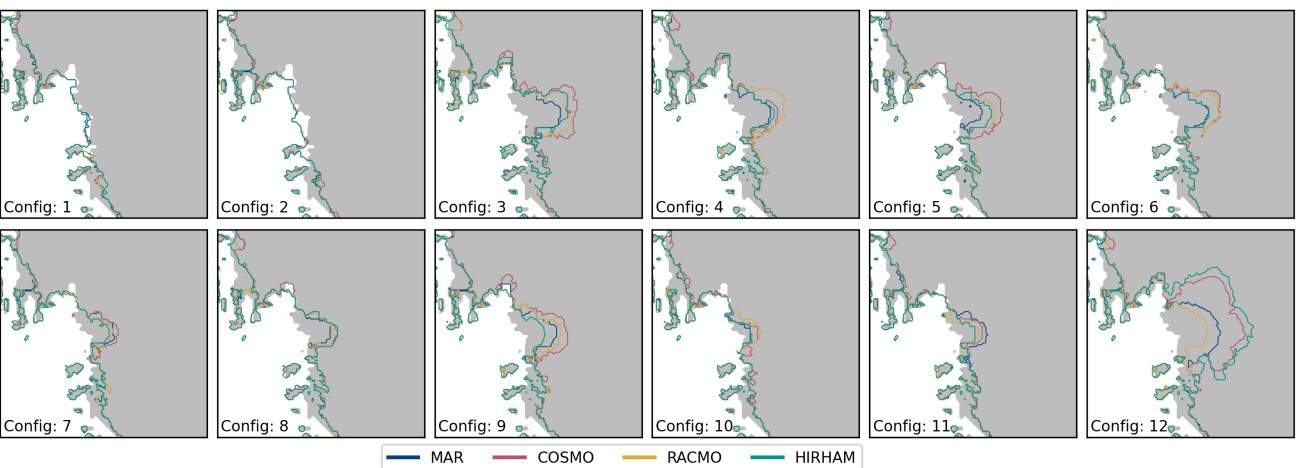

**Figure C8.** Individual grounding line extend for each individual RCM forcing and parameter configuration (c.f. Tab. C1) for the RCP4.5 scenario at the year 2300.

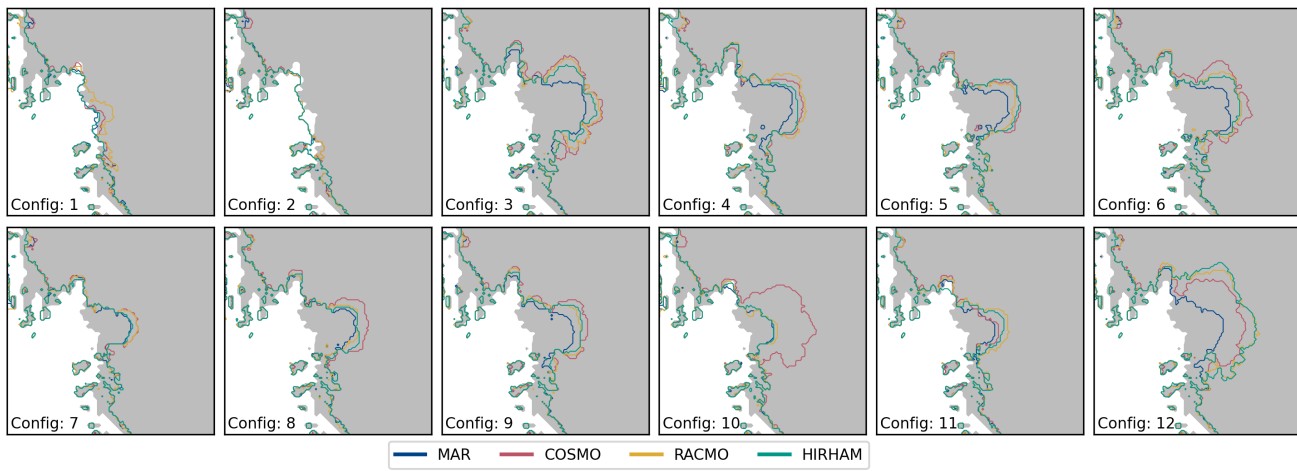

**Figure C9.** Individual grounding line extend for each individual RCM forcing and parameter configuration (c.f. Tab. C1) for the RCP8.5 scenario at the year 2300.

*Code and data availability.* The PISM model code is publicly available under www.pism.io and for this study version 1.2.2 was used. The applied RCM, GCM forcing fields are publicly available with the corresponding publication. Simulation results as well as code used in processing the data and illustrating the figures are available upon request.

*Author contributions.* CW, JS, and TS conceptualized this study, decided on the methodology, analyzed the data, edited and wrote this manuscript. CW lead the writing of the manuscript, ran the simulations and visualized the data under supervision of JS and TS

*Competing interests.* The authors declare that JS serves as editor for The Cryosphere.

*Acknowledgements.* Calculations were performed on UBELIX, the high-performance computing cluster at the University of Bern. We are grateful to Ruth Mottram and Nicolaj Hansen for discussion about the HIRHAM data. CW acknowledges funding by the Swiss National Science Foundation through the pleistoCEP2 project (grant no. 200492). J.S. acknowledges funding from the Deutsche Forschungsgemeinschaft under grant no. SU 1166/1-1 and from the Swiss National Science Foundation (grant no. 211542). TS acknowledges funding from the Swiss National Science Foundation (grant no. 200492) as well as funding from the Swiss National Science Foundation (grant no. 211542). TS and JS acknowledge funding from the European Union's Horizon 2020 research and innovation program under grant agreement no. 820970 (project TiPES). We thank the MAR team which make available the model outputs, as well agencies (F.R.S - FNRS, CÉCI, and the Walloon Region) that provided computational resources for MAR simulations. We thank Christoph Kittel and two anonymous referees for reviewing this manuscript.

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
