# Peer review of "The influence of present-day regional surface mass balance uncertainties on the future evolution of the Antarctic Ice Sheet"

_EGUsphere, 2023_

## Referee Comment (RC3)

This paper explores the projections of sea-level rise (SLR) from the Antarctic Ice Sheet using the Parallel Ice Sheet Model (PISM), driven by Surface Mass Balance (SMB) forcing derived from four distinct Regional Climate Models (RCMs). Specifically, the study assesses the impact of these RCMs on SLR projections under the global Climate Model HadGEM2-ES. The research reveals that the choice of RCM reference forcing introduces uncertainties in future sea-level rise predictions, comparable to influential factors like ice sheet model parameterization and global climate model choices. Notably, the study emphasizes that the selection of the RCM can influence the timing of the West Antarctic Ice Sheet (WAIS) grounding line retreat under RCP8.5. A parallel investigation examines the present-day forcing from ERA 5 on the 30ka long-term stability for the four different RCMs.

While the paper holds promise for publication, there is room for improvement in synthesizing the results, particularly regarding the equilibrium experiments. Further clarification is sought for the 2100 and 2300 experiments, with a specific focus on the rationale behind the SLR projection calculations and whether the numbers are subtracted by control runs.

Equilibriums runs:

While the same parameters tuned to RACMO yield different results for the other RCMs, I understand that it might be computationally prohibitive to conduct a spin-up for every RCM and parameterization. However, my concern lies in whether the obtained results convey physical insights. Typically, a glacial spin-up is undertaken to mitigate model shock, ensuring that projections are grounded in physical processes rather than numerical artifacts. Given this, I find it surprising that RACMO still exhibits considerable model shock.

Could you clarify whether there was a change in resolution from the glacial spin-up to the equilibrium run? If not, kindly include the 16km resolution in your experimental design details. Additionally, I am curious about the parameters utilized for the glacial spin-up.

I am grappling with the interpretation of the results, uncertain about their physical significance versus numerical artifacts. It would be immensely helpful if you could articulate your key take-home messages from the equilibrium experiments for the reader's clarity. Notably, you mentioned that differences between RCM forcings are four times smaller than the overall model bias. In your opinion, can uncertainty be adequately captured by selecting just one RCM with an ensemble of ice sheet model parameters? The similarities between COSMO, RACMO, and HIRHAM raise questions about whether a recommendation for the future could be to choose MAR and one of the three RCMs to encompass uncertainty. Additionally, would you advocate for a separate glacial spin-up for MAR? These considerations could potentially enhance the abstract of your study.

In Figure 4 I cannot see the purple line.

Centennial Projections:

Regarding Figure 6: Could you confirm whether all Sea-Level Rise (SLR) contributions are subtracted by the control run? I might have overlooked this detail, and it would be helpful if you could explicitly state whether such subtraction has been performed. Notably, Seroussi et al. subtracted all the runs by control runs. Additionally, consider showcasing only the HadGEM2-ES results from Seroussi's work or, alternatively, emphasize the PISM run(s) for comparison.

On page 12, line 321, you mention calculating the maximum SLR contribution in a specific manner. I am curious about the choice of not subtracting the control run in this calculation. Considering that the glacial spin-up involved a single RACMO forcing and parameter set, wouldn't it be necessary to subtract the control run for each member individually? Especially after the results obtained from the equilibrium runs show so different behaviour for each RCM. This consideration becomes especially relevant when examining projected SLR uncertainties. Could you conduct this subtraction and provide insights into how it influences the projected uncertainties? Based on Figure D1, it appears that the control runs might not align with the values from 2005, particularly noticeable in the year 2300. Further clarification on this aspect would be appreciated.

Figure 7,9: which Year are you showing? I cannot see a purple line either.

Figure 9: Is there maybe a number to quantify this change? Mean thickness deviation for each RCM or something similar. This way we can see more easily if these difference arise more for the RCPs or RCMs.

---

## Author Comment (AC1)

**Reply to comments by Reviewer 1: The influence of present-day regional surface mass balance uncertainties on the future evolution of the Antarctic Ice Sheet (egusphere-2023-2233)**

**Summary of Changes**

We are grateful to the reviewer for evaluating our work, and the valuable and constructive comments that help improve the manuscript. To address the major comments, we now

- include individual thermal spinups for every RCM in the present-day equilibrium simulations.
- adapt the spinup for centennial-scale projections to limit model drift under preindustrial conditions.
- perform additional pre-industrial control runs.

Below, we respond to the reviewer's individual comments in detail and describe the actions we took to address them.

**Detailed response**

(Original report cited in italics)

**General remarks**

*In this study, the authors investigate how the applied present-day atmospheric climatology (specifically surface mass balance and air temperature) influences the simulated evolution of the Antarctic ice sheet. They employ outputs from four regional climate models (MAR, RACMO, COSMO, and HIRHAM), all boundary-forced by the ERA-interim climate reanalysis. To gauge the impact of the present-day climatology, the ice-sheet model PISM is used in two sets of experiments. In the first, the ice sheet evolves over 30 000 years under a constant present-day climate. In the second, Antarctic simulations spanning 1860 to 2300 are generated by adding HadGEM-ES anomalies to the respective RCM present-day climatologies. In both cases, for each RCM, an ensemble of simulations is run, covering uncertainties in model parameters such as enhancement factors, sliding parameters, and oceanic heat conductivity.*

*I appreciate the study's focus on quantifying uncertainties related to the atmospheric boundary conditions (and especially the surface mass balance) derived by regional climate models. I also value the concept of applying an ensemble of simulations sampling uncertainties in model structure for each RCM. However, I have concerns about the*

*methodology employed in the study, particularly regarding the model initialisation procedure.*

*In the PD-equilibrium experiment, the authors notably assess which RCM present-day climate triggers the greatest ice-sheet deviation from present-day observations. However, these results may be biased by the fact that the thermal spin-up is performed using RACMO's surface air temperature field. In my view, a more robust approach would involve conducting the thermal spin-up individually for each RCM. Alternatively, the thermal spin-up could use ERA-interim as direct boundary conditions (similar to the approach by Li et al., 2023, where ERA5 is employed to approximate the present-day climate).*

*The Reviewer raises an important point here. We therefore now perform individual thermal spinups for every RCM forcing and use those to carry out the long-term equilibrium simulations. The revised manuscript will feature these new simulations. Figure R1 illustrates the updated simulation results corresponding to Figure 3 from the initial manuscript. This change in initialization procedure resulted in an overall decrease of the ensemble mean ice mass change. However, the relative changes we focus on stay rather unaffected. On the level of individual simulation, we observe changes between the old RACMO thermal- and the individual spinup. Nevertheless, one of our main findings (ISM parameter configurations in which one RCM forcing triggers a (partial) collapse of the WAIS while other do not) still holds.*

[Figure]

**Figure R1**

*In their future projections experiment, the authors quantify the uncertainty arising from the choice in RCM baseline climatology and compare it with the spread observed in the ISMIP6*

*ensemble. However, I feel that the sea-level projections produced in this study are significantly influenced by the initialisation procedure. Based on my interpretation of Figure 2 and section 2.2 (if incorrect, I recommend clarifying the methods section), it appears that the simulations spanning 1860-2300 initiate directly from the fixed geometry thermal spin-up. If this is indeed the case, I believe it induces significant model drift, stemming from (i) the transition in the parameter-set model parameters for each ensemble member, (ii) the shift from the RACMO climatology used in the thermal spin-up to the present-day climatology of the respective investigated RCMs, and (iii) the abrupt imposition of pre-industrial anomalies derived from HadGEM-ES, while suddenly allowing the ice-sheet geometry to evolve. Model drift can be gauged by comparing control runs in Figure D1 (though it would be better approximated by a control run with a constant pre-industrial climate): the spread among the control runs from the four RCMs is similar to that observed in the RCP projections. Therefore, my impression is that the modelled responses stem more from model drift rather than from the climate forcing itself (especially given that the HadGEM anomalies are consistent across all simulations).*

The reviewer correctly understood our methodology for our centennial ISM-projections. We also agree with the reviewer that our simulations are subject to change due to the transition in the model parameters. In contrast, the divergence of individual runs due to the difference in the underling RCM forcing is in fact the subject of our investigation. Nevertheless, we acknowledge the necessity to quantify the model-specific drift under PI conditions. Therefore, we will provide additional PI control runs for a revised version of the manuscript.

*While I acknowledge that the study's aim is to quantify the influence of different forcings on future projections rather than to generate robust Antarctic sea-level projections, the results are nonetheless compared to such robust projections (i.e., the ISMIP6 ensemble). Given that current sea-level estimates prioritise minimal model drift by initialising the ice-sheet model with the starting climatology (whether pre-industrial, 1950, or present-day climatology, as seen in studies by, e.g.,  Seroussi et al., 2020; Reese et al., 2023; Coulon et al., 2023; Klose et al., 2023; Li et al., 2023), I find it challenging to grasp the value and interpretation of the numbers presented here.*

We would like to point out, that the ISMIP6 ensemble is associated with significant uncertainties and (in case of individual model contributions) featuring very large model drift (see supplements of the respective paper, specifically table B2). Our model drift is actually within the range of comparable model contributions in ISMIP6. This being said, our study aims to quantify the impact of different RCM baseline forcings on centennial changes in Antartica, rather than producing robust sea level rise projections. However, we acknowledge the fact that our findings will be more robust if our simulations would feature relatively small model drift.   To achieve this, we employ a thermal spinup using the mean of all four RCMs modified with PI anomalies, referred to as PI forcing. From there we initialize 18 individual freely evolving simulations forced by constant PI forcing for 300 years on 8km resolution. After those 300 years all our applied parameter combinations feature annual sea level contributions of less than 0.15 mm/yr (currently observed rates are ~0.3 mm/yr (Smith

et al. 2020)), with close to observation grounding zone positions and overall smaller-than-PD thinning rates for the WAIS (Smith et al. 2020). In a next step, for every parameter configuration we branch off simulations with the individual RCM forcings and historical and RCP anomalies, as well as PI control runs using either one of the four RCM forcings + PI anomalies or the mean of all four RCM forcings + PI anomalies. By doing so we can ensure considerably smaller model drift than in the previous version. Below you find an updated version of Figure 2, illustrating the new setup.

[Figure]

**Figure R2**

*If my understanding is accurate and the aforementioned points are applicable, I believe that the model initialisation procedure should be reconsidered, ensuring that the simulations start from an ice-sheet configuration in equilibrium with the initial pre-industrial boundary conditions (see, for example, the initialisation procedures in Li et al., 2023, Reese et al., 2023, Klose et al., 2023). It is worth noting, however, that even if such a strategy is applied to the present study's investigation of the four RCM present-day climatologies + GCM anomalies, it may be that the spread in different projections would result more from geometry differences arising during initialisation (and therefore potentially considered as 'initial state uncertainty') rather than from variations in the different RCM climatologies, to which the ice-sheet initial state is equilibrated. This is because identical temperature and SMB anomalies are added to these respective RCM present-day climatologies. Instead, the authors may consider investigating the spread due to different RCMs projections forced at their boundaries by identical GCM projections. Alternatively, they could apply an approach similar to that of Li et al. 2023, Klose et al., 2023, or Coulon et al., 2023, where climate models air temperatures and precipitation rates (in the case of the latter two, anomalies are added to RCM present-day climatologies) are corrected for elevation changes and used as input to a positive degree day scheme which then calculates surface melt and runoff amounts.*

The overall aim of this study is to investigate the interplay between different RCM forcings and the Antarctic ice sheet system to quantify the impact of the choice of RCM baseline forcing.

In the following we want to quickly motivate our methodology. Generally, one could

describe the simulated ice sheet state at time t by S(S_0, F_0, F_ano, θ, t). Here, S depends on several driving components which are the initial state S_0, the baseline forcing F_0, time dependent forcing anomalies F_ano (please note that F_ano carries the entire temporal information form from time 0 to t) and the parameter configuration θ. Now one could decompose S into terms which are only affected by one driving component (in the later referred to by $\mathfrak{S}$, single), terms which are affected by a mixing of two driving components (in the later referred to by $\mathfrak{D}$, double), terms which depends on a mixing of three driving components (in the later referred to by $\mathfrak{T}$, triple), and a term dependent on a mixing of four driving components (in the later referred to by $\mathfrak{Q}$, quadruple). Mathematically, one could achieve such separation by performing a Taylor expansion and then rearranging the individual terms into the desired shape. For simplicity we briefly demonstrate this in a simplified example with only two variables x and y.

Let's define f(x,y) as a differentiable function. Then we could derive the n-th order Taylor approximation around the point (a,b) as shown below.

$$
\begin{aligned}
f(x,y) &\approx \sum_{i=0}^{n}\sum_{j=0}^{n-i}\frac{1}{i!j!}\frac{\partial^{(i+j)}}{\partial x^i \partial y^j}f(a,b)(x-a)^i(y-b)^j \\
&= f(a,b) + \sum_{i=1}^{n}\frac{1}{i!}\frac{\partial^i}{\partial x^i}f(a,b)(x-a)^i + \sum_{i=1}^{n}\frac{1}{i!}\frac{\partial^i}{\partial y^i}f(a,b)(y-b)^i \\
&\quad + \sum_{i=1}^{n}\sum_{j=1}^{n-i}\frac{1}{i!j!}\frac{\partial^{(i+j)}}{\partial x^i \partial y^j}f(a,b)(x-a)^i(y-b)^j \\
&= f(a,b) + \mathfrak{S}_1(x) + \mathfrak{S}_2(y) + \mathfrak{D}_1(x,y)
\end{aligned}
$$

One can now rearrange the terms and identify $\mathfrak{S}_1$ and $\mathfrak{S}_2$, which only depend on x and y respectively and $\mathfrak{D}_1$, which only depends on mixing terms of x and y.
The same formalism could now be applied to decompose S(S_0, F_0, F_ano, θ, t) into individual parts.

$$
\begin{aligned}
\mathbf{S}(S_0,F_0,F_{ano},\theta,t) &= I + \mathfrak{S}_1(S_0,t) + \mathfrak{S}_2(F_0,t) + \mathfrak{S}_3(F_{ano},t) + \mathfrak{S}_4(\theta,t) \\
&\quad + \mathfrak{D}_1(S_0,F_0,t) + \mathfrak{D}_2(S_0,F_{ano},t) + \mathfrak{D}_3(S_0,\theta,t) \\
&\quad + \mathfrak{D}_4(F_0,F_{ano},t) + \mathfrak{D}_5(F_0,\theta,t) + \mathfrak{D}_6(F_{ano},\theta,t) \\
&\quad + \mathfrak{T}_1(S_0,F_0,F_{ano},t) + \mathfrak{T}_2(S_0,F_0,\theta,t) + \mathfrak{T}_3(S_0,F_{ano},\theta,t) + \mathfrak{T}_4(F_0,F_{ano},\theta,t) \\
&\quad + \mathfrak{Q}_1(S_0,F_0,F_{ano},\theta,t)
\end{aligned}
$$

Note that *I* denote the ice sheet state around which the Taylor expansion is performed (equivalent to f(a,b)) and vanishes for differences of simulations. As a quick additional explanation, the term $\mathfrak{S}_1$(F_0,t), only carries the ice sheet response due to the change of the baseline forcing, while $\mathfrak{D}_4$(F_0, F_ano, t) only carries the response which occurs due to the interaction of baseline forcing and the forcing anomalies.

To study the impact of the choice of the RCM forcing, one wants now to isolate the terms depending on the baseline forcing from the rest. This can be achieved by looking at the

difference between two simulations which only differ in F_0. It is important to mention that this requires the same initial state S_0 and parameter configuration θ, since otherwise terms containing S_0 or θ (but not the baseline forcing) would not vanish in the difference. In other words, if we were to compare simulations starting from different initial states, they would observe different driving stresses, which would lead to different outcomes. One could label this initial state uncertainty. For our study we want to avoid having different initial states for different RCMs, because that would also mean we would have translated some of the RCM difference into the initial state as well.

Nevertheless, for projection like simulation, one often assumes equilibrium at the pre-industrial initial state. This means all terms not containing the time dependent F_ano are time independent and could be summarized to the equilibrium initial state. The impact of the choice of RCM baseline forcing would then be described by terms containing both the RCM baseline (F_0) and the time dependent anomalies (F_ano). Those terms could be estimated by comparing different "projections" after subtracting them by their control run. We have adopted this approach in the revised version of the manuscript.

*In summary, I propose two key recommendations: (i) improve the initialisation procedure for the PD-equilibrium experiment, and (ii) reconsider the approach and methodology employed in the future projections experiment. These suggestions aim to positively contribute to refining the study's methodology for a more robust outcome. I align with the authors on the significance of elucidating and quantifying uncertainties in Antarctic projections related to surface mass balance, particularly those arising from regional climate models. Therefore, I believe that the study holds significant value for the scientific community and would be well suited for the scope of The Cryosphere. However, some major issues need to be addressed to make it a valuable contribution. Also, it is important to acknowledge that adequately addressing these recommendations would require rerunning the entire set of experiments, impacting not only the results but also reshaping the manuscript and its core findings.*

We thank the reviewer for the extensive review and fruitful comments and will briefly summarize our proposed actions here:

I) We improve the initialization setup for PD-equilibrium experiments, by performing individual thermal initializations for every RCM forcing.
II) We improve our projections by performing a 300-year PI-spinup simulation forced by the mean of all RCMs together with PI anomalies, to absorb the initialization shock and achieve ice sheet states with very small model drift.
III) We additionally improve the interpretability by assessing the isolated impact of the choice of RCM in interaction with GCM anomalies

**Specific points**

1. *Abstract, l. 10: It is not clear here what is meant by 'underlying ice sheet model parameterization'. Please clarify for better understanding.*

We refer here to the applied parametrization as representations of physical processes as well as the used parameters in those parametrizations. We will clarify this.

2. *Abstract, l.8-9: 'Uncertainties in future sea-level predictions of 8.7 (7.3-9.5) cm …' --> I find this sentence confusing, as uncertainties are mentioned, but it looks like the sea-level prediction and their uncertainties are presented. I think that it would be helpful to clarify what the numbers between brackets represent.*

We will clarify that the values represent min and max of the observed difference.

3. *Introduction, l. 24: Include a reference to Goelzer et al. 2020 ISMIP6 projections when comparing GrIS and AIS sea-level projections by 2100.*

In our study we only look at the AIS response to different RCM-forcing thus we include the Seroussi et al. 2020 ISMIP6 community paper.

4. *Introduction, l.28: I'd suggest adding more references to the concept of calibration reducing uncertainties in sea-level projections, such as, e.g., Edwards et al., 2019, Coulon et al., 2023, Nias et al., 2019, Lowry et al., 2021.*

Will be added.

5. *Introduction, l.29: please check these numbers.*

We did and they are correct., Compare to Seroussi et al. (2020) under 4.4: "Runs with HadGEM2-ES lead to significant sea level rise, with a mean ice mass loss of 96 mm SLE (standard deviation: 72 mm SLE)" and "Runs performed with CCSM4 show the largest ice mass gain, with a mean gain of 37 mm SLE (standard deviation: 34 mm SLE)"

6. *Introduction, l.31-32: I'd suggest adding a reference to Coulon et al. 2023 here, as they investigate uncertainties in ice-ocean and ice-atmosphere interactions.*

Will be added.

7. *Introduction, l.46: I'd suggest specifying that there is no specific reason to exclusively use one model given that other RCMs such as MAR are also designed to simulate polar regions by accounting for these processes.*

We will clarify this.

8. *Introduction, l.48: Seroussi et al. showed the influence of the choice of the GCM used to derive the forcing on Antarctic projections and not on its equilibrium state. Also, I don't think that they isolated the specific influence on the SMB, as the oceanic forcings also vary for each GCM.*

They did not isolate that. However, they showed that projections heavily depend on the applied GCM. We will clarify that. This Includes differences in ocean conditions as well.

9. *Methods, l. 65: refer to Mottram et al. 2021?*

Will be added.

*10. Figure 1: It is not clear to me from the caption what exactly is represented in the second line (figures f—j). I'd suggest clarifying this in the caption and maybe also in the figure itself.*

It shows the difference between ERA Interim and the RCMs. We will clarify this.

*11. Methods, l. 105: refer to Figure 2 here.*

Will be added.

*12. Methods, l. 112: an 8-km resolution was mentioned above, please clarify.*

The applied forcing is on 8 km resolution, while we perform our PD-equilibrium ice sheet simulations on 16 km resolution. Projections are carried out on 8 km resolution.

*13. Methods, l. 117: when is this evaluated? At the end of the 30-ky run? Please clarify, as the calibration step remains a little bit unclear so far. Also, in Table 2, an ensemble of 54 simulations is presented, which ones are the 14 selected ones? Maybe highlight them in bold in the table? It could also be interesting to visualise the obtained equilibrium ice-sheet geometry for each, maybe in the supplementary material.*

For our parameter ensemble we choose parameter combinations which after 15 ka of constant RACMO present-day forcing produced ice sheet geometries and velocities sensibly close to present-day observation. To minimize overfitting to the RACMO model we kept the ensemble spread relatively broad as seen in Figure 3. However, there might still be a RACMO bias in our ensemble. In contrast one could perform a parameter selection individually with each of the four RCM forcings and then combine all parameters. Nevertheless, this would be computationally very costly .

Table 2 only sates the individual values for the changed parameters, not all the combinations. We will add another table to the Supplements explicitly denoting every used combination.

*14. Methods, l. 138: I do not understand how the computation is rendered more cost-effective. Please clarify.*

The formulation used by us is unclear here, the computation is not rendered more cost efficient but there is just less computation to be done using our proposed methodology. We will specify this.

*15. Methods, l. 143: was an 8-km resolution also used for the thermal spin-up?*

The thermal spinup was also performed on a 16 km ice sheet model resolution as geometry is fixed so resolution does not really matter here.

*16. Methods, l. 143: I am confused by the abrupt shift from the fixed geometry thermal spin-up under present-day RACMO climatology to the RCM + 1860 anomaly climate for the historical spin-up. Why not start from an equilibrated state, i.e., as in the PD-equilibrium experiment, but for the 1860 climate, as is performed in e.g., Reese et al., 2023, Li et al., 2023, or Klose et al., 2023? Could the authors comment on this, and*

*ideally show the model drift when applying constant 1860 climate for the ensemble of simulations?*

As described above we will start from a PI-thermal spinup and additionally show PI model drift in the updated methods.

17. *Methods, l. 145: Does HadGEM2-ES has projections outputs available until 2300 under RCP2.6, 4.5 and 8.5? If not, how are the projections extended to 2300?*

Projected HadGEM2-ES until 2300 was provided.

18. *Methods, l. 151: Could the authors comment on why the list of ensemble parameters in Table 2 differs from the 'PD-equilibrium' experiment and what guided this choice? Also, I understand that configurations without long-term stability are no longer excluded, it would be good to clarify which ones are the ones selected by the calibration procedure.*

We perform PD-equilibrium simulations on 16km while we perform projections on 8km resolution. Therefore, other parameters were selected. Nevertheless. The applied parameters for the projection like simulations will change such that model drift under constant PI forcing is minimized as requested by the Reviewers. All applied parameter combinations will be stated in the supplements.

19. *Methods, l. 151-152: What is meant by 'model spin up' here? Is the thermal spin up, or the short historical run? Please clarify.*

Thermal spinup+historical run.

20. *Methods, l. 153: What initial ice-sheet configuration is referred to here? If my understanding is correct, the ice-sheet initial state obtained from the thermal spin-up was produced with fixed ice-sheet geometry. Deviations with respect to ice thickness should therefore be zero.*

After the thermal spinup, ice thickness deviations are zero. The sentence is a little misleading, since we wanted to state that we expect the model to have relatively large ice thickness deviations between the equilibrium state and observations, since no extensive initialization (e.g. inversion) was performed. We will clarify this.

21. *Methods, l. 155: 'have often been used in the past' --> I'd suggest adding some references to support this. I would also suggest clarifying what exactly is meant by 'simple spin-up' routines, is it the thermal spin-up?*

We will provide additional references.

22. *Figure 2, caption: 'First the model is initialized from present-day ice sheet observations. Then a 200-ka thermal spin up is performed.' à My understanding was that the initialisation was the thermal spin up itself. Here, it is implied that the spin-up is performed after a first initialisation procedure. Please clarify.*

With initialization here we mean that we regrid from the observational grid to our 16km PISM grid. From there on perform the thermal spinup.

23. *Figure 2: The figure says 'BEDMAP topography' while Bedmachine is mentioned in the manuscript, please correct. Also, I would suggest specifically writing on the figure that the thermal spin up is performed with a fixed ice-sheet geometry.*

We will clarify this.

24. *Figure 3: I'd suggest clarifying in the figure caption that these are the timeseries under constant present-day conditions, i.e., the PD-equilibrium experiment. In addition, please clarify what change rate is meant in figures (e-h). Also, I suppose from the figures that (i-l) represent the change in ice fraction area? Finally, please clarify how the total ice mass change is translated in m s.l.e? Is only the ice above floatation accounted for here?*

(e-h) are the change rate of above floatation ice mass. (i-l) shows the ice area extend relative to PD observations. Only the ice above flotation is account for here. We will clarify this in the caption.

25. *Results, l.168: Could this be influenced by the fact that the thermal spin up was performed with RACMO only? The trend would hence not be influenced by the RCM itself, but rather by the difference between the RCM surface temperature field and RACMO's one. Why not performing a thermal spin up for each RCM to exclude this possibility?*

See response above.

26. *Results, l.176: I'd refer to Figure 1 here.*

Will be added.

27. *Figure B3: Why not simply combine figures 3 and B3?*

Figure B3 was produced in reply to minor revisions by the Editor. For readability of Figure 3 we still tend to keep this as additional information in the supplements.

28. *Results, l.183-184: What is meant by 'mainly driven by ice-sheet model parameterisation' here? I think that this requires more clarification.*

Due to the model intrinsic parametrization, especially the applied heuristics calculating the till friction angle, results in anomalies to present-day observations, seen in many studies (Martin et al. 2011, Albrecht et al. 2020, Sutter et al. 2023). We will specify this in the manuscript.

29. *Results, l.188: '(effect of ice sheet model spin up and parameter choices)' --> Again, I think that this requires a bit more explanation.*

As stated above we mean here the effect the applied parameterizations and chosen parameters, as well as the chosen initial state (e.g. ice- geometry and temperature), has.

In our decomposition introduced above that would be mainly D(theta, t) and AD(S_0, theta, t). Nevertheless, we will clarify this in the manuscript together with the Point mentioned above.

30. *Results, l.190-193: Alternatively, a control run under constant present-day climate conditions used for the thermal spin up could be deduced from each simulation from the ensemble, allowing to isolate changes in the AIS due to the evolving climate for each configuration.*

We don't understand what the reviewer intends to say here referring to l.190-193.

31. *Results, l.195: It could be interesting/helpful to the reader to highlight, on one or several figures, some of the regions/locations that you refer to in the text.*

Since there are already a lot of details in the Figure, we try to avoid overloading the Figure with additional information.

32. *Figure B4: It is not clear to me what exactly is represented in Figure B4.*

Figure B4 (a) shows the mean state of the ice sheet after 30 ka when forced with the mean out of the four RCM forcings. (b) shows the difference between the ice sheet thickness depicted in (a) compared with the mean of the simulations forced with the RCMs individually.

33. *Results, l.205: Why were these specific simulations selected? Where do they lie compared to the rest of the ensemble?*

They were chosen, because they showed a collapse under one RCM forcing but not under another. Since we now intend to not discard simulations on being long term stable under RACMO forcing, all simulations will also be depicted in Figure 3.

34. *Figure 5: Writing the parameter values in each of the subfigures is confusing as it gives the impression that each parameter value is associated with the panel itself, I would suggest removing it. In addition, please clarify in the caption what experiment is represented in the figure.*

We will remove the parameters from the figure since they are named in the caption.

35. *Results, l.212: please clarify what is meant by 'similar' here.*

Collapse with one forcing but no collapse with another. For clarity, we will remove the restriction on 100 kyrs stability with RACMO forcing, such that we won't have to separate cases anymore.

36. *Results, l.216-219: in which figures can we see this? Please clarify. It would also probably be easier to indicate the parameter-set subset on the figure directly.*

Figure C1. Will be merged with Figure 5.

37. *Results, l.213-219: as a few of these simulations do not seem to have reached a steady state nor a quasi-steady-state yet, one could wonder whether running these simulations for more than 30kyr would lead to a WAIS collapse in all of the configurations, implying that the committed ice-sheet state is mainly driven by the parameter set itself, while the RCM climatology modulates the timing of the potential collapse? This is only a guess, but it could be interesting to discuss this somewhere?*

We will add this to the discussion.

38. *Results, l.223-224: Figure D1 seems like an important figure which, I believe, has its place (along with its discussion) in the main manuscript.*

Since the focus of our study is not the projection itself but the impact of the choice of RCM on the projection uncertainty, we choose to leave it in the Appendix while updating Figure 6 with additional control runs.

39. *Results, l.242-243: SMB over the ice shelves has no direct contribution to sea-level rise, but it does indirectly influence the ice-shelves stability and hence buttressing effect on the ice-sheet flow. Maybe it is worth briefly commenting on this?*

We will mention that the SMB over ice shelves has an effect on grounded ice-flow.

40. *Figures 7, D2-D3: I see no purple line on these figures. Also, the grey line does not seem to be the observed present-day grounding-line position. Are these the median grounding line positions? Also, are these the ice-sheet configurations by the end of the simulations, i.e., 2300? Please clarify.*

The purple line was removed for readability in, we will adjust the caption. The black line indicates the simulated grounding line position.

41. *Figures 7, D2-D5: I find the use of the difference to the common mean hard to read and interpret. Alternatively, a control run under constant pre-industrial climate conditions could be deduced from each simulation, allowing to isolate changes in the AIS due to the evolving climate for each configuration (something similar is performed in Li et al.'s Exp. CMIP6_RAW_1850-2100).*

We can add additional plots showing the difference to a control run. However, the aim of Figure D2-D5 is to show the difference between the RCM forcings. Comparing with the control run would also show the differences due to the GCM anomalies (compare with discussion above).

42. *Results, l.267-269: I think that this makes sense, given that the parameters included in the ensemble do not have a strong impact in this region, which is instead strongly influenced by the SMB.*

We will remove the word "surprising" in the text.

43. *Results, l.276: What about the control (i.e., constant present-day as of 2005) simulations? It could be interesting to show these as well to have a better grasp of the influence of this signal.*

We will add a panel showing the control.

44. *Figures D4-D5: It should be clarified in the figures' captions that these represent the ensemble member 10 only. Also, what do the different coloured lines represent in these figures? Overall, it would be good to clarify figure captions throughout the manuscript.*

Thank you for pointing that out. We will clarify the caption. For a revised version we'll improve the figure captions.

45. *Results, l.281-282: I don't think that I would say that the RCM baseline will 'significantly affect the onset and pacing of a marine ice sheet instability'. First, I don't believe that Figure 8, given that it is a snapshot at year 2300, allows us to draw conclusions about the pacing itself. In addition, except for the 5 and 95 percentiles, the grounding line positions are overall relatively similar. I think that it is more correct to say that the choice of the RCM baseline modulates the grounding line retreat. Also, I don't think that it makes sense to refer to a marine ice sheet instability mechanism here. We do not know whether a self-reenforcing retreat has been triggered. I would simply refer to a grounding-line retreat.*

We will remove the statement on "pacing" of the grounding line retreat as well as the usage of "marine ice sheet instability" here as we didn't assess whether our model results show a self-reenforcing retreat.

46. *Figure 8: It is not clear to me how the percentiles of grounding-line positions are calculated. Could the authors specify it in the caption?*

We calculated grounded ice mask density, which states for every gridbox i,j the percentage of simulations which have grounded ice. From there contour lines were drawn for the individual percentiles. We will add a precise description on how those percentiles were calculated to the caption.

47. *Results, l.287: Here it is referred to ensemble member n°10 while before the ensemble members were referenced using letters (AY, etc.), maybe consider using either letters or numbers for both for consistency?*

We will clarify this.

48. *Results, l.288-289:'similar to the already observed patterns in the present-day equilibrium runs' à I am not sure which figure I should refer to for the comparison, I am guessing Figure 4, but it would be good to specify.*

Indeed Figure 4, we will mention this in the text.

49. *Discussion, l.304-305: I think that this formulation is clearer than the one used in the abstract, maybe use an equivalent sentence as this one for the abstract as well? Also, I think that it is important to specify that these are the maximum differences between two RCM configurations.*

We will update the abstract guided by the sentence in the discussion.

*50. Discussion, l.306-308: I find that how both (different) numbers are compared is confusing, as, e.g., 8.7(7.3 – 9.5) represents a spread in sea-level contribution, while 9.6 +- 7.2 represents the sea-level contribution itself. I'd suggest presenting the spread of the ISMIP6 ensemble instead. The authors may also consider calculating an equivalent indicator as the 'mean maximum sea level contribution difference' on the ISMIP6 ensemble for a more robust comparison.*

We will compare with the IMSPI6 spread directly. However, since the ISMIP6 data contains simulations from different ice sheet models with different parameters used, a calculation of a 'mean maximum sea level contribution difference' as described in our Methods is not possible.

*51. Discussion, l.311: My impression is that the uncertainty presented here is instead mainly driven by the initialisation procedure. I think that this requires a more thorough discussion and presentation of control (i.e., constant pre-industrial climate) simulations.*

As discussed above we will provide extensive control runs in a revised version.

*52. Discussion, l.352: 'may be simulated'. It is in fact only for specific RCM and parameter set that divergences appear. Your median grounding line positions are in fact relatively similar.*

We are unsure to what exactly the reviewer refers in l.352.

*53. Discussion, l.329: what is meant by 'unforced' grounding-line retreat here?*

We mean not forced by GCM projection anomalies. We will clarify this in the text.

*54. Discussion, l.341&345: Grounding-line retreat does not necessarily imply reduced buttressing and hence acceleration in ice flow…*

We will correct this and only mention potential reduced buttressing.

*55. Discussion, l.355: I don't understand what is meant by 'the ice sheet gradually responds to the SMB forcing', please clarify.*

Indeed, this formulation is misleading. We mean a response in line with the SMB forcing e.g. ice thickness increases for SMB increase and vice versa. We will change the text accordingly.

*56. Discussion, l.358-359: I don't understand this. The evolution of the ice flow can be investigated with the evolution of the ice velocities through time.*

We now have checked the velocity fields. The ice divides are shifting in all runs in response to the applied forcing. However, a more in-depth analysis would be necessary to assess if this is the main driver of the observed behavior

*57. Discussion, l. 360: I find this title a little confusing. I would suggest reformulating it.*

We will simplify to: Parameter sensitivity

58. *Conclusion, l.377-378: I don't think that the differences in thickness and grounding line positions that are presented here may be considered as 'considerable' (see, especially, Figure B1)*

It is true that RCM induced differences in thickness and grounding line positions are not always considerable when compared with difference to present-day observation. However, there are cases where this is the case (see Figure 5, C1). Additionally, RCM induced differences can reach values up to several hundred meters in some regions, which is in absolute and relative terms quite considerable.

59. *Figure D1: How come that Figure D1 shows only one curve for the control runs? What parameter values are used for these control runs? For consistency, control simulations should be performed for each parameter configuration.*

We have performed control simulations for every parameter configuration. For simplicity Figure D1 only shows the ensemble mean of all control runs per RCM forcing. We will state this clearly in the caption.

60. *Appendix D, l.399: 'minor ice loss' --> I am not sure that the 1860-2005 ice loss (of several dm) can be considered 'minor'. It is the same order of magnitude as the sea-level contribution between 2100 and present under RCP8.5, as shown in Figure 6. Also, as mentioned above, I suggest moving this entire section to the main manuscript.*

Indeed, the ice loss is not minor if compared with ice-loss until the year 2100. Most of this ice loss can probably be attributed to the initialization shock when the ice starts to evolve freely. With our new simulation setup proposed above we try to absorb this shock beforehand and reach a considerable smaller drift. However, we would like to point out here, that there are many publications where the SLE difference to PD in the respective spinup/initialisation/control experiments amounts to several meters. Compared to this several dm can be considered small.

*Overall,*

61. *the methodology, particularly outlined in Section 2.2, is unclear. The inconsistent use of terms such as 'spin up' and 'initialisation' makes it challenging to comprehend the precise procedures, even with the aid of Figure 2, especially for the 'Future projections' experiment (section 2.2.2). Similarly, the calibration procedure, and how it varies between experiments (resulting in different parameter values) remains unclear. To enhance clarity, the study would benefit from a clear list of experiments, similar to Table 1 in Li et al. (2023), where climate forcing, initial conditions, and objectives are explicitly stated.*

Thank you for pointing this out. We will add a table describing all simulations performed. Further we will ensure consistent use of the terms"initialization" and "spinup". As described above we will also revise our methods according to the reviewer comments, therefore Section 2.2.2 is expected to change as well.

62. *the figure captions should be enhanced for consistency, providing clear information on the represented experiments, years, and the significance of various elements (e.g., grounding-line position). Improved consistency and clarity in figure captions would enhance the overall understanding of the figures and contribute to a more straightforward interpretation of the study's findings.*

As stated before we will improve the Figure captions to allow for better accessibility.

63. *the discussion lacks consideration and comparison with related works (other than ISMIP6).*

The comments by the reviewer have brought up several important publications which will be included in the discussion of a revised manuscript.

**Minor comments/Typos**

64. *Abstract, l.1: remove coma after 'impacts'.*
65. *Abstract, l.7: 'constant forcing quasi-equilibrium state' --> I find this formulation confusing, try to rephrase?*
66. *Abstract, l.8: 'uncertainties of' --> uncertainties in?*
67. *Abstract, last sentence: remove coma after 'importance'.*
68. *Introduction, l. 17: add come after 'Until the end of this century'*
69. *Introduction, l. 17: 'see level rise'*
70. *Introduction, l. 25: 'century's' --> centuries*
71. *Introduction, l. 32: 'The latter, estimates' --> 'Uncertainties in estimates of'?*
72. *Introduction, l.38-41: I'd suggest splitting this sentence in two.*
73. *Introduction, l.50: I'd suggest splitting this sentence in two: 'We address the following questions:…'*
74. *Methods, l. 70: 'drainage basis'*
75. *Methods, l. 70: remove come after 'All four models'*
76. *Methods, l. 83: 'togeher'*
77. *Methods, l. 86: 'Antarctic Ice sheet' --> 'Antarctic Ice Sheet' for consistency. I believe that this is the case at other places in the text, please check.*
78. *Methods, l. 90: 'shelf's'*
79. *Methods, l. 100: to improve the readability of this sentence, consider using 'two model set ups: (i) …, and (ii) …'.*
80. *Methods, l. 102: 'scenario' --> 'scenarios'.*
81. *Methods, l. 102: 'BedMachine'.*
82. *Methods, l. 104: remove come after (2004).*
83. *Methods, l. 112: 'on 16 km resolution' à 'at 16 km resolution'.*
84. *Methods, l. 113: 'RCM-'*
85. *Methods, l. 113: 'we employ' --> 'we run/produce'?*
86. *Methods, l. 118: 'An additional constrained'*
87. *Results, l. 165 and l.172: 'initialization shock' --> 'initial shock'?*
88. *Results, l. 229: 'maxmimum'*
89. *Results, l. 244: 'SMB The accumulated…'*

90. *Figure 8, caption: 'siumaltions'*
91. *Results, l. 285-286: remove comes after 'both' and 'forcing sets'*
92. *Results, l. 286: 'chosen ice sheet model parameter choice'.*
93. *Discussion, l.300: 'onto' --> 'on'?*
94. *Discussion, l.322: 'forcing data' --> 'baseline climatology'?*
95. *Discussion l.341&345: 'butsstressing'*
96. *Discussion l.345: 'In these simulation'*

Since we expect out Manuscript to change quite significantly, we will implement those comments unless the text passage hasn't been changed.

In conclusion, we would like to thank the Reviewer for his extensive and detailed comments. We are convinced that our proposed changes will significantly improve the manuscript.

**References**

*Coulon et al.: Disentangling the drivers of future Antarctic ice loss with a historically-calibrated ice-sheet model, EGUsphere [preprint], https://doi.org/10.5194/egusphere-2023-1532, 2023.*

*Goelzer, H., Nowicki, S., Payne, A., Larour, E., Seroussi, H., Lipscomb, W. H., Gregory, J., Abe-Ouchi, A., Shepherd, A., Simon, E., Agosta, C., Alexander, P., Aschwanden, A., Barthel, A., Calov, R., Chambers, C., Choi, Y., Cuzzone, J., Dumas, C., Edwards, T., Felikson, D., Fettweis, X., Golledge, N. R., Greve, R., Humbert, A., Huybrechts, P., Le clec'h, S., Lee, V., Leguy, G., Little, C., Lowry, D. P., Morlighem, M., Nias, I., Quiquet, A., Rückamp, M., Schlegel, N.-J., Slater, D. A., Smith, R. S., Straneo, F., Tarasov, L., van de Wal, R., and van den Broeke, M.: The future sea-level contribution of the Greenland ice sheet: a multi-model ensemble study of ISMIP6, The Cryosphere, 14, 3071–3096, https://doi.org/10.5194/tc-14-3071-2020, 2020.*

*Li et al.: Climate model differences contribute deep uncertainty in future Antarctic ice loss. Sci. Adv. 9, eadd7082 (2023). DOI:10.1126/sciadv.add7082*

*Nias, I. J., Cornford, S. L., Edwards, T. L., Gourmelen, N., & Payne, A. J. (2019). Assessing uncertainty in the dynamical ice response to ocean warming in the Amundsen Sea Embayment, West Antarctica. Geophysical Research Letters, 46, 11253–11260. https://doi.org/10.1029/2019GL084941*

*Klose, A. K., Coulon, V., Pattyn, F., and Winkelmann, R.: The long–term sea–level commitment from Antarctica, The Cryosphere Discuss. [preprint],*

*https://doi.org/10.5194/tc-2023-156, in review, 2023.*

*Lowry, D.P., Krapp, M., Golledge, N.R. et al. The influence of emissions scenarios on future Antarctic ice loss is unlikely to emerge this century. Commun Earth Environ 2, 221 (2021). https://doi.org/10.1038/s43247-021-00289-2*

*Reese, R., Garbe, J., Hill, E. A., Urruty, B., Naughten, K. A., Gagliardini, O., Durand, G., Gillet-Chaulet, F., Gudmundsson, G. H., Chandler, D., Langebroek, P. M., and Winkelmann, R.: The stability of present-day Antarctic grounding lines – Part 2: Onset of irreversible retreat of Amundsen Sea glaciers under current climate on centennial timescales cannot be excluded, The Cryosphere, 17, 3761–3783, https://doi.org/10.5194/tc-17-3761-2023, 2023.*

**References:**

Albrecht, T., Winkelmann, R., and Levermann, A.: Glacial-cycle simulations of the Antarctic Ice Sheet with the Parallel Ice Sheet Model (PISM) – Part 2: Parameter ensemble analysis, The Cryosphere, 14, 633–656, https://doi.org/10.5194/tc-14-633-2020, 2020.

Martin, M. A., Winkelmann, R., Haseloff, M., Albrecht, T., Bueler, E., Khroulev, C., and Levermann, A.: The Potsdam Parallel Ice Sheet Model (PISM-PIK) – Part 2: Dynamic equilibrium simulation of the Antarctic ice sheet, The Cryosphere, 5, 727–740, https://doi.org/10.5194/tc-5-727-2011, 2011.

Seroussi, H., Nowicki, S., Payne, A. J., Goelzer, H., Lipscomb, W. H., Abe-Ouchi, A., Agosta, C., Albrecht, T., Asay-Davis, X., Barthel, A., Calov, R., Cullather, R., Dumas, C., Galton-Fenzi, B. K., Gladstone, R., Golledge, N. R., Gregory, J. M., Greve, R., Hattermann, T., Hoffman, M. J., Humbert, A., Huybrechts, P., Jourdain, N. C., Kleiner, T., Larour, E., Leguy, G. R., Lowry, D. P., Little, C. M., Morlighem, M., Pattyn, F., Pelle, T., Price, S. F., Quiquet, A., Reese, R., Schlegel, N.-J., Shepherd, A., Simon, E., Smith, R. S., Straneo, F., Sun, S., Trusel, L. D., Van Breedam, J., van de Wal, R. S. W., Winkelmann, R., Zhao, C., Zhang, T., and Zwinger, T.: ISMIP6 Antarctica: a multi-model ensemble of the Antarctic ice sheet evolution over the 21st century, The Cryosphere, 14, 3033–3070, https://doi.org/10.5194/tc-14-3033-2020, 2020.

J. Sutter, A. Jones, T. L. Frölicher, C. Wirths, and T. F. Stocker. Climate intervention on a high-emissions pathway could delay but not prevent West Antarctic Ice Sheet demise. *Nature Climate Change*, 2023. doi:10.1038/s41558-023-01738-w.

Ben Smith *et al.,* Pervasive ice sheet mass loss reflects competing ocean and atmosphere processes. Science**368**,1239-1242(2020). DOI:10.1126/science.aaz5845

---

## Author Response (AR1)

**Reply to comments by Reviewer 1: The influence of present-day regional surface mass balance uncertainties on the future evolution of the Antarctic Ice Sheet (egusphere-2023-2233)**

**Summary of Changes**

We are grateful to the reviewer for evaluating our work, and the valuable and constructive comments that help improve the manuscript. To address the major comments, we now

- include individual thermal spinups for every RCM in the present-day equilibrium simulations.
- adapt the spinup for centennial-scale projections to limit model drift under preindustrial conditions.
- perform additional pre-industrial control runs, which are used to assess the isolated impact of the choice of RCM in interaction with GCM anomalies.

Below, we respond to the reviewer's individual comments in detail and describe the actions we took to address them.

**Detailed response**

(Original report cited in italics)

**General remarks**

*In this study, the authors investigate how the applied present-day atmospheric climatology (specifically surface mass balance and air temperature) influences the simulated evolution of the Antarctic ice sheet. They employ outputs from four regional climate models (MAR, RACMO, COSMO, and HIRHAM), all boundary-forced by the ERA-interim climate reanalysis. To gauge the impact of the present-day climatology, the ice-sheet model PISM is used in two sets of experiments. In the first, the ice sheet evolves over 30 000 years under a constant present-day climate. In the second, Antarctic simulations spanning 1860 to 2300 are generated by adding HadGEM-ES anomalies to the respective RCM present-day climatologies. In both cases, for each RCM, an ensemble of simulations is run, covering uncertainties in model parameters such as enhancement factors, sliding parameters, and oceanic heat conductivity.*

*I appreciate the study's focus on quantifying uncertainties related to the atmospheric boundary conditions (and especially the surface mass balance) derived by regional climate*

*models. I also value the concept of applying an ensemble of simulations sampling uncertainties in model structure for each RCM. However, I have concerns about the methodology employed in the study, particularly regarding the model initialisation procedure.*

*In the PD-equilibrium experiment, the authors notably assess which RCM present-day climate triggers the greatest ice-sheet deviation from present-day observations. However, these results may be biased by the fact that the thermal spin-up is performed using RACMO's surface air temperature field. In my view, a more robust approach would involve conducting the thermal spin-up individually for each RCM. Alternatively, the thermal spin-up could use ERA-interim as direct boundary conditions (similar to the approach by Li et al., 2023, where ERA5 is employed to approximate the present-day climate).*

We now perform individual thermal spinups for every RCM forcing and use those to carry out the long-term equilibrium simulations. Please find an extended description of the updated simulation setup in section 2.2.1 in the revised manuscript, which features these new simulations. Figure R1 illustrates the updated simulation results corresponding to Figure 3 from the initial manuscript. This change in initialization procedure resulted in an overall decrease of the ensemble mean ice mass change. However, the relative changes we focus on stay rather unaffected. On the level of individual simulations, we observe changes between the old RACMO thermal-spinup and the individual RCM thermal-spinups. Nevertheless, our main findings (ISM parameter configurations in which one RCM forcing triggers a (partial) collapse of the WAIS while other do not) still holds, as shown in Figure R4 (Figure 5 in the revised manuscript).

[Figure]

**Figure R1:** *Time series of the total change in ice mass above flotation since the start of the simulation (a-d), the annual rate of change in ice mass above flotation (e-h), and the fraction of grounded (solid line) and floating (dashed line) ice area (i-l) relative to observations for the four different RCM forcing fields. Bold line shows ensemble median while shaded lines indicate the individual ensemble members.*

*In their future projections experiment, the authors quantify the uncertainty arising from the choice in RCM baseline climatology and compare it with the spread observed in the ISMIP6 ensemble. However, I feel that the sea-level projections produced in this study are*

*significantly influenced by the initialisation procedure. Based on my interpretation of Figure 2 and section 2.2 (if incorrect, I recommend clarifying the methods section), it appears that the simulations spanning 1860-2300 initiate directly from the fixed geometry thermal spin-up. If this is indeed the case, I believe it induces significant model drift, stemming from (i) the transition in the parameter-set model parameters for each ensemble member, (ii) the shift from the RACMO climatology used in the thermal spin-up to the present-day climatology of the respective investigated RCMs, and (iii) the abrupt imposition of pre-industrial anomalies derived from HadGEM-ES, while suddenly allowing the ice-sheet geometry to evolve. Model drift can be gauged by comparing control runs in Figure D1 (though it would be better approximated by a control run with a constant pre-industrial climate): the spread among the control runs from the four RCMs is similar to that observed in the RCP projections. Therefore, my impression is that the modelled responses stem more from model drift rather than from the climate forcing itself (especially given that the HadGEM anomalies are consistent across all simulations).*

We indeed initialized our simulations from the thermal-spinup in the original setup for our centennial ISM-projections. We agree with the reviewer that our simulations are subject to model drift due to the different model parameter-combinations. In contrast, the divergence of individual runs due to the difference in the underling RCM forcing is in fact the subject of our investigation. Based on the reviewer's suggestion, we now quantify the model-specific drift under PI-conditions. We performed additional PI control runs for the revised version of the manuscript shown in Fig. R2 below (Fig. 6 of the revised manuscript).

[Figure]

**Figure R2:** *Time series of the median (solid lines) total ice mass change (a-c) and the annual rate of change (d-f) for the four different RCM forcing fields and the RCP2.6, RCP4.5, and RCP8.5 climate scenarios. Dashed lines represent the median PI-control simulations. Shadings indicate the 5th to 95th quantile.*

We agree that the initialisation of the historical simulations from the fixed BedMachine geometry might impose substantial model drift. To adress this, we improved our simulation setup for the revised manuscript (see Figure R3 below). Now, we employ a thermal-spinup using the mean of all four RCMs modified with PI-anomalies, referred to as PI forcing. From

there we initialize 12 individual and freely evolving simulations forced by constant PI forcing for 300 years on 8km resolution. After those 300 years all our applied parameter combinations feature annual sea level contributions of less than 0.15 mm/yr (currently observed rates are ~0.3 mm/yr (Smith et al. 2020)), grounding line positions close to observations and overall smaller-than-PD thinning rates for the WAIS (Smith et al. 2020). In a next step, for every parameter configuration we branch off simulations with the individual RCM forcings and historical and RCP anomalies, as well as PI control runs using one of the four RCM forcings + PI anomalies. By doing so we can ensure considerably smaller model drift than in the previous version. Our PI-control simulations feature a drift within the range of comparable model contributions in ISMIP6. For comparison we added a summary of our integrated model drift in Table D2 in the revised manuscript. Integrated from 2015 to 2100, our PI-control simulations show a mean drift of 50939 Gt (MAR) -1098 Gt (RACMO) -8536 Gt (HIRHAM) and –46946 Gt (MAR). For comparison, the ISMIP6 ensemble shows a drift from 47080 Gt (ULB_fETISh_32_std) to –458 Gt (DOE_MALI_std). Please be aware that since the relaxation was performed with the mean of all RCM, we expect larger trends for RCM forcings which differ more from the mean (e.g. MAR and COSMO). Additionally, Figure R3 (Figure 2 in the manuscript) illustrates the new centennial simulation setup. Consequently, we have also updated Section 2.2.2 as well added a Table (Tab. B2) listing all performed simulations.

[Figure]

**Figure R3:** *Illustration of the present-day equilibrium simulation and future projections setup*

*While I acknowledge that the study's aim is to quantify the influence of different forcings on future projections rather than to generate robust Antarctic sea-level projections, the results are nonetheless compared to such robust projections (i.e., the ISMIP6 ensemble). Given that current sea-level estimates prioritise minimal model drift by initialising the ice-sheet model with the starting climatology (whether pre-industrial, 1950, or present-day climatology, as seen in studies by, e.g., Seroussi et al., 2020; Reese et al., 2023; Coulon et al., 2023; Klose et al., 2023; Li et al., 2023), I find it challenging to grasp the value and interpretation of the numbers presented here.*

We would like to point out, that the ISMIP6 ensemble is associated with significant uncertainties and (in case of individual model contributions) featuring very large model drift (see supplements of the respective paper, specifically table B2). With the new simulation setup, our model drift is within the range of comparable model contributions in ISMIP6 (cmp. Table D2 in the revised manuscript).

*If my understanding is accurate and the aforementioned points are applicable, I believe that the model initialisation procedure should be reconsidered, ensuring that the simulations start from an ice-sheet configuration in equilibrium with the initial pre-industrial boundary conditions (see, for example, the initialisation procedures in Li et al., 2023, Reese et al., 2023, Klose et al., 2023). It is worth noting, however, that even if such a strategy is applied to the present study's investigation of the four RCM present-day climatologies + GCM anomalies, it may be that the spread in different projections would result more from geometry differences arising during initialisation (and therefore potentially considered as 'initial state uncertainty') rather than from variations in the different RCM climatologies, to which the ice-sheet initial state is equilibrated. This is because identical temperature and SMB anomalies are added to these respective RCM present-day climatologies. Instead, the authors may consider investigating the spread due to different RCMs projections forced at their boundaries by identical GCM projections. Alternatively, they could apply an approach similar to that of Li et al. 2023, Klose et al., 2023, or Coulon et al., 2023, where climate models air temperatures and precipitation rates (in the case of the latter two, anomalies are added to RCM present-day climatologies) are corrected for elevation changes and used as input to a positive degree day scheme which then calculates surface melt and runoff amounts.*

We believe that generating SMB via a PDD-approach would convolute the direct effects of the RCM-forcing and the PDD-related uncertainties, the latter being considerable. Therefore, we prefer not to apply PDD for this kind of study.

The overall aim of this study is to investigate the interplay between different RCM forcings and Antarctic Ice Sheet dynamics to quantify the impact of the choice of RCM baseline forcing.

The following is a detailed motivation of our methodology which can be summarized quickly as: We decompose the simulated ice sheet response into terms dependent on one or a combination of several drivers of ice sheet evolution (e.g. SMB, initial geometry, parameter configuration) For future projections we consider the term driven by an interaction between the RCM and the GCM forcing as the best estimate for the RCM induced projection uncertainty. We have adopted this approach in the revised version of the manuscript, and now present the "control-corrected" estimates of $\Delta slr\_max$ as our main estimates of RCM induced uncertainties.

Generally, one could describe the simulated ice sheet state at time t by $S(S\_0, F\_0, F\_{ano}, \theta, t)$. Here, S depends on several driving components which are the initial state $S\_0$, the baseline forcing $F\_0$, time dependent forcing anomalies $F\_{ano}$ (please note that $F\_{ano}$ carries the entire temporal information form from time 0 to t) and the parameter

configuration θ. Now one could decompose S into terms which are only affected by one driving component (in the later referred to by $\mathfrak{S}$, single), terms which are affected by a mixing of two driving components (in the later referred to by $\mathfrak{D}$, double), terms which depends on a mixing of three driving components (in the later referred to by $\mathfrak{T}$, triple), and a term dependent on a mixing of four driving components (in the later referred to by $\mathfrak{Q}$, quadruple). Mathematically, one could achieve such separation by performing a Taylor expansion and then rearranging the individual terms into the desired shape. For simplicity we briefly demonstrate this in a simplified example with only two variables x and y.

Let's define f(x,y) as a differentiable function. Then we could derive the n-th order Taylor approximation around the point (a,b) as shown below.

$$
\begin{aligned}
f(x,y) &\approx \sum_{i=0}^{n}\sum_{j=0}^{n-i} \frac{1}{i!j!}\frac{\partial^{(i+j)}}{\partial x^i \partial y^j} f(a,b)(x-a)^i(y-b)^j \\
&= f(a,b) + \sum_{i=1}^{n}\frac{1}{i!}\frac{\partial^i}{\partial x^i} f(a,b)(x-a)^i + \sum_{i=1}^{n}\frac{1}{i!}\frac{\partial^i}{\partial y^i} f(a,b)(y-b)^i \\
&+ \sum_{i=1}^{n}\sum_{j=1}^{n-i} \frac{1}{i!j!}\frac{\partial^{(i+j)}}{\partial x^i \partial y^j} f(a,b)(x-a)^i(y-b)^j \\
&= f(a,b) + \mathfrak{S}_1(x) + \mathfrak{S}_2(y) + \mathfrak{D}_1(x,y)
\end{aligned}
$$

One can now rearrange the terms and identify $\mathfrak{S}_1$ and $\mathfrak{S}_2$, which only depend on x and y respectively and $\mathfrak{D}_1$, which only depends on mixing terms of x and y.
The same formalism could now be applied to decompose S(S_0, F_0, F_ano, θ, t) into individual parts.

$$
\begin{aligned}
\mathbf{S}(S_0, F_0, F_{ano}, \theta, t) = &\ I + \mathfrak{S}_1(S_0, t) + \mathfrak{S}_2(F_0, t) + \mathfrak{S}_3(F_{ano}, t) + \mathfrak{S}_4(\theta, t) \\
&+ \mathfrak{D}_1(S_0, F_0, t) + \mathfrak{D}_2(S_0, F_{ano}, t) + \mathfrak{D}_3(S_0, \theta, t) \\
&+ \mathfrak{D}_4(F_0, F_{ano}, t) + \mathfrak{D}_5(F_0, \theta, t) + \mathfrak{D}_6(F_{ano}, \theta, t) \\
&+ \mathfrak{T}_1(S_0, F_0, F_{ano}, t) + \mathfrak{T}_2(S_0, F_0, \theta, t) + \mathfrak{T}_3(S_0, F_{ano}, \theta, t) + \mathfrak{T}_4(F_0, F_{ano}, \theta, t) \\
&+ \mathfrak{Q}_1(S_0, F_0, F_{ano}, \theta, t)
\end{aligned}
$$

Note that *I* denote the ice sheet state around which the Taylor expansion is performed (equivalent to f(a,b)) and vanishes for differences of simulations. As a quick additional explanation, the term $\mathfrak{S}_1(F\_0, t)$, only carries the ice sheet response due to the change of the baseline forcing, while $\mathfrak{D}_4(F\_0, F\_ano, t)$ only carries the response which occurs due to the interaction of baseline forcing and the forcing anomalies.

To study the impact of the choice of the RCM forcing, one wants now to isolate the terms depending on the baseline forcing from the rest. This can be achieved by looking at the difference between two simulations which only differ in F_0. It is important to mention that this requires the same initial state S_0 and parameter configuration θ, since otherwise terms containing S_0 or θ (but not the baseline forcing) would not vanish in the difference. In other words, if we were to compare simulations starting from different initial states, they

would observe different driving stresses, which would lead to different outcomes. One could label this initial state uncertainty. For our study we want to avoid having different initial states for different RCMs, because that would also mean we would have translated some of the RCM difference into the initial state as well.

Nevertheless, for projections, one often assumes some form of equilibrium state under pre-industrial conditions. This means all terms not containing the time dependent F_ano are time independent and could be summarized to the equilibrium initial state. The impact of the choice of RCM baseline forcing would then be described by terms containing both the RCM baseline (F_0) and the time dependent anomalies (F_ano). Those terms could be estimated by comparing different "projections" after subtracting them by their control run.

*In summary, I propose two key recommendations: (i) improve the initialisation procedure for the PD-equilibrium experiment, and (ii) reconsider the approach and methodology employed in the future projections experiment. These suggestions aim to positively contribute to refining the study's methodology for a more robust outcome. I align with the authors on the significance of elucidating and quantifying uncertainties in Antarctic projections related to surface mass balance, particularly those arising from regional climate models. Therefore, I believe that the study holds significant value for the scientific community and would be well suited for the scope of The Cryosphere. However, some major issues need to be addressed to make it a valuable contribution. Also, it is important to acknowledge that adequately addressing these recommendations would require rerunning the entire set of experiments, impacting not only the results but also reshaping the manuscript and its core findings.*

Concluding on the major comments, we thank the reviewer for the extensive review and fruitful comments and will briefly summarize our major actions here:

I) We improved the initialization setup for PD-equilibrium experiments, by performing individual thermal initializations for every RCM forcing.
II) We improved our projections by performing a 300-year PI-spinup simulation forced by the mean of all RCMs together with PI anomalies, to absorb the initial shock and achieve ice sheet states with considerably smaller model drift than before which is now in-line with e.g. the ISMIP6 Antarctica model drift.
III) We additionally improved the overall interpretability of our model results by assessing the isolated impact of the choice of RCM in interaction with GCM anomalies.

**Specific points**

1. *Abstract, l. 10: It is not clear here what is meant by 'underlying ice sheet model parameterization'. Please clarify for better understanding.*

We refer here to the applied parametrization as representations of physical processes as well as the used parameters in those parametrizations.

New formulation "[...] what is estimated from uncertainties related to ice sheet and climate models." (l.10)

2. *Abstract, l.8-9: 'Uncertainties in future sea-level predictions of 8.7 (7.3-9.5) cm …' --> I find this sentence confusing, as uncertainties are mentioned, but it looks like the sea-level prediction and their uncertainties are presented. I think that it would be helpful to clarify what the numbers between brackets represent.*

We now only mention the model mean.

3. *Introduction, l. 24: Include a reference to Goelzer et al. 2020 ISMIP6 projections when comparing GrIS and AIS sea-level projections by 2100.*

In our study we only look at the AIS response to different RCM-forcing thus we include the Seroussi et al. 2020 ISMIP6 community paper.

4. *Introduction, l.28: I'd suggest adding more references to the concept of calibration reducing uncertainties in sea-level projections, such as, e.g., Edwards et al., 2019, Coulon et al., 2023, Nias et al., 2019, Lowry et al., 2021.*

Added. (l.29)

5. *Introduction, l.29: please check these numbers.*

We did and they are correct., Compare to Seroussi et al. (2020) under 4.4: "Runs with HadGEM2-ES lead to significant sea level rise, with a mean ice mass loss of 96 mm SLE (standard deviation: 72 mm SLE)" and "Runs performed with CCSM4 show the largest ice mass gain, with a mean gain of 37 mm SLE (standard deviation: 34 mm SLE)"

6. *Introduction, l.31-32: I'd suggest adding a reference to Coulon et al. 2023 here, as they investigate uncertainties in ice-ocean and ice-atmosphere interactions.*

Added. (l.34)

7. *Introduction, l.46: I'd suggest specifying that there is no specific reason to exclusively use one model given that other RCMs such as MAR are also designed to simulate polar regions by accounting for these processes.*

We clarified this.: "However, there is not a specific reason to exclusively use one model since other models are also built to simulate polar regions by taking those processes into account (Mottram et al., 2021)." (l.48)

8. *Introduction, l.48: Seroussi et al. showed the influence of the choice of the GCM used to derive the forcing on Antarctic projections and not on its equilibrium state. Also, I don't think that they isolated the specific influence on the SMB, as the oceanic forcings also vary for each GCM.*

They did not isolate that. However, they showed that projections heavily depend on the applied GCM. Sentence added :

"Furthermore, Seroussi et al. (2020) showed that there is also a significant impact of GCM differences in future projections." (l.51)

9. *Methods, l. 65: refer to Mottram et al. 2021?*

Added.

10. *Figure 1: It is not clear to me from the caption what exactly is represented in the second line (figures f—j). I'd suggest clarifying this in the caption and maybe also in the figure itself.*

It shows the difference between ERA Interim and the RCMs. We have clarified this: "SMB of the ERA-Interim dataset (Dee et al., 2011) (f) and SMB differences between [...]"

11. *Methods, l. 105: refer to Figure 2 here.*

Added.  (l.104)

12. *Methods, l. 112: an 8-km resolution was mentioned above, please clarify.*

The applied forcing is on 8 km resolution, while we perform our PD-equilibrium ice sheet simulations on 16 km resolution. Projections are carried out on 8 km resolution.

13. *Methods, l. 117: when is this evaluated? At the end of the 30-ky run? Please clarify, as the calibration step remains a little bit unclear so far. Also, in Table 2, an ensemble of 54 simulations is presented, which ones are the 14 selected ones? Maybe highlight them in bold in the table? It could also be interesting to visualise the obtained equilibrium ice-sheet geometry for each, maybe in the supplementary material.*

For our parameter ensemble we choose parameter combinations which after 15 ka of constant RACMO present-day forcing produced ice sheet geometries and velocities sensibly close to present-day observation. To minimize overfitting to the RACMO model and account for a general fit to present day observations we kept the ensemble spread relatively broad as seen in Figure 3. However, a small parameter bias to the RACMO forcing cannot be entirely excluded. In contrast one could perform a parameter selection individually with each of the four RCM forcings and then combine all parameters. Nevertheless, this would be computationally very costly.

Table 2 only sates the individual values for the changed parameters, not all the combinations. We have added table B1 and D1 to the Supplements explicitly denoting every used combination.

14. *Methods, l. 138: I do not understand how the computation is rendered more cost-effective. Please clarify.*

The formulation used by us is unclear here, the computation is not rendered more cost efficient but there is just less computation to be done using our proposed methodology. We will specify this. "This not only reduces the computational cost, but also mitigates the possibility [...]" (l.141)

*15. Methods, l. 143: was an 8-km resolution also used for the thermal spin-up?*

The thermal spinup was also performed on a 16 km ice sheet model resolution as geometry is fixed so resolution does not really matter too much..

*16. Methods, l. 143: I am confused by the abrupt shift from the fixed geometry thermal spin-up under present-day RACMO climatology to the RCM + 1860 anomaly climate for the historical spin-up. Why not start from an equilibrated state, i.e., as in the PD-equilibrium experiment, but for the 1860 climate, as is performed in e.g., Reese et al., 2023, Li et al., 2023, or Klose et al., 2023? Could the authors comment on this, and ideally show the model drift when applying constant 1860 climate for the ensemble of simulations?*

As described above we now perform a PI-thermal spinup and additionally present PI-control simulations. The drift under PI forcing is illustrated in Figure 6 in the revised manuscript.

*17. Methods, l. 145: Does HadGEM2-ES has projections outputs available until 2300 under RCP2.6, 4.5 and 8.5? If not, how are the projections extended to 2300?*

Projected HadGEM2-ES until 2300 was provided by Anthony Jones for Sutter et al. 2023 (https://www.nature.com/articles/s41558-023-01738-w) .

*18. Methods, l. 151: Could the authors comment on why the list of ensemble parameters in Table 2 differs from the 'PD-equilibrium' experiment and what guided this choice? Also, I understand that configurations without long-term stability are no longer excluded, it would be good to clarify which ones are the ones selected by the calibration procedure.*

We perform PD-equilibrium simulations on 16km while we perform projections on 8km resolution. Therefore, other parameters were selected. The applied parameters for the projectionswere selected to show ice volume and grounding line position close to present day observation after the end of the 300 years inital relaxation under constant PI forcing. Additionally, we only selected parameters which features limited model drift (<0.15mm/yr) as requested by the Reviewers. All applied parameter combinations are stated in the table B1 and D1 in the supplements.

*19. Methods, l. 151-152: What is meant by 'model spin up' here? Is the thermal spin up, or the short historical run? Please clarify.*

Thermal spinup+historical run.
We clarified this: "Please note that since the model spinup (thermal + constant PI forcing) we chose here is relatively simple [...]" (l. 161 & l.166)

*20. Methods, l. 153: What initial ice-sheet configuration is referred to here? If my understanding is correct, the ice-sheet initial state obtained from the thermal spin-up was produced with fixed ice-sheet geometry. Deviations with respect to ice thickness should therefore be zero.*

After the thermal spinup, ice thickness deviations are zero. The sentence is a little misleading, since we wanted to state that we expect the model to have relatively large ice thickness deviations between a simulated present-day state and observations, since no . inversion was performed. We clarified this in the revised version as we now state: "[...] model deviations with respect to ice thickness, between a simulated present-day state and observations, can be large which is typical for continental scale model setups not employing inversion (see e.g. Reese et al 2023)." (l.165)

*21. Methods, l. 155: 'have often been used in the past' --> I'd suggest adding some references to support this. I would also suggest clarifying what exactly is meant by 'simple spin-up' routines, is it the thermal spin-up?*

As stated above we refer with spinup to both the thermal spinup as well as the relaxation under constant PI forcing. We have clarified this in the revised manuscript. We additionally refer to the  Seroussi et al., 2019 and Levermann et al., 2020 were similar – albeit different integration times – setups were used.

*22. Figure 2, caption: 'First the model is initialized from present-day ice sheet observations. Then a 200-ka thermal spin up is performed.' à My understanding was that the initialisation was the thermal spin up itself. Here, it is implied that the spin-up is performed after a first initialisation procedure. Please clarify.*

With initialization here we mean that we regrid from the observational grid to our 16km PISM grid. From there on perform the thermal spinup.

The caption now reads as: "Starting from present-day ice sheet geometry, a 200-ka thermal spinup for each RCM forcing is performed individually. [...] "

*23. Figure 2: The figure says 'BEDMAP topography' while Bedmachine is mentioned in the manuscript, please correct. Also, I would suggest specifically writing on the figure that the thermal spin up is performed with a fixed ice-sheet geometry.*

We correct this in the figure and clarify in the figure caption: "During the thermal spinup, the ice sheet geometry is fixed." .

*24. Figure 3: I'd suggest clarifying in the figure caption that these are the timeseries under constant present-day conditions, i.e., the PD-equilibrium experiment. In addition, please clarify what change rate is meant in figures (e-h). Also, I suppose from the figures that (i-l) represent the change in ice fraction area? Finally, please clarify how the total ice mass change is translated in m s.l.e? Is only the ice above floatation accounted for here?*

(e-h) show the change rate of ice mass above floatation. (i-l) depicts the ice area extend relative to PD observations. Only the ice above flotation is account for here. We have clarified this in the caption.

25. *Results, l.168: Could this be influenced by the fact that the thermal spin up was performed with RACMO only? The trend would hence not be influenced by the RCM itself, but rather by the difference between the RCM surface temperature field and RACMO's one. Why not performing a thermal spin up for each RCM to exclude this possibility?*

See response above.

26. *Results, l.176: I'd refer to Figure 1 here.*

We have removed the sentence completely due to the new simulation results from our changed setup.

27. *Figure B3: Why not simply combine figures 3 and B3?*

Figure B3 was produced in reply to minor revisions by the Editor. For readability of Figure 3 we still kept this as additional information in the supplements.

28. *Results, l.183-184: What is meant by 'mainly driven by ice-sheet model parameterisation' here? I think that this requires more clarification.*

Due to the model intrinsic parametrization, especially the applied heuristics calculating the till friction angle, results in anomalies to present-day observations, seen in many studies (Martin et al. 2011, Albrecht et al. 2020, Sutter et al. 2023, Reese et al. 2023).
We have specified this in the revised manuscript, which now reads: "All our simulations show a strong negative ice thickness anomaly in the WAIS which is mainly driven by ice sheet model parameterization. Specifically, the applied heuristics calculating the till friction angle results in anomalies with respect to present-day observations. This is a persistent model bias for the setup employed here and in other studies (Martin et al., 2011; Albrecht et al., 2020a; Sutter et al., 2023, Reese et al., 2023)". (l.195-198)

29. *Results, l.188: '(effect of ice sheet model spin up and parameter choices)' --> Again, I think that this requires a bit more explanation.*

As stated above we refer to the effect of the applied parameterizations and chosen parameters, as well as the initial state (e.g. ice- geometry and temperature).
In our decomposition introduced above that would be mainly D(theta, t) and AD(S_0, theta, t). Nevertheless, we have clarified this in the manuscript together with the points mentioned above. "The inter-model differences caused by the different RCM-forcings (mainly the impact of SMB forcing differences) are around four times smaller compared to the overall model bias (effect of ice sheet model spinup and parameter choices mentioned above)." (l.201-203)

30. *Results, l.190-193: Alternatively, a control run under constant present-day climate conditions used for the thermal spin up could be deduced from each simulation from the ensemble, allowing to isolate changes in the AIS due to the evolving climate for each configuration.*

We have adressed this point with the PI-control simulations in the revised manuscript.

31. *Results, l.195: It could be interesting/helpful to the reader to highlight, on one or several figures, some of the regions/locations that you refer to in the text.*

Since there are already a lot of details in the Figure, we try to avoid overloading the Figure with additional information.

32. *Figure B4: It is not clear to me what exactly is represented in Figure B4.*

Figure B4 (a) shows the mean state of the ice sheet after 30 ka when forced with the mean out of the four RCM forcings. (b) shows the difference between the ice sheet thickness depicted in (a) compared with the mean of the simulations forced with the RCMs individually. We have updated the Figure caption accordingly.

33. *Results, l.205: Why were these specific simulations selected? Where do they lie compared to the rest of the ensemble?*

They were chosen, because they showed a collapse under one RCM forcing but not under another. Since we now do not discard simulations due to grounding line retreat on millenial timescales under RACMO forcing, all simulations will also be depicted in Figure 3.

34. *Figure 5: Writing the parameter values in each of the subfigures is confusing as it gives the impression that each parameter value is associated with the panel itself, I would suggest removing it. In addition, please clarify in the caption what experiment is represented in the figure.*

We removed the parameters from the figure since they are named in the caption.

35. *Results, l.212: please clarify what is meant by 'similar' here.*

Collapse with one forcing but no collapse with another. For clarity, we removed the restriction on 100 kyrs stability with RACMO forcing, such that we won't have to separate cases anymore.

36. *Results, l.216-219: in which figures can we see this? Please clarify. It would also probably be easier to indicate the parameter-set subset on the figure directly.*

Figure C1. In the revised manuscript, we have removed this figure, since all simulations showing WAIS collapse under one forcing but not under another are shown in Figure 5.

37. *Results, l.213-219: as a few of these simulations do not seem to have reached a steady state nor a quasi-steady-state yet, one could wonder whether running these simulations for more than 30kyr would lead to a WAIS collapse in all of the configurations, implying that the committed ice-sheet state is mainly driven by the parameter set itself, while the RCM climatology modulates the timing of the*

*potential collapse? This is only a guess, but it could be interesting to discuss this somewhere?*

We added this to the discussion. "We therefore can not exclude a potential WAIS collapse at a later stage (i.e. after the initial 30kyrs of our simulations). This would imply that the committed ice sheet response is mainly driven by the parameter set itself, while the RCM climatology might modulate the timing of the collapse. However, this hypothesis would require longer simulations which are beyond the scope of this study." (l.373-376)

38. *Results, l.223-224: Figure D1 seems like an important figure which, I believe, has its place (along with its discussion) in the main manuscript.*

We have moved the Figure to the main body of the manuscript. Now Figure 6 in Section 3.2.

39. *Results, l.242-243: SMB over the ice shelves has no direct contribution to sea-level rise, but it does indirectly influence the ice-shelves stability and hence buttressing effect on the ice-sheet flow. Maybe it is worth briefly commenting on this?*

In the revised manuscript, we now correct for the PI-control when we calculate Δslr_max, due to these direct differences in SMB should already be accounted for such that we removed the sentence the reviewer refers to here. In general, the importance of the local SMB for the grounding line becomes apparent in Section 3.2.4 and is discussed in Section 4.2.

40. *Figures 7, D2-D3: I see no purple line on these figures. Also, the grey line does not seem to be the observed present-day grounding-line position. Are these the median grounding line positions? Also, are these the ice-sheet configurations by the end of the simulations, i.e., 2300? Please clarify.*

The purple line was removed for readability in an erlier version of the manuscript. We therefore adjusted the caption. The black line indicates the simulated grounding line position.

41. *Figures 7, D2-D5: I find the use of the difference to the common mean hard to read and interpret. Alternatively, a control run under constant pre-industrial climate conditions could be deduced from each simulation, allowing to isolate changes in the AIS due to the evolving climate for each configuration (something similar is performed in Li et al.'s Exp. CMIP6_RAW_1850-2100).*

We added Figure D5 showing the ice thickness difference to the PI control runs to the revised manuscript. However, the aim of Figure D2-D5 (in the old manuscript) is to show the difference between the RCM forcings. Comparing with the control run also shows the differences due to the GCM anomalies (compare with discussion above), which can be significantly larger than the difference due to the RCM choice. Please also notice the changed colobar scale in Figure D5 in the revised manuscript.

*42. Results, l.267-269: I think that this makes sense, given that the parameters included in the ensemble do not have a strong impact in this region, which is instead strongly influenced by the SMB.*

We removed the word. (l.296)

*43. Results, l.276: What about the control (i.e., constant present-day as of 2005) simulations? It could be interesting to show these as well to have a better grasp of the influence of this signal.*

We have added a panel showing the PI control simulation.

*44. Figures D4-D5: It should be clarified in the figures' captions that these represent the ensemble member 10 only. Also, what do the different coloured lines represent in these figures? Overall, it would be good to clarify figure captions throughout the manuscript.*

Thank you for pointing that out. We clarified it in the caption.

*45. Results, l.281-282: I don't think that I would say that the RCM baseline will 'significantly affect the onset and pacing of a marine ice sheet instability'. First, I don't believe that Figure 8, given that it is a snapshot at year 2300, allows us to draw conclusions about the pacing itself. In addition, except for the 5 and 95 percentiles, the grounding line positions are overall relatively similar. I think that it is more correct to say that the choice of the RCM baseline modulates the grounding line retreat. Also, I don't think that it makes sense to refer to a marine ice sheet instability mechanism here. We do not know whether a self-reenforcing retreat has been triggered. I would simply refer to a grounding-line retreat.*

We will remove the statement on "pacing" of the grounding line retreat as well as the usage of "marine ice sheet instability" here as we didn't assess whether our model results show a self-reenforcing retreat.

We rewrote the paragraph not drawing a conclusion on pacing and MISI. However, we would like to mention here that the model response strongly indicates an ongoing self-sustained retreat based on our experience from other studies.

*46. Figure 8: It is not clear to me how the percentiles of grounding-line positions are calculated. Could the authors specify it in the caption?*

We calculated grounded ice mask density, which states for every gridbox i,j the percentage of simulations which have grounded ice. From there contour lines were drawn for the individual percentiles. We will add a precise description on how those percentiles were calculated to the caption.

We describe how the percentiles were calculated in the caption now.

*47. Results, l.287: Here it is referred to ensemble member n°10 while before the ensemble members were referenced using letters (AY, etc.), maybe consider using either letters or numbers for both for consistency?*

We clarified this.

We know have labeled every parameter configuration with numbers and state all of them in table D1 and B1.

*48. Results, l.288-289:'similar to the already observed patterns in the present-day equilibrium runs' à I am not sure which figure I should refer to for the comparison, I am guessing Figure 4, but it would be good to specify.*

Indeed Figure 4, we will mention this in the text.

We removed this section from the manuscript since in the updated manuscript it wont add any additional insight.

*49. Discussion, l.304-305: I think that this formulation is clearer than the one used in the abstract, maybe use an equivalent sentence as this one for the abstract as well? Also, I think that it is important to specify that these are the maximum differences between two RCM configurations.*

We updated the abstract guided by the sentence in the discussion.

The sentence now reads as: "Our simulations suggest differences in projected Antarctic sea-level contributions, due to the choice of present-day SMB and temperature baseline forcing of [...]". (l.320-321)

*50. Discussion, l.306-308: I find that how both (different) numbers are compared is confusing, as, e.g., 8.7(7.3 – 9.5) represents a spread in sea-level contribution, while 9.6 +- 7.2 represents the sea-level contribution itself. I'd suggest presenting the spread of the ISMIP6 ensemble instead. The authors may also consider calculating an equivalent indicator as the 'mean maximum sea level contribution difference' on the ISMIP6 ensemble for a more robust comparison.*

We now compare with the IMSPI6 spread directly. However, since the ISMIP6 data contains simulations from different ice sheet models employing different parameters and parameterisations, a calculation of a 'mean maximum sea level contribution difference' as described in our Methods is not possible.

*51. Discussion, l.311: My impression is that the uncertainty presented here is instead mainly driven by the initialisation procedure. I think that this requires a more thorough discussion and presentation of control (i.e., constant pre-industrial climate) simulations.*

As discussed above we performed PI-control runs for the the revised version. We use those control simulations to correct for when calculating Δslr_max as described above. Therefore,

we are now quite confident that the uncertantie we present in the revised manuscript is quite independent of the initalisation procedure.

52. *Discussion, l.352: 'may be simulated'. It is in fact only for specific RCM and parameter set that divergences appear. Your median grounding line positions are in fact relatively similar.*

We are unsure to what exactly the reviewer refers in l.352.

53. *Discussion, l.329: what is meant by 'unforced' grounding-line retreat here?*

We mean not forced by GCM projection anomalies.

We removed the word "unforced". (l.357)

54. *Discussion, l.341&345: Grounding-line retreat does not necessarily imply reduced buttressing and hence acceleration in ice flow…*

We will correct this and only mention potential reduced buttressing.

Sentence now reads as: "[...] which might lead to reduced buttressing [...]" (l.368)

55. *Discussion, l.355: I don't understand what is meant by 'the ice sheet gradually responds to the SMB forcing', please clarify.*

Indeed, this formulation is misleading. We mean a linear response to the SMB forcing e.g. ice thickness increases for SMB increase and vice versa.

Changed accordingly to: "[...] the ice sheet model responds in line with the SMB forcing [...]" (l.384)

56. *Discussion, l.358-359: I don't understand this. The evolution of the ice flow can be investigated with the evolution of the ice velocities through time.*

We now have checked the velocity fields. The ice divides are shifting in all runs in response to the applied forcing. However, a more in-depth analysis would be necessary to assess if this is the main driver of the observed behavior.

We now state that "a more in-depth analysis would be necessary to assess if this is the main driver of the observed behavior". (l.388)

57. *Discussion, l. 360: I find this title a little confusing. I would suggest reformulating it.*

We simplified to: Parameter sensitivity

58. *Conclusion, l.377-378: I don't think that the differences in thickness and grounding line positions that are presented here may be considered as 'considerable' (see, especially, Figure B1)*

It is true that RCM induced differences in thickness and grounding line positions are not always considerable when compared with difference to present-day observation. However, there are cases where this is the case (see Figure 5). Additionally, RCM induced differences

can reach values up to several hundred meters in some regions, which is in absolute and relative terms quite considerable.

59. *Figure D1: How come that Figure D1 shows only one curve for the control runs? What parameter values are used for these control runs? For consistency, control simulations should be performed for each parameter configuration.*

We have performed control simulations for every parameter configuration. For simplicity Figure D1 only shows the ensemble mean of all control runs per RCM forcing.

We now state this in the Figure Caption (now Figure 6) to avoid confusion.

60. *Appendix D, l.399: 'minor ice loss' --> I am not sure that the 1860-2005 ice loss (of several dm) can be considered 'minor'. It is the same order of magnitude as the sea-level contribution between 2100 and present under RCP8.5, as shown in Figure 6. Also, as mentioned above, I suggest moving this entire section to the main manuscript.*

Indeed, the ice loss is not minor if compared with ice-loss until the year 2100. Most of this ice loss can probably be attributed to the initialization shock when the ice starts to evolve freely. With our new simulation setup, we absorb this shock beforehand and reach a considerable smaller drift (cmp. with Figure 6 in the revised version) . However, we would like to point out here, that there are many publications where the SLE difference to PD in the respective spinup/initialisation/control experiments amounts to several meters. Compared to this several dm can be considered small.

*Overall,*

61. *the methodology, particularly outlined in Section 2.2, is unclear. The inconsistent use of terms such as 'spin up' and 'initialisation' makes it challenging to comprehend the precise procedures, even with the aid of Figure 2, especially for the 'Future projections' experiment (section 2.2.2). Similarly, the calibration procedure, and how it varies between experiments (resulting in different parameter values) remains unclear. To enhance clarity, the study would benefit from a clear list of experiments, similar to Table 1 in Li et al. (2023), where climate forcing, initial conditions, and objectives are explicitly stated.*

We added a table describing all simulations performed in the appendix of the revised manuscript (Table B2). Further we now ensure consistent use of the terms "initialization" and "spinup".

We now use "initialization" only for the initial step starting PISM from BedMachine obervation as in line 106:"In both cases, the model is initialized from the BedMachine [...]" and "spinup" when we refer to the thermal spinup.

Since our simulation setups have changed, we also reworked Section 2.2.1 and 2.2.2.

62. *the figure captions should be enhanced for consistency, providing clear information on the represented experiments, years, and the significance of various elements (e.g., grounding-line position). Improved consistency and clarity in figure captions would enhance the overall understanding of the figures and contribute to a more straightforward interpretation of the study's findings.*

We have reworked the figure captions ensuring, we mention all neccesary information to understand the figure.

63. *the discussion lacks consideration and comparison with related works (other than ISMIP6).*

The comments by the reviewer have brought up several important publications which are now included and discussed in the revised manuscript.

E.g. : Coulon et al., 2024; Li et al., 2023; Lowry et al., 2021; Reese et al., 2020;   Bulthuis et al. 2019

**Minor comments/Typos**

64. *Abstract, l.1: remove coma after 'impacts'.* --> Adapted
65. *Abstract, l.7: 'constant forcing quasi-equilibrium state' --> I find this formulation confusing, try to rephrase?* --> We removed "constant forcing" for clarity
66. *Abstract, l.8: 'uncertainties of' --> uncertainties in?* --> Adapted
67. *Abstract, last sentence: remove coma after 'importance'.* --> Adapted
68. *Introduction, l. 17: add come after 'Until the end of this century'* --> Adapted
69. *Introduction, l. 17: 'see level rise'* --> Adapted
70. *Introduction, l. 25: 'century's' --> centuries* --> Adapted
71. *Introduction, l. 32: 'The latter, estimates' --> 'Uncertainties in estimates of'?* --> Adapted
72. *Introduction, l.38-41: I'd suggest splitting this sentence in two.* --> Adapted
73. *Introduction, l.50: I'd suggest splitting this sentence in two: 'We address the following questions:…'* --> Adapted
74. *Methods, l. 70: 'drainage basis'* --> Adapted
75. *Methods, l. 70: remove come after 'All four models'* --> Adapted
76. *Methods, l. 83: 'togeher'* --> Adapted
77. *Methods, l. 86: 'Antarctic Ice sheet' --> 'Antarctic Ice Sheet' for consistency. I believe that this is the case at other places in the text, please check.* --> Adapted
78. *Methods, l. 90: 'shelf's'* --> Adapted
79. *Methods, l. 100: to improve the readability of this sentence, consider using 'two model set ups: (i) …, and (ii) …'.* --> Adapted
80. *Methods, l. 102: 'scenario' --> 'scenarios'.* --> Adapted
81. *Methods, l. 102: 'BedMachine'.* --> Adapted
82. *Methods, l. 104: remove come after (2004).* --> Adapted
83. *Methods, l. 112: 'on 16 km resolution' à 'at 16 km resolution'.* --> Adapted
84. *Methods, l. 113: 'RCM-'* --> Adapted

85. *Methods, l. 113: 'we employ' --> 'we run/produce'?* --> Adapted
86. *Methods, l. 118: 'An additional constrained'* --> Adapted
87. *Results, l. 165 and l.172: 'initialization shock' --> 'initial shock'?* --> Adapted
88. *Results, l. 229: 'maxmimum'* --> Adapted
89. *Results, l. 244: 'SMB The accumulated…'* --> We reworked the paragraph due to which the sentence got delted
90. *Figure 8, caption: 'siumaltions'* --> We have changed the caption text
91. *Results, l. 285-286: remove comes after 'both' and 'forcing sets'* --> We have reworked the paragraph due to which this sentence got deleted.
92. *Results, l. 286: 'chosen ice sheet model parameter choice'.* --> We reworked the paragraph due to which the sentence got delted
93. *Discussion, l.300: 'onto' --> 'on'?* --> Adapted
94. *Discussion, l.322: 'forcing data' --> 'baseline climatology'?* --> Adapted
95. *Discussion l.341&345: 'butsstressing'* --> Adapted
96. *Discussion l.345: 'In these simulation'* --> We reworked the paragraph due to which the sentence got delted

Since we expect out Manuscript to change quite significantly, we will implement those comments unless the text passage hasn't been changed.

In conclusion, we would like to thank the Reviewer for his extensive and detailed comments. We are convinced that our proposed changes will significantly improve the manuscript.

**References**

*Coulon et al.: Disentangling the drivers of future Antarctic ice loss with a historically-calibrated ice-sheet model, EGUsphere [preprint], https://doi.org/10.5194/egusphere-2023-1532, 2023.*

*Goelzer, H., Nowicki, S., Payne, A., Larour, E., Seroussi, H., Lipscomb, W. H., Gregory, J., Abe-Ouchi, A., Shepherd, A., Simon, E., Agosta, C., Alexander, P., Aschwanden, A., Barthel, A., Calov, R., Chambers, C., Choi, Y., Cuzzone, J., Dumas, C., Edwards, T., Felikson, D., Fettweis, X., Golledge, N. R., Greve, R., Humbert, A., Huybrechts, P., Le clec'h, S., Lee, V., Leguy, G., Little, C., Lowry, D. P., Morlighem, M., Nias, I., Quiquet, A., Rückamp, M., Schlegel, N.-J., Slater, D. A., Smith, R. S., Straneo, F., Tarasov, L., van de Wal, R., and van den Broeke, M.: The future sea-level contribution of the Greenland ice sheet: a multi-model ensemble study of ISMIP6, The Cryosphere, 14, 3071–3096, https://doi.org/10.5194/tc-14-3071-2020, 2020.*

*Li et al.: Climate model differences contribute deep uncertainty in future Antarctic ice loss. Sci. Adv. 9, eadd7082 (2023). DOI:10.1126/sciadv.add7082*

*Nias, I. J., Cornford, S. L., Edwards, T. L., Gourmelen, N., & Payne, A. J. (2019). Assessing uncertainty in the dynamical ice response to ocean warming in the Amundsen Sea Embayment, West Antarctica. Geophysical Research Letters, 46, 11253–11260. https://doi.org/10.1029/2019GL084941*

*Klose, A. K., Coulon, V., Pattyn, F., and Winkelmann, R.: The long–term sea–level commitment from Antarctica, The Cryosphere Discuss. [preprint], https://doi.org/10.5194/tc-2023-156, in review, 2023.*

*Lowry, D.P., Krapp, M., Golledge, N.R. et al. The influence of emissions scenarios on future Antarctic ice loss is unlikely to emerge this century. Commun Earth Environ 2, 221 (2021). https://doi.org/10.1038/s43247-021-00289-2*

*Reese, R., Garbe, J., Hill, E. A., Urruty, B., Naughten, K. A., Gagliardini, O., Durand, G., Gillet-Chaulet, F., Gudmundsson, G. H., Chandler, D., Langebroek, P. M., and Winkelmann, R.: The stability of present-day Antarctic grounding lines – Part 2: Onset of irreversible retreat of Amundsen Sea glaciers under current climate on centennial timescales cannot be excluded, The Cryosphere, 17, 3761–3783, https://doi.org/10.5194/tc-17-3761-2023, 2023.*

**References:**

Albrecht, T., Winkelmann, R., and Levermann, A.: Glacial-cycle simulations of the Antarctic Ice Sheet with the Parallel Ice Sheet Model (PISM) – Part 2: Parameter ensemble analysis, The Cryosphere, 14, 633–656, https://doi.org/10.5194/tc-14-633-2020, 2020.

Martin, M. A., Winkelmann, R., Haseloff, M., Albrecht, T., Bueler, E., Khroulev, C., and Levermann, A.: The Potsdam Parallel Ice Sheet Model (PISM-PIK) – Part 2: Dynamic equilibrium simulation of the Antarctic ice sheet, The Cryosphere, 5, 727–740, https://doi.org/10.5194/tc-5-727-2011, 2011.

Seroussi, H., Nowicki, S., Payne, A. J., Goelzer, H., Lipscomb, W. H., Abe-Ouchi, A., Agosta, C., Albrecht, T., Asay-Davis, X., Barthel, A., Calov, R., Cullather, R., Dumas, C., Galton-Fenzi, B. K., Gladstone, R., Golledge, N. R., Gregory, J. M., Greve, R., Hattermann, T., Hoffman, M. J., Humbert, A., Huybrechts, P., Jourdain, N. C., Kleiner, T., Larour, E., Leguy, G. R., Lowry, D. P., Little, C. M., Morlighem, M., Pattyn, F., Pelle, T., Price, S. F., Quiquet, A., Reese, R., Schlegel, N.-J., Shepherd, A., Simon, E., Smith, R. S., Straneo, F., Sun, S., Trusel, L. D., Van Breedam, J., van de Wal, R. S. W., Winkelmann, R., Zhao, C., Zhang, T., and Zwinger, T.: ISMIP6 Antarctica: a multi-model ensemble of the Antarctic ice sheet evolution over the 21st century, The Cryosphere, 14, 3033–3070, https://doi.org/10.5194/tc-14-3033-2020, 2020.

J. Sutter, A. Jones, T. L. Frölicher, C. Wirths, and T. F. Stocker. Climate intervention on a high-emissions pathway could delay but not prevent West Antarctic Ice Sheet demise. *Nature Climate Change*, 2023. doi:10.1038/s41558-023-01738-w.

Ben Smith *et al.,* Pervasive ice sheet mass loss reflects competing ocean and atmosphere processes. Science**368**,1239-1242(2020). DOI:10.1126/science.aaz5845

Bulthuis, K., Arnst, M., Sun, S., and Pattyn, F.: Uncertainty quantification of the multi-centennial response of the Antarctic ice sheet to
climate change, The Cryosphere, 13, 1349–1380, https://doi.org/10.5194/tc-13-1349-2019, publisher: Copernicus GmbH, 2019

Reese, R., Levermann, A., Albrecht, T., Seroussi, H., and Winkelmann, R.: The role of history and strength of the oceanic forcing in sea level projections from Antarctica with the Parallel Ice Sheet Model, The Cryosphere, 14, 3097–3110, https://doi.org/10.5194/tc-14-3097-2020, publisher: Copernicus GmbH, 2020.

**Reply to comments by Christoph Kittel: The influence of present-day regional surface mass balance uncertainties on the future evolution of the Antarctic Ice Sheet (egusphere-2023-2233)**

**Summary of Changes**

We are grateful to C. Kittel for evaluating our work, and the valuable and constructive comments that help improve the manuscript. In response, we now

- Relax our restriction on stability for 100 kyrs under RACMO forcing for our parameter ensemble, to account for parameter configurations which might not work with RACMO but with other RCM forcings.
- Perform additional pre-industrial control runs and improve the spinup of our centennial simulations to ensure minimal model drift.

Below, we respond to C. Kittel's individual comments in detail and describe the actions we took to address them.

**Detailed response**

(Original report cited in italics)

*This study focuses on the impact of anthropogenic global warming on rising sea levels, specifically examining the Antarctic Ice Sheet (AIS). It underscores the crucial role of selecting appropriate regional climate model (RCM) references for predicting future sea level rise contributions from ice sheets. By using the Parallel Ice Sheet Model (PISM), the researchers find that the choice of RCM reference forcing introduces uncertainties in sea level rise predictions. Additionally, the study highlights how the choice of RCM reference influences grounding line retreat in West Antarctica.*

We thank, C. Kittel for evaluating our manuscript and his constructive and helpful comments. We address the raised points below and propose actions to clarify the points to further improve the manuscript.

*Overall the manuscript is clear but some sections could be improved (notably the Methods). The topic is interesting as the influence of the SMB baseline has not yet been assessed. Differences of less than 100Gt/yr ie lower than the annual variability (for instance between MAR and RACMO whose results are really close to the observations following Mottram et al., 2021) seem to lead to large mass differences.*

In order to address the raised points by all reviewers, the applied methodology (e.g. model initialization) has been revised. Reflecting this we updated and refined the Methods section of our manuscript.

**Major comments**

*The paper is worth publishing, but I'm particularly concerned about the initialisation method and wonder to what extent the results are influenced by it. Like most models, PISM was originally calibrated to RACMO over Antarctica, and its development was based on this forcing. We can already assume that part of the model's behaviour is linked to RACMO (or at least a similar SMB field). The idea of redoing calibrations with other parameters seems to*

*me to be an interesting way of overcoming this problem, but I have the impression that the results are still essentially influenced by RACMO and PISM's intrinsic behaviour, and therefore favour differences as soon as another forcing is applied. Only "good" calibrations for RACMO (or giving correct results and stability under the RACMO forcing) are conserved while other combinations could work for other models but not for the RACMO forcing. My concern is that the authors want to analyse the influence of the SMB between different models, but that they rely heavily on one of the models in question.*

To address the fact that we have chosen the parameter combinations for our ensemble from calibration against the RACMO model, we removed the restriction on long term (100kyrs) stability under RACMO forcing. By doing so, we now account for parameter combinations which won't work for RACMO but might work for other RCM forcings.

To further, decrease the dependency of the simulation outcome on the thermal initialization performed with the RACMO forcing, we performed individual thermal spinups for every RCM forcing as we described in our reply to Reviewer 1.

Furthermore, as the reviewer stated, PISM is often used in combination with RACMO forcing. Therefore, it is possible that some of the parameters we didn't touch in our simulation might have a bias towards the RACMO model, in the case RACMO was used in the calibration process (however, they might also be informed by Greenland ice sheet model experiments or just set to some standard values used in the community). However, as PISM is a complex model it is unfeasible to individually recalibrate all parameters for this study. In conclusion we had to limit ourselves to parameters governing or affecting ice flow, grounding line behavior, ice shelf mass balance etc.. This is reflected in the choice of the "flow parameters" (sia_e, pQ), basal friction and the ice-ocean heat exchange coefficient (gamma), which have the strongest influence on the evolution of the ice sheet. Again, It is important to note that many default model parameters might also represent Greenland conditions or are simply initial guesses. The selection of the baseline parameter combinations derives from a longer history of PISM studies and mostly deviates from the default parameter settings when running PISM as a black box.

In addition to this the reviewer is right that in our initial parameter selection (described in Section 2.2.1 we use RACMO as a forcing, which could lead to priming our configurations towards the RACMO model. However, as apparent from Figure 3 or R1, for every RCM forcing there are parameter configurations which produce ice sheet configurations close to present day observations. We additionally, don't conclude that any RCM yields "better" results in any form, since the goal of this study is just to quantify how differences between those RCM products might influence ISM simulations.

*The ensemble set could be enlarged by keeping the combinations that also work for another forcing, and a comparison of the best combinations for each forcing would also allow us to see how a less good combination influences the results. It would also be possible to take the*

*ideal calibration for one forcing and apply it to the others, to see how PISM responds in this case. What about PISM bias? Despite the different calibrations, isn't there a PISM component in the results? If PISM has a tendency to discharge the ice too slowly or too quickly (poor discharge due to poor basal or dynamic ) obviously a "better adapted" SMB will always work better, especially when only the parameters that fit a model are kept.*

As already described above we aim to enlarge the ensemble by keeping parameter combinations which might not work for long term stability under RACMO forcing, but potentially work under different forcings.

Concerning the PIMS bias, we agree that PISM, as any other ice sheet model, tends in some regions to discharge ice to slow or fast. Nevertheless, since we perform all simulations with PISM, we assume that all results contain the same ice sheet model specific bias. Since we then calculate the difference between the individual simulations, we can assume that the ice sheet model bias will vanish in first order. There, might still be a higher order bias one could tackle by repeating our study with different ice sheet models. However, this is beyond the scope of this study.

*Similarly, what is the impact of model drift? From figure d1 (a,b,c) (*which should be in the main text), only one simulation seems to have no drift. What happens to these differences if we remove the drift from PISM? For the scenarios with little warming, apart from MAR it looks like most of the differences could be caused by drift alone. Is it the drift of the model itself or also the result of an ice sheet that was out of balance at the start of the simulation because of another shape? This drift or imbalance would then be less significant in the simulation with a stronger anthropogenic forcing (rcp8.5).*

Figure D1 (a,b,c,) illustrates the ice mass change (in meters sea level equivalent) for the historical as well as the RCP scenarios. The dashed lines indicate control runs with the individual RCM forcings and constant 2005 HadGem2-ES anomalies. It is correct is that a significant portion of the absolute ice mass change, especially in the RCP2.6 and RCP4.5 scenario, occurs as well in the control run. Since we don't expect the AIS to be necessary in steady-state under 2005 climate conditions, we would also not assume the control run to be constant. In addition, both the control run as well as the RCP scenario simulations might be affected by a model drift. To have a better understanding of the model drift in our simulation setup, we now perform additional PI-control simulations as we have also described in the reply to Reviewer 1 and 3. The updated simulation results are depicted in Figure 6. As described above we applied those control runs as corrections for our calculations of $\Delta slr\_max$.

The aim of this study is to investigate how different RCM forcings affect the evolution of the ice sheet. To do so we calculate the maximum difference in modelled sea level contribution ($\Delta slr\_max$) given in equation 4. In first order approximation $\Delta slr\_max$ should be unaffected by the model specific drift, since we only look at differences of individual simulations and not at absolute ice mass change.

***Specific and minor comments***

***P3L66*** *: Please refer to Mottram et al., 2021 where MAR is described and not the dataset on Zenodo. I also encourage the authors to respect the data usage notice concerning MAR outputs that are available on Zenodo.*

Sincere apologies for failing to acknowledge the MAR team.  We updated the reference to the publication and not the stand-alone dataset and additionally acknowledge the MAR team.

*The authors use models with the same forcing (ERA-Interim), which is a good point. They refer to Mottram et al., 2021 (**P4 L75- L76**) for the comparison between these models. However, RACMO2.3p2 is a more recent version of RACMO than the one use in Mottram et al., 2021. I won't say that the conclusions remain valid.*

Thanks for spotting this. We will clarify this in the revised manuscript.

We now state: "A more detailed discussion and comparison of the applied RCMs can be found in Mottram et al. (2021). Please note that Mottram et al. (2021) used data from RACMO2.3p2 while this study uses RACMO2.3p3." (l.79-80)

***Figure 1f****: This is not SMB ERAint vs the Ensemble Mean, but rather the SMB from ERAinterim. Please check your caption as some of them are not clear.*

We will fixed this in the manuscript.

We now state: "SMB of the ERA-Interim dataset (Dee et al., 2011) (f) and SMB differences between".

*Generally speaking, I recognise that a lot of work has been done to free ourselves from the problems of initialisation and calibration depending on a single model, which is already a good thing, but I'm not convinced that we're free enough. I hope that the authors can improve this aspect of their study because I really think that this article is interesting and highlights the importance of multi-model studies.*

*Best regards,*

*C. Kittel*

**Reply to comments by Reviewer 3: The influence of present-day regional surface mass balance uncertainties on the future evolution of the Antarctic Ice Sheet (egusphere-2023-2233)**

**Summary of Changes**

We are grateful to the reviewers for evaluating our work, and the valuable and constructive comments that help improve the manuscript. In response, we now

- Perform individual thermal spinups for PD-equilibrium simulations.
- Perform additional pre-industrial control simulations.
- Calculate the maximum sea level contribution difference taking into account the control simulations.

Below, we respond to the reviewer's individual comments in detail and describe the actions we took to address them.

**Detailed response**

(Original report cited in italics)

*This paper explores the projections of sea-level rise (SLR) from the Antarctic Ice Sheet using the Parallel Ice Sheet Model (PISM), driven by Surface Mass Balance (SMB) forcing derived from four distinct Regional Climate Models (RCMs). Specifically, the study assesses the impact of these RCMs on SLR projections under the global Climate Model HadGEM2-ES. The research reveals that the choice of RCM reference forcing introduces uncertainties in future sea-level rise predictions, comparable to influential factors like ice sheet model parameterization and global climate model choices. Notably, the study emphasizes that the selection of the RCM can influence the timing of the West Antarctic Ice Sheet (WAIS) grounding line retreat under RCP8.5. A parallel investigation examines the present-day forcing from ERA 5 on the 30ka long-term stability for the four different RCMs.*

For clarification, we do not employ ERA5 forcing to drive our model. We employ Regional Climate Models which are driven by ERA-Interim boundary conditions.

*While the paper holds promise for publication, there is room for improvement in synthesizing the results, particularly regarding the equilibrium experiments. Further clarification is sought for the 2100 and 2300 experiments, with a specific focus on the rationale behind the SLR projection calculations and whether the numbers are subtracted by control runs.*

*Equilibriums runs:*

*While the same parameters tuned to RACMO yield different results for the other RCMs, I understand that it might be computationally prohibitive to conduct a spin-up for every RCM and parameterization. However, my concern lies in whether the obtained results convey physical insights. Typically, a glacial spin-up is undertaken to mitigate model shock, ensuring that projections are grounded in physical processes rather than numerical artifacts. Given this, I find it surprising that RACMO still exhibits considerable model shock.*

For our PD equilibrium simulations, we performed an initial thermal spinup using the RACMO forcing. In this thermal spinup we keep the ice sheet geometry fixed while letting the ice-temperature profile adjust. As discussed in our replies to Reviewer 1 and 2, this carries the risk that the thermal state is primed towards the RACMO model. Therefore, we have now performed individual thermal-spinups for every RCM forcing individually.

Please find the updated simulation setup in Section 2.2.1 and especially Figure 2 (Figure R2 above) of the revised manuscript.

After the thermal spinup, we simulated 30 kyrs of ice sheet evolution starting from the Bedmachine (Morlighem et al., 2020) geometry. Since we employ a wide range of

parameters, we assume that not all parameter configurations will be in immediate equilibrium with the initial ice sheet geometry. Therefore, we let the ice sheet simulation evolve for 30 krys such that the simulations approach equilibrium.

We are not sure if the Reviewer addresses our equilibrium simulations in the second half of the paragraph or if the Reviewer addresses already our model projections. To us it is unclear what the Reviewer means with "glacial spin-up". If the reviewer refers to a glacial thermal spinup in which the ice geometry is kept constant, but a transient glacial thermal forcing is applied to the ice sheet, we agree that this is another option to thermally initialize the ice sheet but it is not necessarily the typical standard option as can be seen in the variety of thermal spinup methods in ISMIP6 (Seroussi et al., 2020). The same also applies if the reviewer refers to a full glacial spinup which additionally allows the ice sheet to freely evolve over one or multiple glacial cycles. Nevertheless, this method would not be useful to just find an equilibrium state of the AIS under present day conditions. For future projection like simulations, it is an approach used by some studies but not the typical one, as also can be seen in the ISMIP6 model methodology.

Our revised spinup approach, suggested by the reviewers' comments, better reflects what has been done in previous studies including ISMIP6 reasonably well. As one of our main goals is to illustrate the impact of RCM-uncertainties in such model setups we are confident that this approach is suitable for the question at hand.

*Could you clarify whether there was a change in resolution from the glacial spin-up to the equilibrium run? If not, kindly include the 16km resolution in your experimental design details. Additionally, I am curious about the parameters utilized for the glacial spin-up.*

The thermal spinup was performed at 16 km resolution. The equilibrium simulations were also performed at 16 km resolution. The parameter configuration for the thermal spinup was [-pseudo_plastic_q 0.75 -topg_to_phi 8,30,-700,0 , sia_e=ssa_e=1 ].

*I am grappling with the interpretation of the results, uncertain about their physical significance versus numerical artifacts. It would be immensely helpful if you could articulate your key take-home messages from the equilibrium experiments for the reader's clarity. Notably, you mentioned that differences between RCM forcings are four times smaller than the overall model bias. In your opinion, can uncertainty be adequately captured by selecting just one RCM with an ensemble of ice sheet model parameters? The similarities between COSMO, RACMO, and HIRHAM raise questions about whether a recommendation for the future could be to choose MAR and one of the three RCMs to encompass uncertainty. Additionally, would you advocate for a separate glacial spin-up for MAR? These considerations could potentially enhance the abstract of your study.*

The PISM-specific model bias occurs in all simulations regardless of the applied forcing. To isolate the signal imposed by the individual forcings from this we calculated the ice thickness differences from the common mean Δh given in equation 3.

Our equilibrium simulations show two main findings. The different RCM forcings lead to different quasi-equilibrium states with over 2m of sea level equivalent ice mass difference, for the same parameter sets. Second, under the same parameters one RCM forcing might lead to strong non-linear responses, while another RCM forcing only exhibits minor changes, which is illustrated in Figure 5. We aim to mention this as well in the abstract of a revised version of the manuscript.

With regard to this finding, we would not generally agree that it is sufficient to look at a model with an overall high SMB (e.g., MAR) and another model with a lower SMB, since the distribution of the SMB plays a key role for regional ice sheet evolution, especially when it comes to nonlinear responses of the ice sheet. Figure 5 can be misleading in this case, since it gives the impression that there is only a difference between simulations forced by MAR and simulations forced by one of the other RCMs. Additional simulations shown in Figure C1 clearly show that there is not only a difference between MAR and the other models.

We now also performed simulations accounting for a thermal spinup with every RCM forcing individually. Results are illustrated in Figure R1 and R4 and incorporated into the revised version. In this new setup, we still observe differences in over 2 meters of ice mass above floatation (Figure R1). Additionally, we observe parameter combinations in which one forcing might trigger a strong nonlinear response while other forcings don't (Figure R4). Especially in Figure R4 it becomes apparent that it is not sufficient to only look at the highest (e.g. MAR) and lowest (e.g. COSMO) SMB regional model, since in both selected runs both of those forcings lead to a collapse of WAIS while intermediate SMB models (e.g. RACMO and HIRHAM) not necessarily show a WAIS collapse.

[Figure]

**Figure R4:** *Evolution of the sea level change relevant ice masses (a,f), for to individual parameter configurations, over the simulation period and ice thickness differences from the common mean as well as grounding line position (grey line) at the end of the simulation (b-e, g-j).*

*In Figure 4 I cannot see the purple line.*

Thank you for spotting this, we have removed the purple line from an earlier version of the manuscript to enhance readability. We have now adjusted the caption accordingly.

*Centennial Projections:*

*Regarding Figure 6: Could you confirm whether all Sea-Level Rise (SLR) contributions are subtracted by the control run? I might have overlooked this detail, and it would be helpful if you could explicitly state whether such subtraction has been performed. Notably, Seroussi et al. subtracted all the runs by control runs. Additionally, consider showcasing only the HadGEM2-ES results from Seroussi's work or, alternatively, emphasize the PISM run(s) for comparison.*

In Figure 6 in the original manuscript, we did not subtract the control run from the simulation. We discussed our reasoning for this in the reply to Reviewer 1. In summary, we primarily investigate simulation differences due to the RCM forcing. Subtracting by a control run which is driven with the same RCM forcing would subtract some of the RCM signal we want to investigate.

Nevertheless, as also discussed in our reply to Reviewer 1, the PI-control corrected mean maximum sea level contribution difference might be a better measure for the RCM induced uncertainty. Therefore, we will now show the PI-control corrected timeseries in Figure 7 of the revised manuscript. Additionally, we present Δslr_max with and without the PI-control subtracted. For the year 2100 this reduces our calculated uncertainties by one order of magnitude and for 2300 by around a factor of three. Accordingly, we note this in the results and discussion as well.

As we stated in the manuscript, our main goal is not to provide robust projections of Antarctic SLR contributions, but rather to assess the uncertainties due to the choice of different RCM reference forcings in such simulations. Therefore, we are more interested in the spread between simulations driven by different underlying RCM forcings, than the difference to the present-day ice sheet configuration. Nevertheless, we agree on the necessity of reasonable model-projections and the importance of bothabsolute SLR as well as the model drift. Therefore, we revised the simulation setup, described in the reply to Reviewer 1, Figure R3 as well as section 2.2.2 in the revised manuscript. In this new setup, we achieve minimal model drift (<0.15 mm/yr) compared to the current setup by performing a 300-year model relaxation under constant PI forcing. Additionally, we also performed control runs under pre-industrial forcing, for every individual RCM forcing. The time series of ice volume and the grounding line evolution for the PI control simulations can be found in Figure 6 and D5 in the revised version of the manuscript.

*On page 12, line 321, you mention calculating the maximum SLR contribution in a specific manner. I am curious about the choice of not subtracting the control run in this calculation. Considering that the glacial spin-up involved a single RACMO forcing, and parameter set, wouldn't it be necessary to subtract the control run for each member individually? Especially after the results obtained from the equilibrium runs show so different behaviour for each RCM. This consideration becomes especially relevant when examining projected SLR uncertainties. Could you conduct this subtraction and provide insights into how it influences the projected uncertainties? Based on Figure D1, it appears that the control runs might not align with the values from 2005, particularly noticeable in the year 2300. Further clarification on this aspect would be appreciated.*

We assume the reviewer refers to equation 4 on page 12, (line 232).
As we state in the manuscript, we calculate for every given parameter set the maximum difference in sea level contribution. Since there are four simulations for every parameter set, one for every RCM, the maximum sea level contribution difference is the difference between the simulations with the highest and lowest sea level contribution. Since, we have many different parameter configurations, we then calculate a mean as well as a min and max value over our ensemble. We interpret this number as an estimate of the maximum impact of the choice of an RCM forcing for SLR projections.

As stated above we agree on the importance of the control corrected ice mass change, which is why we now provide pre-industrial control runs. As stated above, subtracting the PI-control from our simulations has significant influence on the calculated values for Δslr_max. In accordance with suggestions by Reviewer 1 and 3 we now present the PI-corrected values as our main estimates for SMB induced uncertainties, while still presenting the uncorrected values for comparison.

The control runs in Figure D1 show the evolution of Antartica under constant 2005 climate, to contrast the significant changes due to further warming in the RCP scenarios. We do not necessarily expect our simulations to be in equilibrium with the 2005 climate. Consequently, we do not expect the control runs to stay at 2005 levels for the next 295 simulation years. Since we now perform PI-control simulations and used them to correct our projections, we removed the control simulations with constant 2005 forcing.

*Figure 7,9: which Year are you showing? I cannot see a purple line either.*

We show the year 2300. We have clarified this in the caption.

*Figure 9: Is there maybe a number to quantify this change? Mean thickness deviation for each RCM or something similar. This way we can see more easily if these difference arise more for the RCPs or RCMs.*

We thank the reviewer for this suggestion. We will calculate the mean thickness difference for each RCM. However, Figure 9 illustrates the thickness deviation from the common mean and not present-day observations. Nevertheless, calculating one scalar number would help to quantify the change of RCM influence for different RCP scenarios.

References:

Morlighem, M., Rignot, E., Binder, T., Blankenship, D., Drews, R., Eagles, G., Eisen, O., Ferraccioli, F., Forsberg, R., Fretwell, P., Goel, V., Greenbaum, J. S., Gudmundsson, H., Guo, J., Helm, V., Hofstede, C., Howat, I., Humbert, A., Jokat, W., Karlsson, N. B., Lee, W. S., Matsuoka, K., Millan, R., Mouginot, J., Paden, J., Pattyn, F., Roberts, J., Rosier, S., Ruppel, A., Seroussi, H., Smith, E. C., Steinhage, D., Sun, B., Broeke, M. R. v. d., Ommen, T. D. v., Wessem, M. v., and Young, D. A.: Deep glacial troughs and stabilizing ridges unveiled beneath the margins of the Antarctic ice sheet, Nature Geoscience, 13, 132–137, https://doi.org/10.1038/s41561-019-0510-8, number: 2 Publisher: Nature Publishing Group, 2020.

Seroussi, H., Nowicki, S., Payne, A. J., Goelzer, H., Lipscomb, W. H., Abe-Ouchi, A., Agosta, C., Albrecht, T., Asay-Davis, X., Barthel, A., Calov, R., Cullather, R., Dumas, C., Galton-Fenzi, B. K., Gladstone, R., Golledge, N. R., Gregory, J. M., Greve, R., Hattermann, T., Hoffman, M. J., Humbert, A., Huybrechts, P., Jourdain, N. C., Kleiner, T., Larour, E., Leguy, G. R., Lowry, D. P., Little, C. M., Morlighem, M., Pattyn, F., Pelle, T., Price, S. F., Quiquet, A., Reese, R., Schlegel, N.-J., Shepherd, A., Simon, E., Smith, R. S., Straneo, F., Sun, S., Trusel, L. D., Van Breedam, J., van de Wal, R. S. W., Winkelmann, R., Zhao, C., Zhang, T., and Zwinger, T.: ISMIP6 Antarctica: a multimodel ensemble of the Antarctic ice sheet evolution over the 21st century, The Cryosphere, 14, 3033–3070, https://doi.org/10.5194/tc-14- 3033-2020, publisher: Copernicus GmbH, 2020.

---

## Author Response (AR2)

**Reply to comments by the Reviewers: The influence of present-day regional surface mass balance uncertainties on the future evolution of the Antarctic Ice Sheet (egusphere-2023-2233)**

We thank all Reviewers for their time and effort in reviewing this manuscript. Please find our detailed response to the comments by the reviewers below.

(Original report cited in italics)

**Reply to Reviewer 1:**

*I appreciate the authors' efforts to implement the suggestions provided by the reviewers. I believe that this has significantly improved the quality of the manuscript. I recommend accepting the manuscript once the following minor comments are implemented:*
We thank the reviewer for the effort and the given detailed remarks and comments, which substantially improved the manuscript.

*- Figure 2 caption: ' Freely evolving ensemble simulations are indicated by solid lines' --> This is confusing as the thermal spinups with a fixed geometry are also represented by solid lines. Please adjust.*
Thank you for pointing this out. In detail the thermal spinup is indicated by a bold solid line, while the freely evolving simulation is indicated by a thin/normal solid line. To avoid confusion, we now explicitly state this in the caption.

*- l.222: refer to panels a—e*
Adapted

*- l.229: refer to panels f—j*
Adapted

*- Figure 6 caption: Specify again that the changes represented in this figure are changes in volume above flotation.*
Now reads as: "Time series of the median (solid lines) total ice mass above flotation change (a-c) […]".

*- l.233: The changes in grounded and floating ice areas are no longer represented, please adjust.*
We apologize for this mistake. The adjusted text now reads as: "The evolution of the total ice volume is shown in Fig. 6 […]".

*- section 3.2.2: I suggest clarifying in the text that delta_h represents the deviation from the common mean.*
We now quickly remind the reader that Δh measures the thickness deviation from the common mean as well as referring to Equation 3. Line 278 now reads like: "The spatial distribution of Δh

(thickness deviation from the common mean; cmp. Eq. 3) [...]"

*- l.307-308: This is indeed surprising. It would be interesting to provide an explanation/hypothesis for this behaviour.*

Indeed, this is extremely interesting. A potential explanation might be given by the fact that the regional SMB around Pine Island and Thwaites glacier is the smallest in COSMO (cmp. Fig. A2), which would explain why a collapse under PI-control conditions happens with COSMO as the baseline forcing in parameter configuration No. 12. The reason, why the PI-control simulation leads to a collapse while the projection does not, might be the fact that in the RCP scenarios an initial increase of local SMB for substantial parts of WAIS is observable, as illustrated in the Figure below for RCP2.6. This increase might again stabilize the WAIS, while the only moderately increasing ocean temperatures are not sufficient to destabilize Thwaites. However, we have not explicitly tested this hypothesis since it only appeared in one specific parameter configuration which in general seemed to be extremely sensitive to SMB differences across all scenarios (cmp. Fig. C6 – C9).

[Figure]

*Figure 1: Difference between 30-year averaged SMB in the RCP2.6 scenario between the period from 1860-1890 and 2085-2115.*

*- l.355: I suggest replacing 'large differences' by 'differences'.*
Adapted.

*- l.370: 'COSMOS' --> 'COSMO'*
Corrected.

*- l.370-371: I still don't see why a retreat of the grounding line would necessarily imply reduced buttressing. I suggest removing this sentence altogether.*
We agree with the reviewer, that there might not be a direct causality between a grounding line retreat and a reduction of ice shelf buttressing. What we refer to here is the fact that ungrounding will lead to reduced/zero basal friction and therefore acceleration of ice flow. We removed buttressing from the sentence which now reads: "[...] which leads to accelerated ice flow."

*- l.421: I suggest replacing 'large' by 'potentially large'.*
Adapted.

**Reply to Reviewer 2:**

*The authors study the influence of the initial SMB-baseline to force the ice sheet model PISM. Compared to their first version, I found the manuscript greatly improved. The authors gave satisfactory answers to all the reviewers' comments and included them into their manuscript. I only have a technical suggestion and recommend the publication of the manuscript in TC.*
We are grateful to Christoph Kittel for his insightful remarks and valuable comments, which have significantly enhanced the quality of the manuscript.

*About the method scheme, if one only reads the scheme, one could think that the SMB for projections is also a future SMB simulated by the RCM while it is still the present SMB. It is well explained in the method, it could perhaps be also highlighted in the scheme.*

To make this even clearer in the scheme itself, we adapted the Figure such that its clear the projections are performed by using the baseline RCM forcings together with HadGem2-ES anomalies.

*I'd also mention it again in section 3.2 (especially P12 - L232-234) as keeping the present SMB should have a strong impact on projections (SMB doesn't increase until ~ 2100 before decreasing).*

We now also remind the reader that our projections are done by adding GCM anomalies to the respective baseline RCMs . The section now reads as: "To investigate the effect of differences in the underlying RCM baseline data in climate scenarios we simulated the historical period from

1860 to 2005, followed by the RCP2.6, RCP4.5, and RCP8.5 scenario until 2300 by additionally applying GCM anomalies to the baseline RCM forcing on a 8 km grid resolution."

Similarly, P20 335, the authors mention that the RCM base-line selection could become even more important for long projections. I'd even go further in the implications as mentioned by other studies (eg., Coulon et al., 2024), the SMB itself should become more important compared to other processes such as basal melt. I think that the combination of RCM-baseline uncertainty and higher SMB importance would even increase the effect of the RCM-baseline uncertainty.

General, we agree that the importance of the RCM baseline uncertainty will increase with increasing impact of the SMB. Especially, for studies of the long-term (e.g., until the year 3000) commitment of Antarctic sea level contributions, this should be considered. One caveat would be that this also depends on the magnitude of basal melting which is crucial for ice sheet stability. Thus, the relative importance of SMB and basal melt would have to be considered on a case-by-case basis. For the period until 2300 a reference like the ISMIP6 ensemble will be instructive, such that one could better estimate the influence of the baseline SMB compared to other processed (e.g. basal melt).

I'd like to thank the authors for all their work.

Best regards,
C. Kittel

PS: I think there is a mistake in your answer to R1:

Integrated from 2015 to 2100, our PI-control simulations show a mean drift of 50939 Gt (MAR) -1098 Gt (RACMO) -8536 Gt (HIRHAM) and −46946 Gt (MAR)
I guess the last model should be COSMO.

This is indeed right. Apologies for this mistake and thank you for pointing this out. Correctly it should be *50939 Gt (MAR) -1098 Gt (RACMO) -8536 Gt (HIRHAM) and −46946 Gt (COSMO).*

**Reply to Reviewer 3:**
We want to thank the Reviewer for his work and his comments to improve this manuscript.

Minor changes:

Figure 6.
Remove black legend line "RCP"
Removed.

Figure 8:

*Please mention what runs are shown. e.g. all regional model and refer to the table that mentions the ensemble number 12*

The boxes in Figure 8 indicate the values within the 25th to 75th percentile, while the diamonds indicate all values outside of that interval. We now state this in the caption and additionally refer to Table C1.

Finally, we again want to thank Christoph Kittel, the two Anonymous Referees as well as Xavier Fettweis, for their thoughtful comments, which substantially helped to improve this manuscript.

Best regards,

Christian Wirths

(On behalf of all authors)